# Single-cell spatial multi-omics and deep learning dissect enhancer-driven gene regulatory networks in liver zonation

**Carmen Bravo González-Blas** [1,2], **Irina Matetovici** [1,3,4], **Hanne Hillen**[5,6], **Ibrahim Ihsan Taskiran**[1,2,3], **Roel Vandepoel** [1,2,3], **Valerie Christiaens**[1,2,3], **Leticia Sansores-García** [5,6], **Elisabeth Verboven** [5,6], **Gert Hulselmans**[1,2,3], **Suresh Poovathingal**[1], **Jonas Demeulemeester** [1,2], **Nikoleta Psatha**[1,2], **David Mauduit** [1,2,3], **Georg Halder**[5,6] **& Stein Aerts** [1,2,3] ✉

In the mammalian liver, hepatocytes exhibit diverse metabolic and functional profiles based on their location within the liver lobule. However, it is unclear whether this spatial variation, called zonation, is governed by a well-defined gene regulatory code. Here, using a combination of single-cell multiomics, spatial omics, massively parallel reporter assays and deep learning, we mapped enhancer-gene regulatory networks across mouse liver cell types. We found that zonation affects gene expression and chromatin accessibility in hepatocytes, among other cell types. These states are driven by the repressors TCF7L1 and TBX3, alongside other core hepatocyte transcription factors, such as HNF4A, CEBPA, FOXA1 and ONECUT1. To examine the architecture of the enhancers driving these cell states, we trained a hierarchical deep learning model called DeepLiver. Our study provides a multimodal understanding of the regulatory code underlying hepatocyte identity and their zonation state that can be used to engineer enhancers with specific activity levels and zonation patterns.

Cell identity is encoded by gene regulatory networks (GRNs), whereby transcriptions factors (TFs) bind to enhancers and promoters to regulate gene expression. Advances in single-cell technologies enable the simultaneous measurement of gene expression (single-cell RNA-sequencing (scRNA-seq)) and accessible chromatin (single-nucleus assay for transposase-accessible chromatin with sequencing (snATAC–seq)) within individual cells, providing opportunities to generate an unbiased view of the entire cell state space of a tissue and probe mechanisms of cell-type-specific GRNs[1–3].

Recent studies define a cell type as a (continuous) set of cell states, which can be aligned with the range of cellular phenotypes resulting from the interaction between a cell type and its microenvironment[4].

Certain cell types can be affected by their spatial location in the tissue, as in the mammalian liver. In each liver lobule, blood flows from the portal vein to the central vein, creating a gradient of nutrients, oxygen, hormones and morphogens that results in a highly variable environmental axis[5] (Fig. 1a). Hepatocyte function and metabolism varies depending on the position along this portocentral axis, as they are exposed to different microenvironments, a phenomenon known as zonation[5]. Previous single-cell and spatial transcriptomics studies have shown that not only hepatocyte function, but also their transcriptome, varies along this axis[6]. Yet, whether and how these variable states are encoded by genomic regulatory programs and enhancer logic, and how the zonation state interacts with the core hepatocyte GRNs are largely unclear.

[1]VIB Center for Brain & Disease Research, Leuven, Belgium. [2]Department of Human Genetics, KU Leuven, Leuven, Belgium. [3]VIB Center for AI and Computational Biology (VIB.AI), Leuven, Belgium. [4]VIB Tech Watch, VIB Headquarters, Ghent, Belgium. [5]VIB Center for Cancer Biology, Leuven, Belgium. [6]Department of Oncology, KU Leuven, Leuven, Belgium. ✉e-mail: stein.aerts@kuleuven.be

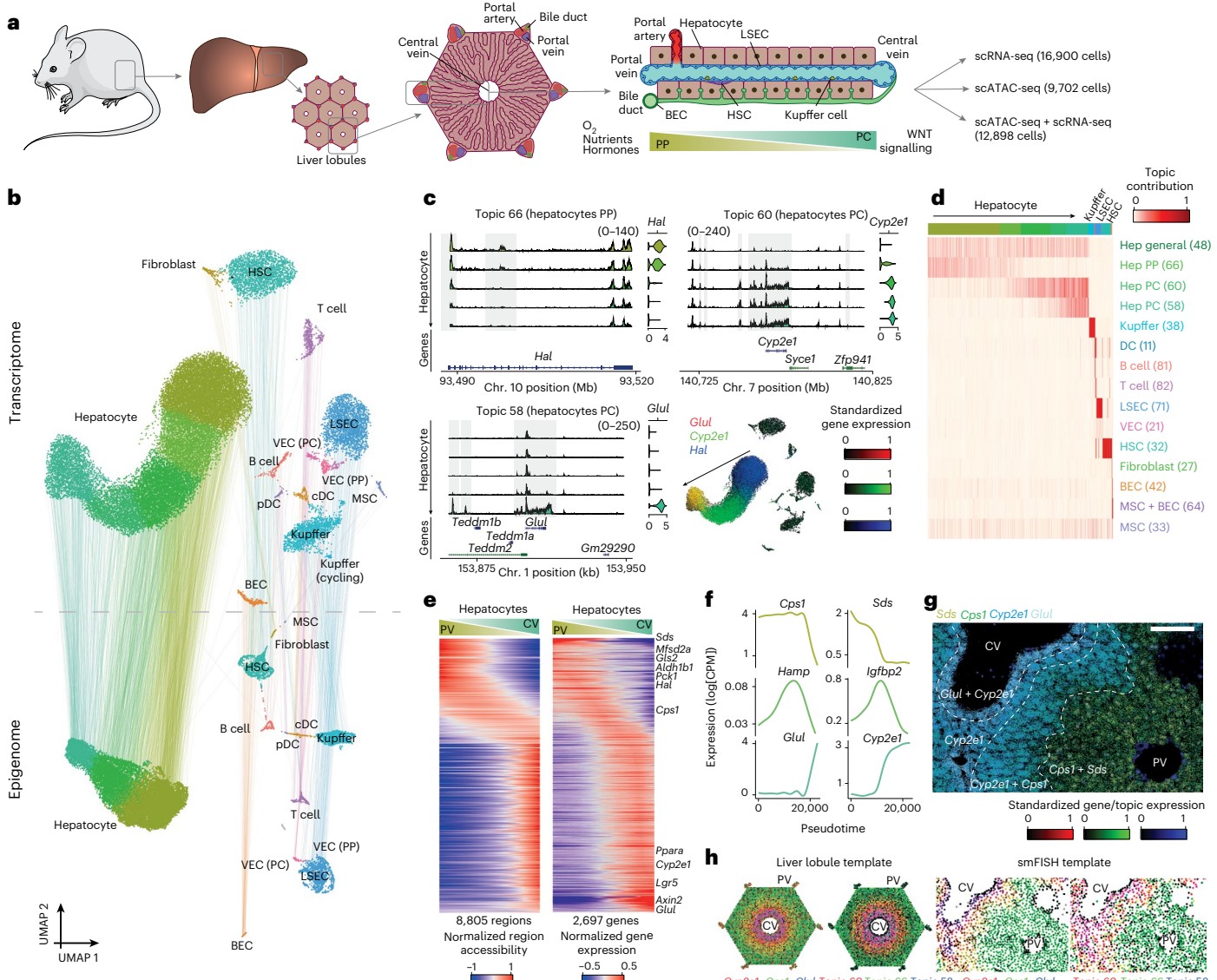

**Fig. 1 | A spatial single-cell multiome atlas of the mouse liver. a**, Overview of the mouse liver and the experimental set-up. The liver is composed of hexagonal structures called liver lobules, in which blood flows from the portal veins and hepatic arteries and drains in the central vein, creating a gradient of oxygen, nutrients, hormones and morphogens (such as WNT). **b**, Transcriptome- and epigenome-based uniform manifold approximation and projection (UMAP) projections (29,798 and 22,600 cells, respectively). The lines linking the UMAP projections connect the transcriptome and the epigenome UMAP positions from the same cell (profiled using single-cell multiomics). **c**, Pseudobulk chromatin profiles at different gene loci for hepatocyte zonation states, accompanied by violin plots representing the normalized gene expression of the relevant gene in each class. The UMAP projections show the gene expression of the relevant genes with RGB encoding. **d**, Cell topic contribution heat map. **e**, Normalized region accessibility and gene expression zonation heat maps. Cells are ordered by pseudotime (from periportal (PP) to pericentral (PC)), and regions and genes affected by zonation are shown (8,805 regions and 2,697 genes). The genes highlighted on the right of the gene expression heat map are located on the following ranked positions (from top to bottom): 3, 155, 201, 207, 233, 239, 866, 2,112, 2,535, 2,544, 2,556, 2,579. CV, central vein; PV, portal vein. **f**, GAM-fitted gene expression profiles for selected genes along the zonation pseudotime. CPM, counts per million. **g**, Liver section image showing smFISH profiles for *Glul*, *Cyp2e1*, *Cps1* and *Sds*. Three independent experiments were performed with similar results. Scale bar, 100 μm. **h**, ScoMAP liver lobule (4,498 cells) and smFISH coloured by gene expression and topic contribution using RGB encoding. For the transcriptome and epigenome data, cells from five and four biological replicates were combined, respectively. cDCs, conventional dendritic cells; pDCs, plasmacytoid dendritic cells. Source numerical data are provided as source data.

# Zonation drives cell state heterogeneity in hepatocytes

To characterize liver cell types and states at the transcriptome and chromatin level, we performed two 10x single-cell multiomics (ATAC + RNA) experiments in the mouse liver, resulting in a total of 12,898 high-quality cells (Methods). To improve the resolution, we also performed four 10x single-nucleus RNA-seq (snRNA-seq) and two 10x single-nucleus ATAC–seq (snATAC–seq) experiments, which provided an additional 16,900 single-cell transcriptomes and 9,702

single-cell ATAC profiles, respectively (Fig. 1a). From the snRNA-seq data, we obtained 5,863 unique molecular identifiers (UMIs) and 2,377 expressed genes per cell on average. The snATAC–seq data yielded 486,888 accessible regions that were grouped into 82 regulatory topics using pycisTopic[7] (Methods), with a mean of 12,083 unique fragments and 7,241 accessible regions per cell, a median transcription start site (TSS) enrichment of 16.1 and a fraction of reads in peaks (FRIP) of 66%. We observed similar overall quality between the multiome experiments and the independent assays (Extended Data Fig. 1). Both the snRNA-seq

and the snATAC−seq data distinguished the same cell populations, corresponding to 14 different cell types (Fig. 1b), including hepatocytes, hepatic stellate cells (HSCs), liver sinusoidal endothelial cells (LSECs), biliary epithelial cells (BECs), Kupffer cells, periportal and pericentral vascular endothelial cells (VECs), mesothelial cells (MSCs), fibroblasts and other immune cells (for example T cells, B cells and plasmacytoid/ conventional dendritic cells). Moreover, we found a significant correlation between the average chromatin accessibility around the genes and gene expression ($P < 2.2 \times 10^{-16}$; Extended Data Fig. 2a−e).

In all animals, we found a unidirectional gradient within hepatocytes, corroborated both by gene expression and region accessibility (Fig. 1c−f). To identify whether this gradient represents spatial variation along the portocentral axis, we performed single-molecule fluorescence in situ hybridization (smFISH) with a panel of 100 selected genes across cell types and cell states in the liver (Fig. 1g, Extended Data Fig. 2f−l and Supplementary Table 1). In hepatocytes, we identified three major zones that agree with the three regulatory topics found across the portocentral axis in hepatocytes; a *Glul*+ zone that comprises the hepatocytes surrounding the central vein (topic 58), a *Cyp2e1*+ zone that includes pericentral and mid-lobular hepatocytes (topic 60) and a *Cps1*+ zone that contains mid-lobular to periportal hepatocytes (topic 66). These zones are in agreement with earlier spatial transcriptomics studies[6,8]. We also identified a mid-lobular area where *Cyp2e1* and *Cps1* are co-expressed, also reflected by the overlap of topic 60 (pericentrally intermediate) and topic 66 (periportal) cell contributions. For each gene and region, we fitted a generalized additive model (GAM) across the pseudospatially ordered cells (from periportal to pericentral hepatocytes; Methods) and found 2,697 genes (out of 6,823 genes expressed in hepatocytes with at least 3 UMI counts in 10 cells) and 8,805 regions in hepatocytes (out of 14,005 shared hepatocyte regions) of which the respective expression and accessibility varies significantly along the portocentral axis (Fig. 1e,f and Methods; adjusted $P < 0.01$). Furthermore, the LSEC and HSC clusters were also represented as a gradient, reflected by distinct chromatin accessibility topics and gene expression (Extended Data Fig. 3). This integrated spatial and single-cell analysis also confirmed that *Ntn4*+ LSECs and *Ngfr*+ HSCs were located periportally, whereas *Kit*+ LSECs and *Spon2*+ HSCs were located pericentrally, as previously reported[9,10] (Extended Data Fig. 3). We identified 220 and 275 genes and 281 and 475 regions that vary along the portocentral axis in LSECs and HSCs, respectively (adjusted $P < 0.01$; Methods). Out of the 220 LSEC zonated genes, 69% overlap with the zonated liver endothelial markers described previously[11]. Moreover, BEC clusters could be clearly located in the bile ducts; pericentral and periportal VECs surround the corresponding vessels, together with fibroblasts; and Kupffer cells are preferentially located in periportal and mid zones without a strong zonation pattern, in agreement with recent studies[8]. Other immune cell types (such as B cells and T cells) are located across all zones (Extended Data Fig. 2k).

To map the whole transcriptome and epigenome into the smFISH spatial map, we implemented a new version of the R package Single-cell omics Mapping into spatial Axes using Pseudotemporal ordering (ScoMAP)[12], resulting in a simplified template of a liver lobule in which both gene expression and region accessibility can be visualized (Fig. 1h and Methods). We also identified an interesting batch effect in hepatocytes, related to differences in physiological state between the mice (Supplementary Note and Supplementary Figs. 1−5). In summary, our spatial single-cell multiome atlas of the mouse liver reveals that both cell type identity and cell states are congruently reflected at both the transcriptome and chromatin-accessibility level.

## Core hepatocyte GRNs are modulated by zonated repressor TFs

To identify candidate TFs underlying the different cell types and zonation states in the liver, we performed motif enrichment analysis in the different regulatory topics and differentially accessible regions (DARs)

across cell types using pycisTarget[7] (Methods). As motifs can often be linked to more than one TF (and, frequently, to several members of the same family), we pruned the list of annotated TFs by requiring a correlation of TF expression with motif enrichment. This resulted in the identification of HNF1A, PPARA, NFIA, NFIB, HNF4A, CEBPA, FOXA1 and ONECUT1 motifs in regions accessible across all hepatocytes, and TBX3 and TCF7L1/2 motifs in regions accessible periportally and pericentrally, respectively (Fig. 2a). Notably, *Tbx3* is expressed only in pericentral hepatocytes, whereas its candidate target regions are accessible only periportally. Vice versa, *Tcf7l1* is expressed periportally, and its candidate target regions are accessible pericentrally, whereas *Tcf7l2* is expressed in all hepatocytes (Fig. 2a). TCF7L1 is a paralogue of TCF7L2, the WNT-effector TF that is active pericentrally[13]; this may suggest that TCF7L1 and TCF7L2 bind to the same motif−TCF7L2 pericentrally for activation and TCF7L1 periportally for repression[14].

Next, we examined how the predicted TF-binding sites and enhancers are linked to candidate target genes. Following the SCENIC+ pipeline[7] (Methods), we compiled enhancer-gene regulatory networks (eGRNs) using as input pycisTopic's imputed chromatin accessibility, pycisTarget's TF cistromes (that is, a TF with its potential target regions) and the gene expression matrix. Using linear correlation and gradient-boosting machines, region−gene links (in a space 150 kb upstream and downstream of each gene) and TF−gene relationships were inferred. Using an enrichment analysis approach, we next assessed whether the TF-coexpression module significantly overlaps with the genes recovered from the motif/region-based links and, subsequently, retained the optimal set of target genes and regions for each TF. A TF with its set of predicted target enhancers and regions is called an enhancer regulon (eRegulon).

This analysis revealed 180 eRegulons, including SPI1, JDP2, RUNX1 in Kupffer cells; EBF1 and PAX5 in B cells; GATA3 and LEF1 in T cells; GATA4, MEIS1 and MAF in LSECs; LHX2 and TEAD1 in HSCs; WT1 in fibroblasts; and TEAD4, DOX4 and HNF1B in BECs, among others, and in agreement with literature[15−20]. We found general hepatocyte-specific eRegulons for CEBPA, HNF1A, HNF4A, ONECUT1, FOXA1 and NFIB[21−23], and zonation-associated eRegulons such as ESR1 and SOX9 periportally[24,25] and PPARG and AR pericentrally[26,27]. Importantly, SCENIC+ identifies TBX3 and TCF7L1 as transcriptional repressors in pericentral and periportal hepatocytes, targeting 193 and 520 regions and 77 and 119 genes, respectively (Fig. 2b). In other words, the chromatin regions in which accessibility is negatively associated with *Tbx3* expression are located nearby genes that are anti-expressed with *Tbx3* (same for *Tcf7l1*). Furthermore, we validated the SCENIC+ eGRNs using publicly available Hi-C data[28] and TF chromatin immunoprecipitation followed by sequencing (ChIP−seq)[29] (Extended Data Fig. 4a−e).

As we previously observed transcriptomic and epigenomic differences between the mice depending on their physiological state (Supplementary Note), we performed principal component analysis (PCA) of the eGRN enrichment scores to classify hepatocyte eRegulons on the basis of their zonation state and mouse specificity (Fig. 2c). This enabled us to identify eRegulons that depend on nutritional status, such as AGMAT[30] and MLXIPL[31]; on hormone levels, such as NR1I2, NR1I3 and RXRA[32]; and on circadian rhythm, such as CLOCK[33]. Some eRegulons were affected both by the physiological status of the mice and zonation, including ESRRA and FOXQ1 (periportal) and PPARA (pericentral). Among the general core (shared across all mice) eRegulons, we identified ONECUT1, CEBPA, HNF1A, FOXA1 and NFIB, while TBX3 and TCF7L1 are core pericentral and periportal (repressive) eRegulons, respectively (Fig. 2d). These physiological states are not independent of the hepatocyte core eGRN. For example, the HNF4A eRegulon, with 3,442 target genes and 26,127 target regions, contributes to both the core and the physiological-state-dependent programs (Extended Data Fig. 4f−i). This cooperativity, or blending, of the hepatocyte GRN with TFs controlling cell state is even stronger for zonation: 94.8% and 89.6% of the target regions of TBX3 and TCF7L1, respectively,

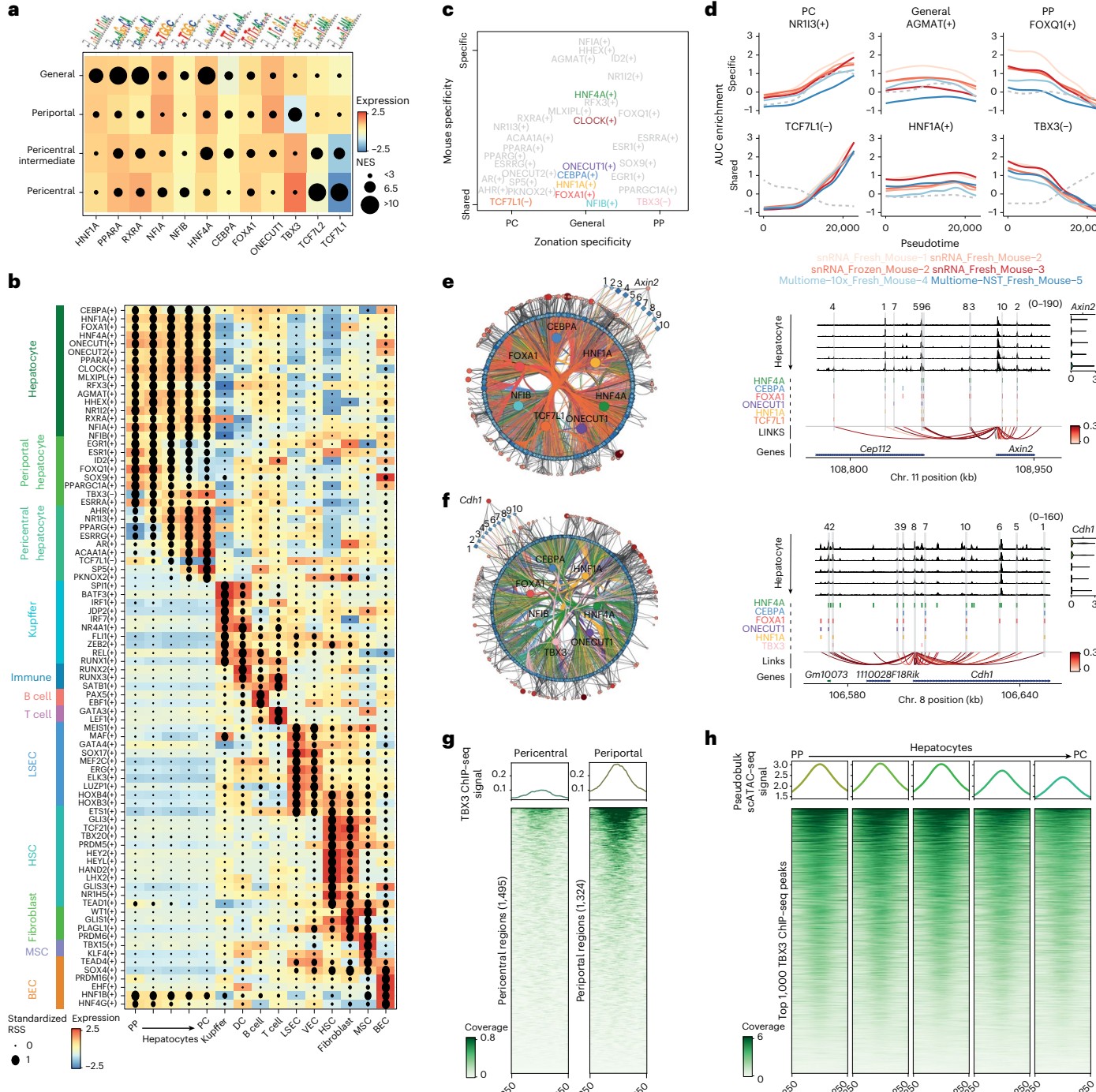

**Fig. 2 | Liver zonation is mediated by repression. a,** Highest normalized enrichment score (NES; circle size) for motifs linked to selected TFs in regions that are specifically accessible in different hepatocyte classes. The tiles are coloured by the expression of the corresponding TF in that hepatocyte class. **b,** SCENIC+ eGRN enrichment dot plot. The gene-based eRegulon specificity score (RSS) in the cell type is shown by circle size, and the colour represents the TF expression in the corresponding cell type. The symbol between brackets indicates whether the TF activates (+) or represses (−) its target genes. **c,** eGRN AUC based PCA plot, in which the first principal component represents the zonation specificity, and the second principal component represents whether eRegulons are shared across mice or specific to certain mice. **d,** GAM-fitted eGRN AUC profiles per mouse for selected eRegulons along the zonation pseudotime. The grey dotted line represents the GAM-fitted TF expression profiles. snRNA-seq samples are indicated in red, single-cell multiomics samples are indicated in blue. **e,** Pericentral core hepatocyte eGRN, with 265 pericentral marker genes and 1,439 regions targeted by the selected core TFs (with CRM score > 3) and conserved across mice. Hepatocyte

pseudobulk accessibility profiles (from periportal to pericentral) at the *Axin2* locus are shown as an example; with predicted TF-binding sites, region-to-gene links coloured by SCENIC+ correlation score and gene expression across the zonated hepatocyte classes (from periportal to pericentral). Regions in the core eGRN are highlighted in grey and numbered. Chr., chromosome. **f,** Periportal core hepatocyte eGRN, with 175 periportal marker genes and 972 regions targeted by the selected core TFs (with CRM score > 3) and conserved across mice. Hepatocyte pseudobulk accessibility profiles (from periportal to pericentral) at the *Cdh1* locus are shown as an example; predicted TF-binding sites, region-to-gene links coloured by SCENIC+ correlation score and gene expression across the zonated hepatocyte classes (from periportal to pericentral). Regions in the core eGRN are highlighted in grey and numbered. **g,** Coverage plot showing TBX3 ChIP– seq coverage at pericentral and periportal hepatocyte regions. **h,** Hepatocyte coverage at the top 1,000 TBX3 ChIP–seq regions. For the transcriptome and epigenome data, cells from five and four biological replicates were combined, respectively. Source numerical data are provided as source data.

overlap with the target regions of at least one of the general core TFs (HNF4A, HNF1A, CEBPA, ONECUT1, FOXA1, NFIB). This suggests that the hepatocyte zonation eGRN is a subset (or a layer) of the general hepatocyte eGRN. For example, among the candidate enhancers near the pericentral hepatocyte gene *Axin2*, we predict six TCF7L1 target regions. Among the candidate enhancers near the periportal gene *Cdh1* we predict two TBX3 target regions. In both cases, these regions are bound by additional core general TFs (Fig. 2e,f). Using publicly available scRNA-seq data from male and female mice, we confirmed that the hepatocyte core eGRN is not affected by sexual dimorphism (Supplementary Note and Supplementary Fig. 6). Together, eGRN inference shows that expression of periportal and pericentral genes is directly regulated at the chromatin level, contrary to recently published studies that focused on the accessibility of their promoters[34] (Extended Data Fig. 5a,b).

To further validate the repressive role of TBX3 in pericentral hepatocytes, we performed a ChIP–seq experiment against TBX3 on fresh mouse livers. In agreement with SCENIC+ predictions, the TBX3 ChIP–seq signal was stronger in periportal hepatocyte regions, and TBX3 ChIP–seq peaks were more accessible in periportal hepatocytes compared with in pericentral hepatocytes (Fig. 2g,h). Together with the TBX3 motif, we also found other hepatocyte motifs enriched in the TBX3 ChIP–seq regions, such as HNF4A, CEBPA, FOXA1 and ONECUT1, and a strong overlap with the ChIP–seq regions for these TFs, which further supports the interaction between the general and the zonated hepatocyte programs (Extended Data Fig. 5c–f).

In summary, SCENIC+ identified HNF4A, HNF1A, CEBPA, ONECUT1, FOXA1 and NFIB as core general hepatocyte TFs, and the repressors TBX3 and TCF7L1 as repressors of the zonation programs, together with additional networks related to the animal's physiological state.

## Enhancer sequence determines activity in hepatocytes

The enhancer and GRN predictions we have made thus far were fundamentally based on gene expression, chromatin accessibility and statistical motif enrichment. However, chromatin accessibility is not necessarily always associated with enhancer activity[35]. To assess whether the predicted enhancers are active, we performed a massively parallel reporter assay (MPRA) using a previously published enhancer-barcoding strategy[36] (Fig. 3a). We selected 10,845 genomic regions based only on their accessibility in hepatocytes, cloned them in a pooled manner (Methods), and transfected this library into the mouse liver (7 replicates) and human HepG2 cells (2 replicates) (Fig. 3a, Methods, Extended Data Fig. 6a–d and Supplementary Table 2). We chose HepG2 cells as an in vitro model based on the expression levels of the core hepatocyte eGRN TFs and the accessibility of the library enhancers (after liftover), in comparison to other mouse hepatocyte models such as AML12 and Hepa1-6 (Supplementary Note and Supplementary Figs. 7 and 8).

Regions that are accessible in hepatocytes show significantly higher enhancer activity compared with shuffled sequences (Fig. 3b,c),

and replicate MPRA analyses are strongly correlated (with a correlation ranging between 0.82 and 1; Extended Data Fig. 6a). We used the shuffled sequences as a background to derive an optimal activity cut-off (Methods), which classified 2,913 enhancers as active in at least one of the two systems (806 only in vivo, 921 only in HepG2 cells, 1,186 in both, adjusted $P < 0.1$; Methods) and 4,285 regions as inactive. In other words, 40.5% of ATAC peaks are active by MPRA, consistent with other studies[37]. Among the mouse regions that are active in vivo in the mouse liver, 64% are distal enhancers and 27% are promoters. By contrast, of the regions active in human HepG2 cells, 54% were promoters (Fig. 3d and Extended Data Fig. 6c).

The SCENIC+-predicted target regions of HNF1A, HNF4A, FOXA1, CEBPA, ONECUT1, NFIB and TCF7L1 are all significantly more active compared with the shuffled regions, with 45%, 39%, 43%, 39%, 35%, 32% and 26% of their predicted target regions active, respectively (Fig. 3e,f). TBX3 target regions are more active in vivo (20% and 5% of the regions are active in vivo and in HepG2 cells, respectively). In agreement with this, motif enrichment analysis of active versus inactive regions followed by classifier-based feature selection using random-forest models identified HNF1A, HNF4A, FOXA1, CREB and AP-1 motifs as determining features in active enhancers (Fig. 3g, Methods and Extended Data Fig. 6d,e). This motif-based classifier predicts enhancer activity with an area under the receiver operating characteristic curve (AUROC) of 0.71 (random AUROC, 0.54) and an area under the precision-recall curve (AUPR) of 0.44 (random AUPR, 0.27; Extended Data Fig. 6e).

## DeepLiver decodes hepatocyte enhancer grammar

To further scrutinize how enhancer logic underlies enhancer activity and zonation, we trained a hierarchical deep learning model, named DeepLiver. We first trained a convolutional neural network (CNN) to classify DNA sequences to the liver regulatory topics (called topic-CNN) using 219,823 annotated regions as input. The weights learned by the topic-CNN model were then used to initialize two additional CNNs, one to predict MPRA activity in vivo (MPRA-CNN) and another to predict zonation (zonation-CNN, using zonated accessibility classes as the output variable). This transfer-learning strategy overcomes the limited number of regions that we have available for activity (4,215) and zonation (4,181 pericentral, 1,372 periportal, 12,122 general) (Methods). For each model, the best epoch was selected on the basis of its accuracy and loss on the test data (10% of the input data; Extended Data Fig. 7a). The three models resulted in a higher AUROC and AUPR compared to a random control classifier, and topics associated with cell types had higher performance than low-contributing topics (Extended Data Fig. 7b–d). To validate DeepLiver predictions, we used a previously published MPRA dataset performed on synthetic sequences in vivo[38], finding that DeepLiver predictions correlate well with the experimental measurements ($R = 0.68$) (Extended Data Fig. 7e). Together, DeepLiver assigns given DNA sequences to cell types (represented by topics) in the liver and predicts activity and zonation patterns of hepatocyte enhancers (Fig. 4a).

**Fig. 3 | MPRAs in HepG2 cells and in vivo hepatocytes uncouple enhancer accessibility and activity. a**, Schematic of MPRA of the mouse liver. **b**, MPRA log$_2$-transformed fold change (log$_2$[FC]) for each enhancer class. $n = 9$ biological samples. The number of enhancers in each class is specified at the top. G, general; I, intermediate. **c**, The correlation between log$_2$[FC] values for high-confidence enhancers ($n = 7,198$) in HepG2 cells and in vivo coloured by enhancer type, with data ellipses indicating each group. **d**, The proportion of enhancer classes per high-confidence activity class. None, not active ($n = 4,285$); in vivo, active only in vivo ($n = 806$); HepG2, active only in HepG2 cells ($n = 921$); both, active in HepG2 cells and in vivo ($n = 1,186$). **e**, MPRA log$_2$[FC] per eRegulon. $n = 9$ biological samples. The number of tested enhancers in each eRegulon is specified at the top. **f**, The correlation between log$_2$[FC] values for high-quality enhancers ($n = 7,198$) in HepG2 cells and in vivo coloured by eRegulon, with data ellipses indicating

each group. **g**, Highest normalized enrichment score (circle size) for motifs linked to selected TFs in regions in the different enhancer (MPRA) activity classes coloured by the expression of the corresponding TF in HepG2 cells, in vivo or as the average (both and none). For the box plots in **b** and **e**, the centre line shows the median value, the top and bottom hinges represent the upper and lower quartiles, and the whiskers extend from the hinge to the largest and smallest values no further than 1.5 × interquartile range from the hinge, respectively. One-sided rank-sum Wilcoxon tests were performed to assess whether the log$_2$[FC] values of each group were greater than those of the shuffled regions. The asterisks represent the Bonferroni-adjusted $P$ values of the comparisons; ****, $P \le 0.0001$; ***, $P \le 0.001$; **, $P \le 0.01$; *, $P \le 0.05$; NS, $P > 0.05$. Seven and two biological replicates were used for in vivo and HepG2 cell experiments, respectively. Source numerical data are provided as source data.

Next, used DeepExplainer[39] to assess the contribution of each nucleotide in the enhancer classification, and TF-MoDISco[40] to identify motifs from recurring patterns in the contribution scores (Fig. 4b). For the MPRA-CNN, we identified patterns promoting enhancer activity corresponding to HNF4A, CEBPA, HNF1A, FOXA1 and AP-1 motifs, and several promoter-related motifs such as ETS, NRF1 and THAP. From the zonation model, motifs for HNF4A, ONECUT1, CEBPA, HNF1A, FOXA1,

CREB and NFIB were identified as regulators of accessibility across all hepatocytes; whereas TCF7L1/2 and TBX3 motifs are associated with pericentral and periportal accessibility, respectively, in agreement with our GRN-level analyses.

We further used DeepLiver to investigate the effect of sequence variation on enhancer specificity, activity and zonation. To validate DeepLiver in silico mutagenesis (Methods), we compared the predicted

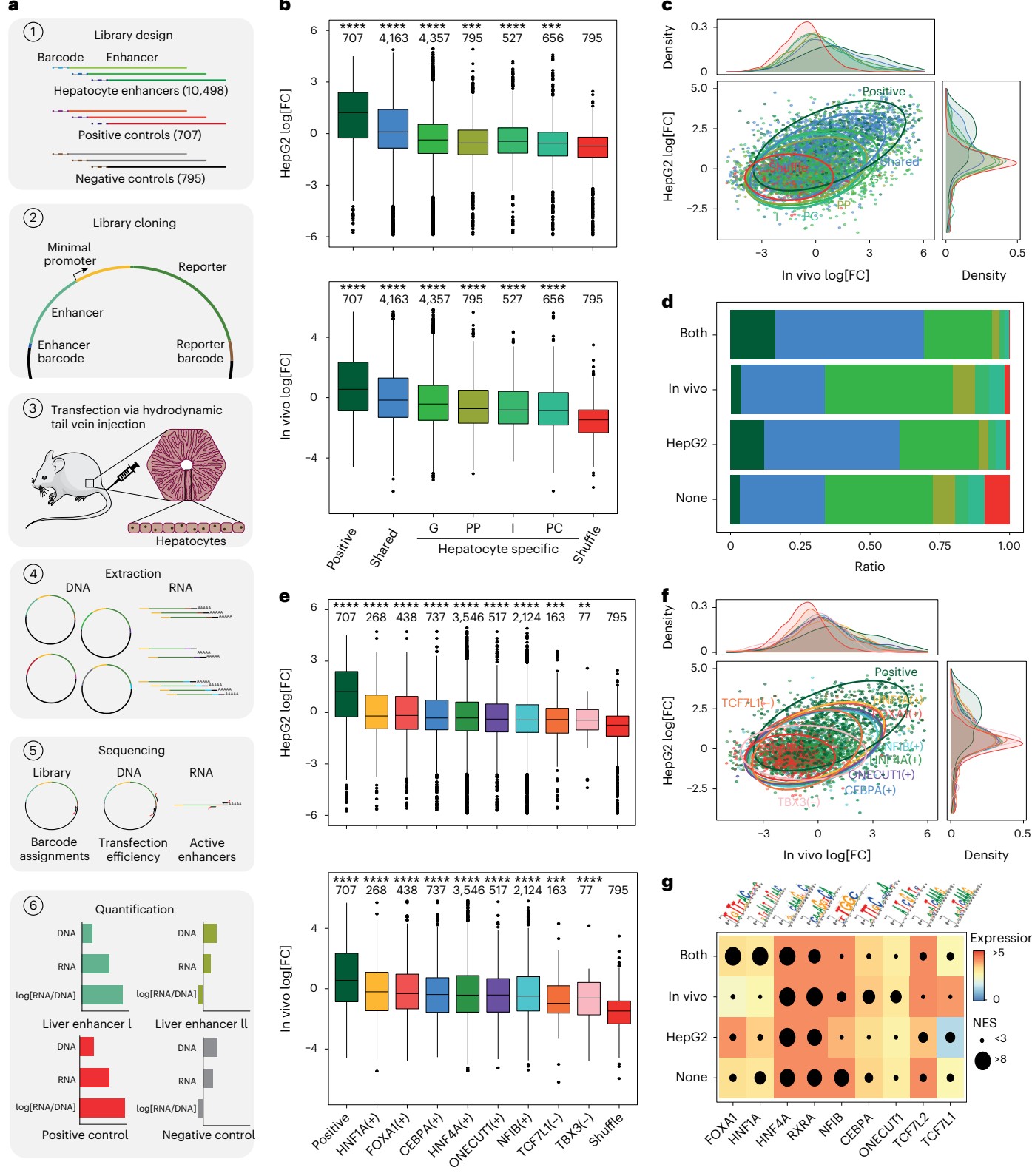

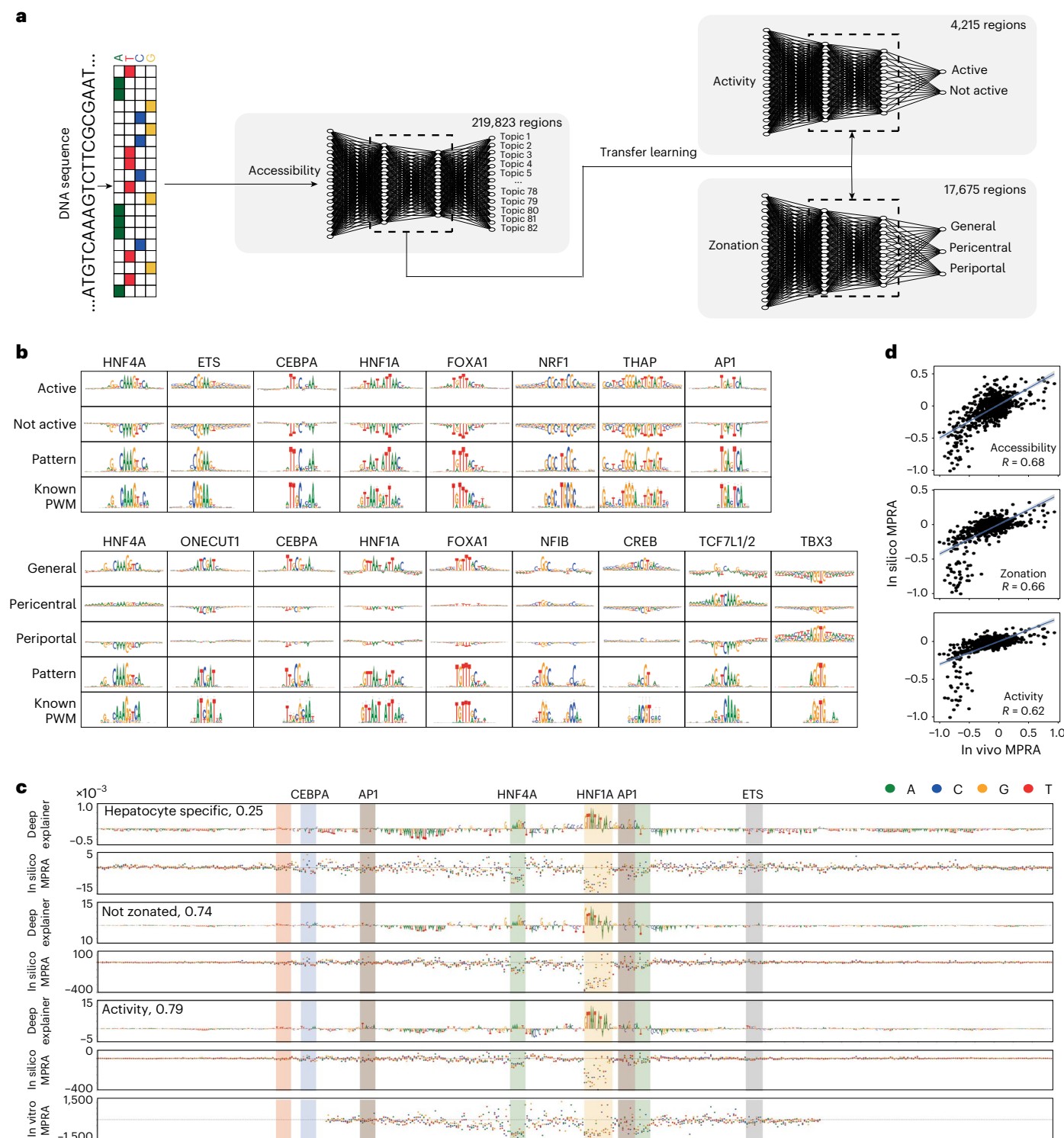

**Fig. 4 | DeepLiver decodes enhancer grammar. a**, DeepLiver overview. First, a CNN is trained to classify DNA sequences into their corresponding regulatory topic (219,823 sequences). The weights learned in the first model are used to initialize the activity and zonation models. The activity model classifies DNA sequences on the basis of their MPRA activity in vivo (using 4,215 high-confidence regions), while the zonation model classifies sequences on the basis of their zonation pattern on hepatocytes (pericentral, periportal or non zonated/general, using 17,675 regions for training). **b**, TF-MoDISco patterns identified in the activity and zonation models, with their contribution score per class and their most similar PWM from the cisTarget motif collection. **c**, DeepExplainer and saturation mutagenesis plots for the accessibility, zonation and activity models on an *Aldob* enhancer (hg19: chromosome 9:104195449–104195449), with motifs highlighted. Saturation mutagenesis, shown below, was performed in this enhancer previously[41]. **d**, The correlation between DeepLiver in silico mutagenesis and experimental saturation mutagenesis in the *Aldob* enhancer. The blue line represents the fitted linear regression and the grey bands represent the 95% confidence interval bands. Source numerical data are provided as source data.

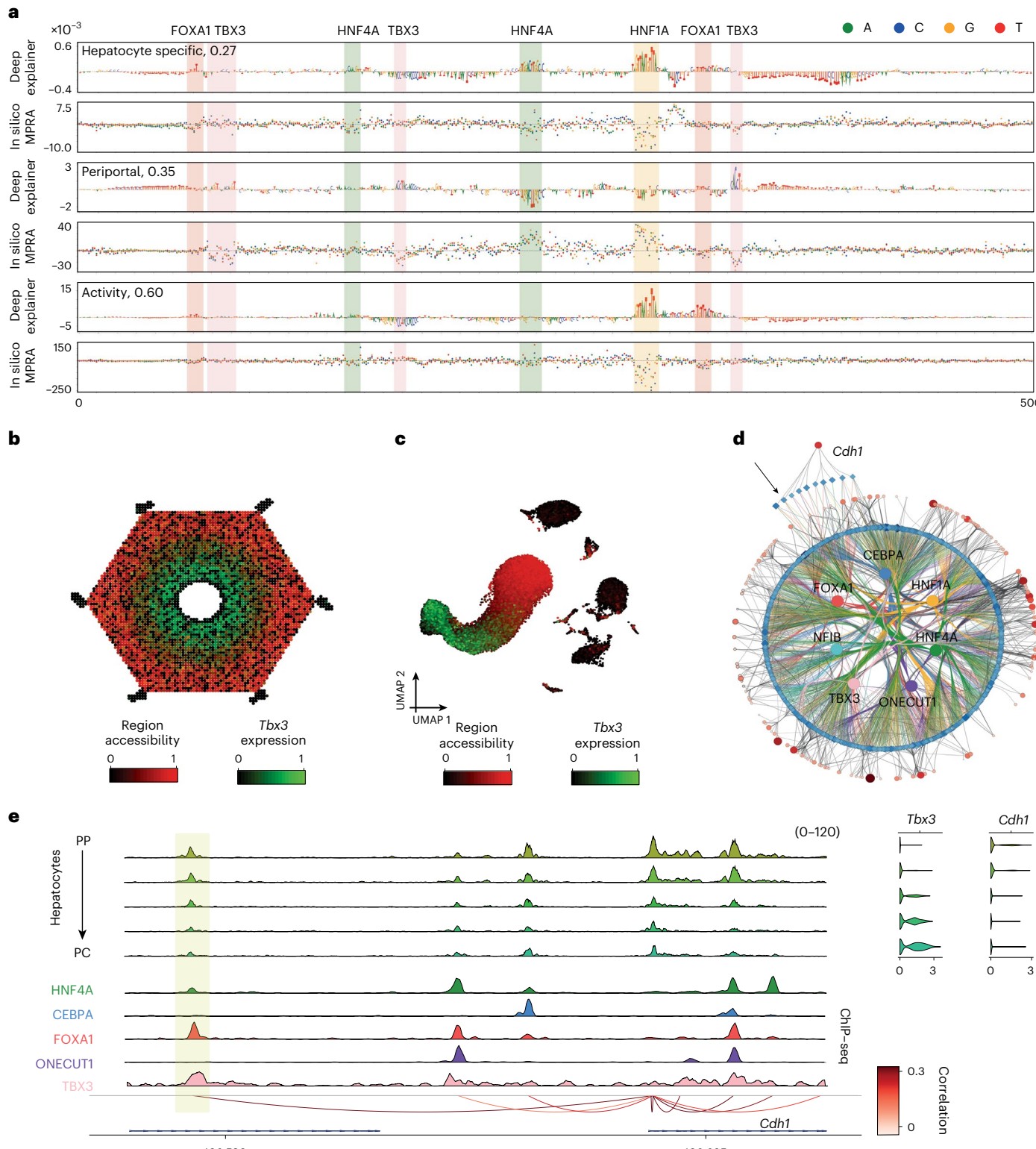

**Fig. 5 | DeepLiver enhancer architecture on a periportal enhancer.**
**a**, DeepExplainer and saturation mutagenesis plots for the accessibility, zonation and activity models on a *Cdh1* periportal enhancer (chromosome 8:106588720–106589220), with motifs highlighted. The accessibility model highlights the nucleotides that make the enhancer accessible in hepatocytes (versus other cell types in the liver); the zonation model highlights the nucleotides that contribute to making the enhancer periportal; and the activity model highlights the nucleotides that have a role in its activity. **b**, ScoMAP liver lobule template (4,498 cells) coloured by region accessibility and TBX3 expression. **c**, Transcriptome-based UMAPs (29,798 cells) coloured by region accessibility

and TBX3 expression. **d**, Periportal core hepatocyte eGRN, with 175 periportal marker genes and 972 regions targeted by the selected core TFs (with CRM score > 3) and conserved across mice. The *Cdh1* enhancer region is indicated by an arrow. **e**, Coverage plot showing pseudobulk accessibility profiles, ChIP–seq coverage (for HNF4A, CEBPA, FOXA1, ONECUT1 and TBX3), SCENIC+ region-to-gene links coloured by correlation score, and *Cdh1* and *Tbx3* expression across the zonated hepatocyte classes (from periportal to pericentral). The *Cdh1* enhancer region is highlighted in yellow. For the transcriptome and epigenome data, cells from five and four biological replicates were combined, respectively. Source numerical data are provided as source data.

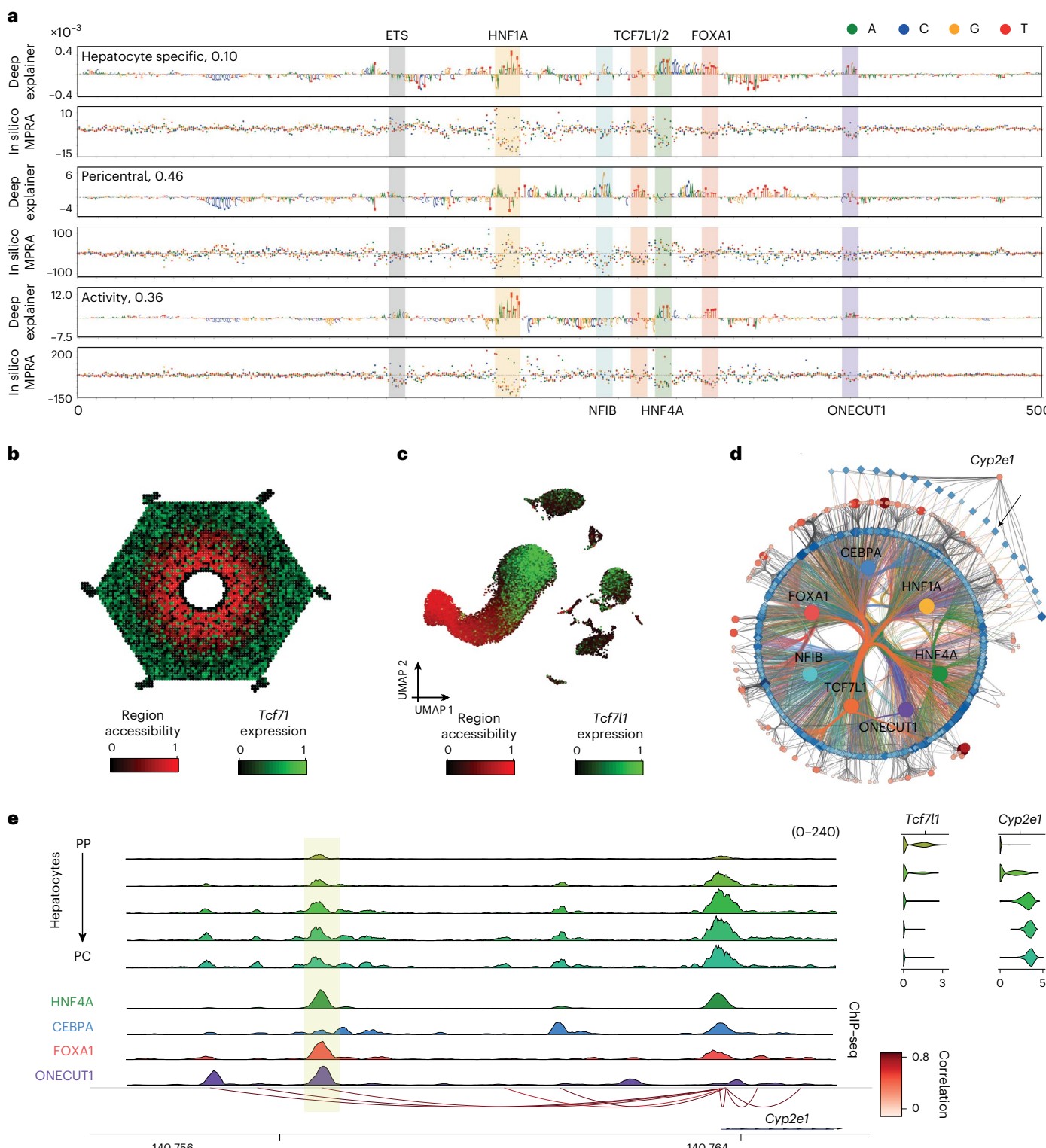

**Fig. 6 | DeepLiver enhancer architecture on a pericentral enhancer.**
**a**, DeepExplainer and saturation mutagenesis plots for the accessibility, zonation and activity models on a *Cyp2e1* pericentral enhancer (chromosome 7:140756424–140756924). The accessibility model highlights the nucleotides that make the enhancer accessible on hepatocytes (versus other cell types in the liver); the zonation model highlights the nucleotides that contribute to making the enhancer pericentral; and the activity model highlights the nucleotides that have a role in its activity. **b**, ScoMAP liver lobule template (4,498 cells) coloured by region accessibility and *Tcf7l1* expression. **c**, Transcriptome-based UMAPs (29,798 cells) coloured by region accessibility and *Tcf7l1* expression.

**d**, Pericentral core hepatocyte eGRN, with 265 pericentral marker genes and 1, 439 regions targeted by the selected core TFs (with CRM score > 3) and conserved across mice. The *Cyp2e1* enhancer region is indicated by an arrow. **e**, Coverage plot showing pseudobulk accessibility profiles, ChIP–seq coverage (for HNF4A, CEBPA, FOXA1 and ONECUT1), SCENIC+ region-to-gene links coloured by correlation score, and *Cyp2e1* and *Tcf7l1* expression across the zonated hepatocyte classes (from periportal to pericentral). The *Cyp2e1* enhancer region is highlighted in yellow. For the transcriptome and epigenome data, cells from five and four biological replicates were combined, respectively. Source numerical data are provided as source data.

effects of mutations with experimental saturation mutagenesis data on six enhancers from earlier studies[41,42] (three each from in vivo and HepG2 cell studies; Fig. 4c,d and Extended Data Fig. 7g,h). DeepLiver predictions of the effect of enhancer mutations correlate with experimental results (with a correlation ranging between 0.36 and 0.75; Extended Data Fig. 7f).

We next used TF-MoDISco patterns and SCENIC+ position weight matrices (PWMs) to identify TF-binding sites among the hepatocyte sequences (Methods). We identified between 1,235 and 6,991 target regions for TBX3, TCF7L1, FOXA1, HNF1A, HNF4A, NFIB, ONECUT1 and CEBPA, with a good overlap with SCENIC+-predicted target regions (17–70%; Extended Data Fig. 8a–c). To validate the predicted binding sites, we compared our predictions with previously published ChIP–seq data for HNF4A, CEBPA, FOXA1 and ONECUT1[29], finding specific signals for the corresponding TFs when centring the regions on the predicted binding sites (Extended Data Fig. 8d). Finally, we assessed the distances between motif instances in overlapping regions. This showed that TCF7L1 and HNF4A often overlap, which is probably due to the similarity between the motifs (GAT**CAAAG** and **CAAAG**TCA, respectively; with the common bases between the motifs highlighted in bold). On the other hand, FOXA1, HNF1A, CEBPA, NFIB and TBX3 are often located close to HNF4A motifs (Extended Data Fig. 8e,f).

We next used DeepLiver to interpret enhancers in the core pericentral and periportal eGRNs from SCENIC+, now at base-pair resolution (Figs. 5 and 6; https://doi.org/10.6084/m9.figshare.24115986). For example, on a *Cdh1* enhancer, DeepLiver finds that FOXA1, HNF4A and HNF1A sites are drivers of enhancer accessibility and activity, whereas TBX3 sites (one dimer motif, and two monomers) are predicted to make the enhancer periportal (Fig. 5a). In agreement, we find HNF4A and FOXA1 ChIP–seq signals in this region, but no CEBPA nor ONECUT1 ChIP–seq signals (Fig. 5e). Both accessibility of this enhancer, and *Cdh1* gene expression, are anticorrelated with TBX3 expression (−0.44 and −0.17, respectively; Fig. 5b–e).

On a pericentral *Cyp2e1* enhancer, DeepLiver identifies HNF1A, FOXA1 and ONECUT1 sites that contribute to enhancer accessibility and activity, and an ETS site that contributes to activity but not accessibility, as observed in other enhancers too (Fig. 6a; https://doi.org/10.6084/m9.figshare.24115986). On the other hand, a NFIB site contributes to accessibility (but not activity), and a TCF7L1/2 site is uniquely found in the zonation model, contributing to make the enhancer pericentral. In agreement, we observed HNF4A, FOXA1 and ONECUT1 ChIP–seq signals in this region (Fig. 6e). TCF7L1 expression is anticorrelated with region accessibility and gene expression (−0.32 and −0.40; Fig. 6b–e). These observations suggest that TBX3 and TCF7L1 may repress these regions. In summary, DeepLiver decodes enhancer accessibility, activity and zonation at the base-pair resolution, and can predict variants that modulate enhancer activity and zonation in hepatocytes.

## Validation of zonated repressor TFs and enhancers

The DeepLiver model provides meaningful interpretations of hepatocyte enhancers and predicts that these enhancers consist of a core

hepatocyte code, mixed with binding sites of the zonated repressor TFs, TBX3 and TCF7L1, which bias enhancer activity to either pericentral or periportal zones, respectively. To test these predictions further, we first performed simulation experiments on the SCENIC+ network, following our previously published perturbation-simulation strategy[7]. Simulation of *Tbx3* or *Tcf7l1* knockdown and overexpression in hepatocytes (Methods) suggests that *Tbx3* overexpression and *Tcf7l1* knockdown can switch periportal hepatocytes to a pericentral state, whereas *Tbx3* knockdown or *Tcf7l1* overexpression can switch pericentral hepatocytes to a periportal state (Fig. 7a–c). The SCENIC+ eGRN predicts that TBX3 and TCF7L1 directly repress each other. Consequently, the knockdown or overexpression of one of the TFs provokes the upregulation or downregulation of the other, respectively (and downregulation and upregulation of the target genes of the other as well; Fig. 7c).

In a second experiment, we introduced specific mutations into a set of hepatocyte enhancers, guided by the DeepLiver model, and then measured their activity using in vivo MPRA. We selected 13 periportal and 21 pericentral enhancers that are predicted to be repressed by TBX3 (pericentrally) and TCF7L1 (periportally), both by SCENIC+ and DeepLiver. We introduced gain-of-function (GOF) and loss-of-function (LOF) mutations affecting HNF4A-, CEBPA-, HNF1A- and FOXA1-binding sites, and mutations of TBX3 and TCF7L1/2 motifs (Fig. 7d and Extended Data Fig. 9a–c), leading to a total of 455 sequences. The activities of these enhancer variants were first tested using bulk MPRA on the mouse liver and human HepG2 cells (Methods, Extended Data Fig. 9d–g and Supplementary Table 3), in which GOF variants of HNF4A, HNF1A, CEBPA and FOXA1 indeed resulted in higher activity, as predicted by DeepLiver (Fig. 7e and Extended Data Fig. 9c,f,g). Variants of the predicted binding sites of the zonation TFs TBX3 and TCF7L1 also showed changes, but these are more difficult to assess from these bulk experiments in which periportal and pericentral hepatocytes are pooled (Fig. 7e).

To solve this problem, we performed MPRA experiments on fluorescence-activated cell sorting (FACS)-sorted hepatocytes, sorted by zone, using pericentral and periportal surface proteins CD73 (encoded by *Nt5e*) and ECAD (encoded by *Cdh1*), respectively, according to a previously published protocol[43] (Fig. 8a, Methods and Extended Data Fig. 10a). The sorted cell fractions indeed represented pericentral and periportal hepatocytes, as shown by bulk ATAC–seq profiles on the separate fractions, which agreed with the snATAC–seq zonated profiles (Fig. 8b). We next analysed enhancer activity on the sorted fractions using MPRA (Methods and Supplementary Table 3). As expected, pericentral enhancers showed higher activity in the pericentral fraction, and vice-versa (Fig. 8c and Extended Data Fig. 10b,c). HNF1A and HNF4A GOF variants resulted in increased activity in both fractions, with milder effects for CEBPA and FOXA1 variants. The destruction of these motifs reduced enhancer activity compared with their wild-type counterparts (Fig. 8c,d and Extended Data Fig. 10d). TBX3 LOF and TCF7L1 GOF resulted in an increase in activity in the pericentral fraction and TCF7L1 LOF resulted in an increase of activity in the periportal population (Fig. 8c,d and Extended Data Fig. 10d).

**Fig. 7 | Validation of zonated repressor TFs through in silico perturbation and MPRA. a**, Simulated cellular shift on the snRNA-seq UMAP (29,798 cells) after *Tbx3* or *Tcf7l1* knockdown (KD) or overexpression (OE), represented by arrows. The arrows are shaded based on the distance travelled by each cell after the simulation. For the UMAP, cells from four snRNA-seq and two single-cell multiome experiments were combined. **b**, Simulated cellular shift on the ScoMAP liver lobule virtual map (VM; 4,498 metacells) after *Tbx3* or *Tcf7l1* knockdown (KD) or overexpression (OE), represented by arrows. The arrows are shaded based on the distance travelled by each cell after the simulation. **c**, The predicted fold change for selected genes (TBX3 targets are shown in purple and TCF7L1 targets are shown in orange) after simulation of *Tbx3* knockdown and *Tcf7l1* overexpression in pericentral hepatocytes and *Tcf7l1* knockdown and *Tbx3* overexpression on periportal hepatocytes. **d**, Overview of DeepLiver-

based sequence mutations (mut) introduced in the wild-type enhancers to shift activity and zonation patterns. These variants cause the appearance of improved motifs (GOF) or their destruction (LOF). On top, a reference TF motif (Ref) from the cisTarget database is shown. The box plots below each variant indicate DeepLiver's predicted shift on activity (active) or zonation (general, pericentral or periportal) scores. In the box plots, the top/lower hinge represents the upper/lower quartile and whiskers extend from the hinge to the largest/smallest value no further than 1.5 × interquartile range from the hinge, respectively. The median is used as the center. DL, Deep Learning. **e**, In vivo MPRA log$_2$[FC] versus the DeepLiver activity score with the highlighted sequence variants for each enhancer. For the MPRA experiments, three and eight biological replicates were performed in HepG2 cells and in vivo, respectively. Source numerical data are provided as source data.

As a third validation experiment, we analysed a public human liver snATAC–seq dataset[44], revealing that the predicted TBX3 and TCF7L1 repressive sites are conserved between species, with similar accessibility patterns along the portocentral axis (Fig. 8e).

Finally, we tested five periportal enhancers and their TBX3 LOF variants using luciferase reporter assays in HepG2 cells and included the pericentral *Cyp2e1* enhancer as a control (Fig. 8f). Human HepG2 cells express *Tbx3* and can therefore be used as a model to test the

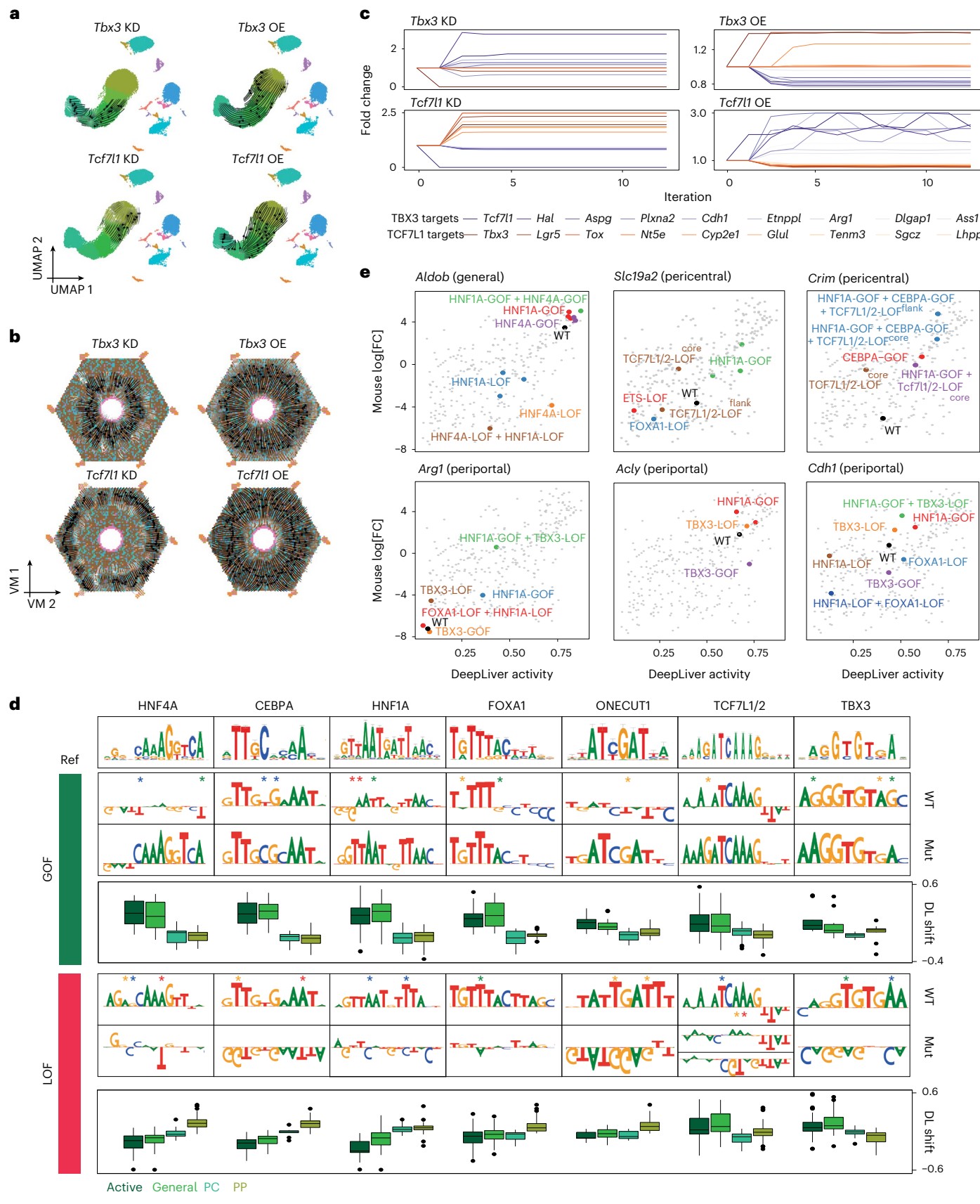

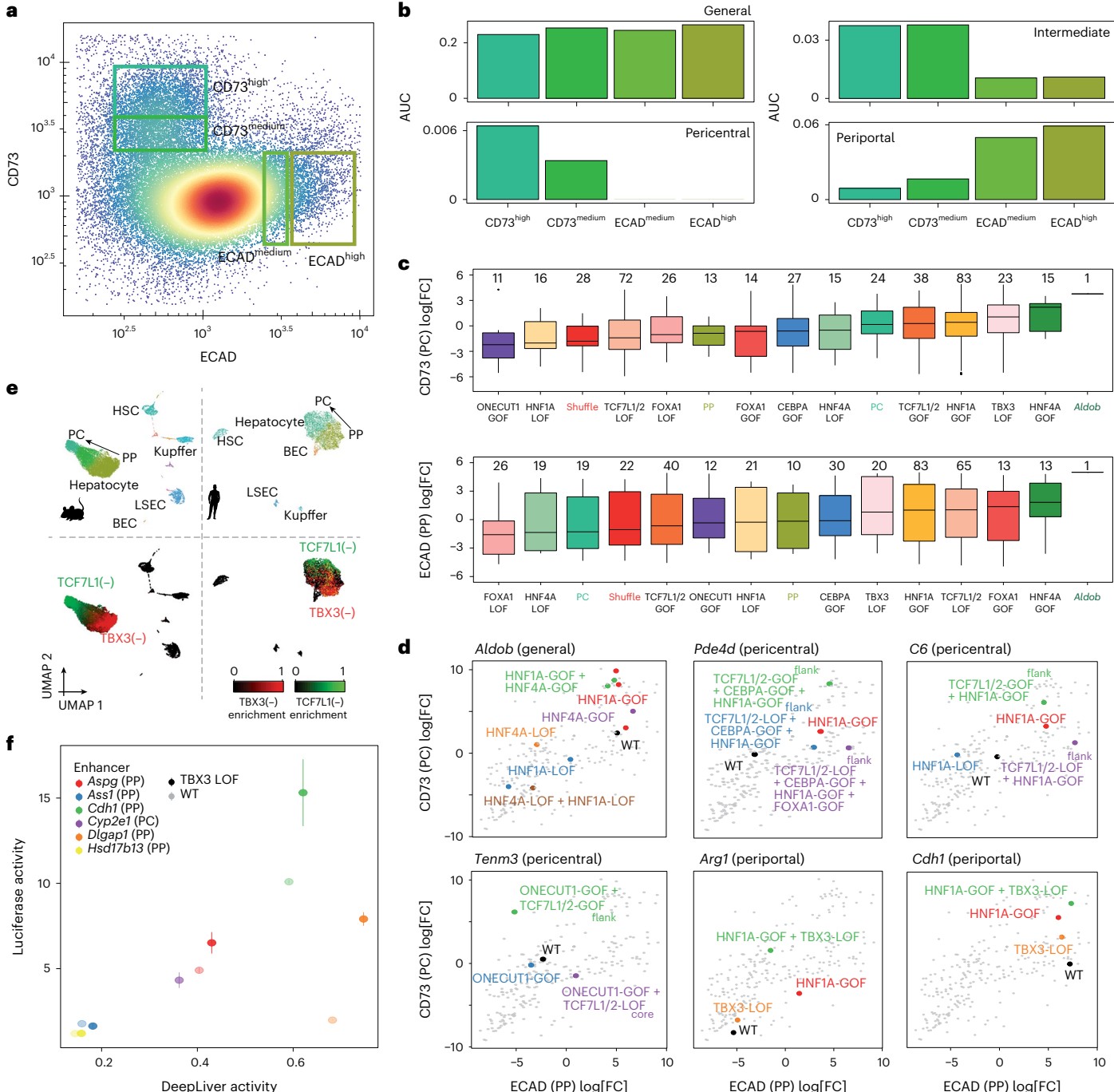

**Fig. 8 | Validation of zonated repressor TFs through FACS–MPRA, cross-species analysis and luciferase assays. a**, FACS analysis of the selected cells according to the intensities of CD73 and ECAD. The rectangles indicate the selected bins along the portocentral axis. **b**, AUCell enrichment of the core general, pericentral, pericentral-intermediate and periportal regions on the sorted populations. **c**, In vivo MPRA $\log_2$[FC] values for the aggregated CD73 and ECAD populations per variant class. The number of enhancers in each group is indicated at the top. The activity of the *Aldob* enhancer is indicated by a line. The centre line shows the median value, the top and bottom hinges represent the upper and lower quartiles, and the whiskers extend from the hinge to the largest and smallest value no further than 1.5 × interquartile range from the hinge, respectively. Control and wild-type sequences are highlighted. Four biological replicates were used. **d**, CD73 MPRA $\log_2$[FC] versus ECAD MPRA $\log_2$[FC] values with the highlighted sequence variants for each enhancer. Four biological replicates were used. **e**, Mouse and human liver snATAC–seq UMAP (22,600 and 6,366 cells, respectively) coloured by cell type (top) and eRegulon enrichment (bottom). For the mouse UMAP, cells from four biological replicates were combined. **f**, Luciferase activity in HepG2 cells versus DeepLiver activity scores for selected enhancers and their variants. $n$ = 4 biologically independent luciferase experiments per enhancer. Data are mean ± s.e.m. WT, wild type. Source numerical data are provided as source data.

effect of mutating TBX3 sites in hepatocyte enhancer sequences[45]. The TBX3-binding sites predicted by SCENIC+ were less active in HepG2 cells compared with in vivo (Fig. 3e), and TBX3 LOF variants showed increased activity in HepG2 cells as determined using MPRA (Extended Data Fig. 9f,g). As predicted by DeepLiver, TBX3 LOF in inactive enhancers did not rescue the enhancers (*Hsd17b13* and *Ass1*). However, the predicted active enhancers (*Aspg*, *Cdh1* and *Dlgap1*) exhibited increased activity when the TBX3-binding site was

mutated. This indicates that these enhancers are directly repressed by TBX3 through these sites. In summary, our results suggest that the grammar of hepatocyte enhancers that encodes their zonation pattern includes TBX3- and TCF7L1/2-binding sites, while HNF1A and HNF4A are the most relevant binding sites regarding activity.

## Discussion

Single-cell omics methods have revolutionized the definition of cell types, as they enable the profiling of up to thousands of snapshots of cell states in a tissue. Cell types can be defined as a continuum of (reversible) cell states that are often binarized based on statistical clustering of their transcriptome or epigenome. Yet, the discretization of dynamic populations is not a trivial task and is strongly affected by parameter selection. An alternative approach to characterize cell states is to study its underlying GRNs, and all of the regulatory variations on that central theme[2,7,46]. Using the mouse liver as a model system, we aimed to depict the core identity, as well as the various cell states, of hepatocytes, alongside their gene regulatory programs.

We used two complementary computational strategies to address this problem. First, SCENIC+[7] identified a core hepatocyte GRN controlled by HNF4A, HNF1A, CEBPA, FOXA1, NFIB and ONECUT1, many of which have been extensively studied in liver development and differentiation[47–50]. As a subset of this program, we could disentangle mechanisms underlying hepatocyte zonation, controlled by the repressor TFs TCF7L1 and TBX3 (Supplementary Fig. 9). TCF7L1 and TBX3 are indeed well-known repressors in development[51,52] and, while it has been previously reported that *Tcf7l1* and *Tbx3* expression is zonated in the adult mouse liver[43,53], here we show their direct implication in liver zonation regulation by enhancer-GRN mapping. Importantly, although we exclusively used male mice, our analyses show that these core regulatory networks are not affected by sex.

SCENIC+ could identify these candidate repressors because their motif is significantly enriched in regulatory regions that are accessible in hepatocytes in which the TF is not expressed, while they are inaccessible in hepatocytes in which the TF is expressed. As a potential mechanism, how repressor binding could result in the absence of an ATAC peak, TF footprinting suggests that direct repressor binding may occur within nucleosome-occupied regions, while activator binding is strongly associated with nucleosome depletion[54]. Accordingly, TF ChIP–seq showed that TBX3 binds to periportal regions that have low accessibility or are not accessible in pericentral hepatocytes, where *Tbx3* is expressed. This illustrates the power of single-cell multi-omic profiling, whereby both positive and negative correlations between accessibility and gene expression can be exploited to infer cell-type-specific regulatory interactions without the need of high-quality antibodies, large amounts of input material or low-throughput perturbation experiments[7]. However, a key limitation is that single-cell data are sparse, which reduces the sensitivity to detect negative correlations, and can lead to false-negative predictions.

In a second complementary strategy, we trained CNN to predict, based on the enhancer sequence as input, its ATAC topic membership or, in other words, in which cell type/state the enhancer is accessible. CNN-based enhancer modelling has recently gained traction, due to the ability of these models to interpret enhancer grammar[55–58]. A key limitation of CNN models is that they require large input datasets for training. Although training on small datasets may lead to overfitting, transfer learning from sequence models trained with large datasets has recently been shown to be a robust alternative[59]. Here we propose several transfer-learning applications, whereby the first (topic-based) model is fine-tuned either to learn cell state (in our case, hepatocyte zonation) or enhancer activity (based on MPRA data). The topic-CNN could recapitulate the core hepatocyte code, with sequence features associated with the same TFs as identified by SCENIC+. The zonation-CNN added TBX3 and TCF7L1 motifs as crucial sequence features to the hepatocyte enhancers, whereas the activity-CNN added ETS and AP-1 sites underlying higher enhancer activity, in agreement with previous MPRA studies in the liver[60]. Importantly, TBX3- and TCF7L1-binding sites are located predominantly within hepatocyte enhancers, in close proximity to binding sites of the hepatocyte core TFs. This shows that, rather than repressing genes through distinct regulatory regions, these repressor sites form an integral, and probably evolutionary selected, part of the state-specific hepatocyte enhancer logic. A remaining question is whether and how mechanistically repressors interact with other TFs that target the same enhancer. For example, proximity ligation assays have been used to identify TF co-factors[61]; however, such approaches may yield negative results in this setting as it is unclear whether the repressors and activators interact with each other or rather compete. New approaches, such as scCUT&Tag[62], may provide opportunities to explore TF binding across cell states in a complex tissue.

In conclusion, we unravelled the regulatory grammar underlying hepatocyte identity. We provide an extensive resource of the adult mouse liver, including a spatial and single-cell multi-omics atlas, eGRNs and enhancer activity, that can be explored in Scope (http://scope.aertslab.org/#/Bravo_et_al_Liver) and the UCSC genome browser (https://genome.ucsc.edu/s/cbravo/Bravo_et_al_Liver). We envision that our workflow can be used as a roadmap to study other biological systems that will further improve our understanding of how cell types and their functional states are encoded in the genome.

## Online content

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

## Methods

### Mice

All of the animal experiments were conducted according to the KU Leuven ethical guidelines and approved by the KU Leuven Ethical Committee for Animal Experimentation (ECD P007/2021). Adult male mice (8 to 10 weeks old) were used in this study. All mice were C57BL/6JaxCrl except for mouse 1 in the single-cell experiments, which was Crl:CD-1. Mice were maintained under standard housing conditions, with continuous access to food and water, except for mice 4 and 5 in the single-cell experiments, for which food was removed approximately 10 h before the experiments.

### Single-cell data generation

**Mouse liver dissection.** Animals were sacrificed by $CO_2$ and the liver was collected for further experiments. For the fresh nucleus isolation, samples were immediately processed. For the frozen nucleus isolation, samples were immediately snap-frozen in liquid nitrogen and stored at −80 °C until processing. In total, 4, 2 and 3 mice were used for snRNA-seq, snATAC–seq and single-cell multiomics experiments, respectively.

**Sample and library preparation for 10x snRNA-seq. Nuclei isolation.** The liver nuclei were isolated following the protocol described previously[63]. For the fresh samples, 200 mg fresh big lobe piece of mouse liver tissue was minced and transferred to a Dounce homogenizer cylinder containing 1 ml of ice-cold homogenization buffer (320 mM sucrose, 5 mM $CaCl_2$, 3 mM magnesium acetate, 10 mM Tris-HCl (pH 7.5), 0.1 mM EDTA, 0.1% IGEPAL CA-63, 0.1 mM PMSF, 1 mM βME and 0.2 U μl$^{-1}$ RNasin Plus RNase Inhibitor (Promega). For the frozen samples, a piece of 200 mg liver big lobe was sectioned on dry ice and transferred to a Dounce homogenizer cylinder containing 1 ml of ice-cold homogenization buffer and let to thaw for 5 min. From this step onward, both the fresh and frozen tissue were homogenized with ten strokes of pestle A and ten strokes of pestle B until a homogeneous nucleus suspension was achieved. The resulting homogenate was filtered through a 70 μm cell strainer (Corning). Furthermore, 1.65 ml of homogenization buffer was topped up and mixed with 2.65 ml of gradient medium (5 mM $CaCl_2$, 50% Optiprep (Stemcell Technologies), 3 mM magnesium acetate, 10 mM Tris-HCl (pH 7.5), 0.1 mM PMSF, 1 mM βME). A total of 4 ml of 29% iodoxanol cushion was prepared with Opti-Prep (Stemcell Technologies) and diluent medium (250 mM sucrose, 150 mM KCl, 30 mM $MgCl_2$, 60 mM Tris-HCl (pH 7.5)), and added into an ultracentrifuge tube (Beckman Coulter). Next, 5.3 ml of sample in homogenization buffer and gradient medium was gently layered on top of the 29% iodoxanol cushion. The samples were centrifuged in the SW41Ti rotor (Beckman Coulter) at 7,700$g$ and 4 °C for 30 min and the obtained supernatant was gently removed without disturbing the nucleus pellet. Nuclei were resuspended in 200 μl resuspension buffer (1× PBS, 1% BSA and 0.2 U μl$^{-1}$ RNasin Plus RNase Inhibitor (Promega)) and transferred to a 1.5 ml Eppendorf tube. A total of 9 μl of sample was mixed with 1 μl of arginine orange/propidium iodide (AO/PI) stain, loaded onto a LUNA-FL slide and visualized using the LUNA-FL automated cell counter for nucleus yield, morphology and presence of clumps/debris.

**Library preparation.** Single-nucleus libraries were generated using the 10x Chromium Single-Cell Instrument and Chromium Single Cell 3′ Reagent v3 Kits (10x Genomics) according to the manufacturer's protocol. In brief, the nucleus suspension was loaded into the Chromium chip for partitioning into nanolitre-scale gel beads-in-emulsion (GEMs). After GEM generation, the obtained emulsion was incubated in the C1000 Touch Thermal Cycler (Bio-Rad) under the following program: 53 °C for 45 min, 85 °C for 5 min and hold at 4 °C. Incubation of the GEMs produced barcoded, full-length cDNA from poly-adenylated mRNA. After incubation, single-cell droplets were dissolved, and full-length

cDNA was isolated using Cleanup Mix containing Silane Dynabeads. To generate sufficient mass for library construction, the cDNA was amplified by PCR as follows: 98 °C for 3 min; 12 cycles of 98 °C for 15 s, 63 °C for 20 s and 72 °C for 1 min; 72 °C for 1 min; and hold at 4 °C. Subsequently, the amplified cDNA was fragmented, end-repaired, A-tailed and index-adapter-ligated, with SPRIselect cleanup between steps. The final gene expression library was amplified by PCR as follows: 98 °C for 45 s; 10–12 cycles of 98 °C for 20 s, 54 °C for 30 s and 72 °C for 20 s; 72 °C for 1 min; and hold at 4 °C. The sequencing-ready libraries were cleaned-up using SPRIselect beads (Beckman Coulter).

**Sequencing.** The final libraries were quantified using the Qubit dsDNA HS Assay Kit (Life Technologies). The fragment size of every library was analysed using the Bioanalyzer high-sensitivity chip and were sequenced on HiSeq 4000 or NovaSeq 6000 instruments with the following sequencing parameters: 28 bp read 1, 8 bp index 1 (i7), 0 bp index 2 (i5), 91 bp read 2.

**snRNA-seq read mapping.** The generated fastq files were processed using the Cell Ranger (v.1.0.0) count function. Reads were aligned to a pre-mRNA *Mus musculus* reference genome (mm10) that listed each gene transcript locus as an exon, and included intronic reads in the counting (10x Genomics; https://support.10xgenomics.com/single-cell-gene-expression/software/pipelines/latest/advanced/references#premrna).

**Sample and library preparation for 10x snATAC–seq. Nuclei isolation.** The liver nuclei were isolated using a modified protocol from the Nuclei Isolation for Single Cell ATAC Sequencing (CG000169) Demonstrated Protocol from 10x Genomics. In brief, 200 mg fresh big lobe piece of mouse liver tissue was minced and transferred into a Dounce homogenizer cylinder containing 1 ml of ice-cold homogenization buffer (10 mM NaCl, 10 mM Tris-HCl (pH 7.4), 3 mM $MgCl_2$, 0.1% Tween-20, 0.1% IGEPAL CA-63, 0.01% Digitonin, 1% BSA) and incubated for 5 min on ice. Next, the tissue was homogenized with 15 strokes of pestle A and 15 strokes of pestle B until a homogeneous nucleus suspension was achieved. The resulting homogenate was filtered through a 70 μm cell strainer (Corning). The tissue material was centrifuged at 500$g$ for 5 min at 4 °C and the supernatant was discarded. The tissue pellet was resuspended in 1 ml wash buffer (20 mM NaCl, 20 mM Tris-HCl (pH 7.4), 6 mM $MgCl_2$, 1% BSA). The wash step was repeated one more time and the resulting final pellet was resuspended in 100 μl diluted nucleus buffer (10x Genomics snATAC-seq kit). A total of 9 μl of sample was mixed with 1 μl of AO/PI stain, loaded onto a LUNA-FL slide and visualized with the LUNA-FL Automated cell counter for nucleus yield, morphology and the presence of clumps/debris.

**Library preparation.** Single-nucleus libraries were generated using the 10x Chromium Single-Cell Instrument and Single Cell ATAC v1 kit (10x Genomics) according to the manufacturer's protocol. In brief, the single mouse liver nuclei were incubated for 60 min at 37 °C with a transposase that fragments the DNA in open regions of the chromatin and adds adapter sequences to the ends of the DNA fragments. After generation of nanolitre-scale GEMs, GEMs were incubated in a C1000 Touch Thermal Cycler (Bio-Rad) under the following program: 72 °C for 5 min; 98 °C for 3 s; 12 cycles of 98 °C for 10 s, 59 °C for 30 s and 72 °C for 1 min; 72 °C for 1 min; and hold at 4 °C. Incubation of the GEMs produced 10x barcoded DNA from the transposed DNA. Next, single-cell droplets were dissolved, and the transposed DNA was isolated using Cleanup Mix containing Silane Dynabeads. Illumina P7 sequence and a sample index were added to the single-strand DNA during ATAC library construction by PCR: 98 °C for 45 s; 9 cycles of 98 °C for 20 s, 67 °C for 30 s and 72 °C for 20 s; 72 °C for 1 min; and hold at 4 °C. The sequencing-ready ATAC library was cleaned-up with SPRIselect beads (Beckman Coulter).

**Sequencing.** Final libraries were quantified using the Qubit dsDNA HS Assay Kit (Life Technologies). The fragment size of every library was analysed using the Bioanalyzer high-sensitivity chip and the libraries were sequenced on NextSeq 500 instruments (Illumina) with the following sequencing parameters: 70 bp read 1, 8 bp index 1 (i7), 16 bp index 2 (i5), 70 bp read 2.

**snATAC-seq read mapping.** The generated fastq files were processed using the cellranger-atac (v.1.2.0) count function. Reads were aligned to the *M. musculus* reference genome (refdata-cellranger-atac-mm10-1.2.0).

**Sample and library preparation for 10x single-cell multiome ATAC and gene expression. Sample preparation.** For 'Multiome-10x_Fresh_Mouse-4' we used a modified protocol from the Nuclei Isolation from Complex Tissues for Single Cell Multiome ATAC + Gene Expression Sequencing Protocol (CG000375) from 10x Genomics. In brief, 100 mg fresh big lobe piece of mouse liver tissue was minced and transferred to a Dounce homogenizer cylinder containing 1 ml of ice-cold homogenization buffer (10 mM NaCl, 10 mM Tris-HCl (pH 7.4), 3 mM MgCl$_2$, 0.1% IGEPAL CA-63, 1 mM DTT, 1 U µl$^{-1}$ of Protector RNase Inhibitor (Sigma-Aldrich)). The tissue was homogenized with five strokes of pestle A and ten strokes of pestle B until a homogeneous nucleus suspension was achieved. The resulting homogenate was filtered through a 70 µm cell strainer (Corning). The tissue material was centrifuged at 500$g$ for 5 min at 4 °C and the supernatant was discarded. The tissue pellet was resuspended in wash buffer (1% BSA in PBS + 1 U µl$^{-1}$ of Protector RNase Inhibitor (Sigma-Aldrich)). Nuclei were stained with 7AAD (Thermo Fisher Scientific) and viability sorted on the BD FACS Fusion (BD Biosciences) system into a 5 ml low-bind Eppendorf tube containing BSA with RNase inhibitor. The sorted nuclei were centrifuged at 500$g$ for 5 min at 4 °C and the supernatant was discarded. Next, the nuclei were permeabilized by resuspending the pellet in 0.1× lysis buffer (10 mM NaCl, 10 mM Tris-HCl (pH 7.4), 3 mM MgCl$_2$, 0.1% IGEPAL CA-63, 0.01% Digitonin, 1% BSA, 1 mM DTT, 1 U µl$^{-1}$ of Protector RNase Inhibitor, Sigma-Aldrich) and incubated on ice for 2 min. A total of 1 ml wash buffer (10 mM NaCl, 10 mM Tris-HCl (pH 7.4), 3 mM MgCl$_2$, 0.1% Tween-20, 1% BSA, 1 mM DTT, 1 U µl$^{-1}$ of Protector RNase Inhibitor (Sigma-Aldrich)) was added. The nuclei were centrifuged at 500$g$ for 5 min at 4 °C and the supernatant was discarded. The nucleus pellet was resuspended in diluted nucleus buffer (1× Nuclei Buffer Multiome kit (10x Genomics)), 1 mM DTT, 1 U µl$^{-1}$ of Protector RNase Inhibitor, Sigma-Aldrich). For the 'Multiome-NST_Fresh_Mouse-5' sample nuclei isolation we used a modified protocol from a previous study[64]. In brief, 100 mg fresh big lobe piece of mouse liver tissue was chopped and transferred to a Dounce homogenizer cylinder containing 1 ml of ice-cold homogenization buffer (salt-Tris solution: 146 mM NaCl, 10 mM Tris (pH 7.5), 1 mM CaCl$_2$, 21 mM MgCl$_2$, 0.2% IGEPAL CA-63, 0.01% BSA, 0.2 U µl$^{-1}$ of RNasin Plus RNase Inhibitor (Promega)). The tissue was homogenized with five strokes of pestle A and ten strokes of pestle B until a homogeneous nucleus suspension was achieved. The resulting homogenate was filtered through a 70-µm cell strainer (Corning). The homogenizer and the filter were rinsed with an additional 1 ml homogenization buffer and 3 ml salt-Tris solution buffer (146 mM NaCl, 10 mM Tris (pH 7.5), 1 mM CaCl$_2$, 21 mM MgCl$_2$). The tissue material was centrifuged at 500$g$ for 5 min at 4 °C. The obtained pellet, after supernatant removal, was resuspended in 1.5 ml salt-Tris solution buffer supplemented with 0.2 U µl$^{-1}$ RNasin Plus RNase Inhibitor (Promega). The tissue material was centrifuged at 500$g$ for 5 min at 4 °C. The obtained pellet, after supernatant removal, was resuspended in 1.5 ml wash buffer (1× PBS, 1% BSA and 0.2 U µl$^{-1}$ RNasin Plus RNase Inhibitor (Promega)). The wash step was repeated one more time. The final pellet was resuspended in 500 µl wash buffer, filtered, stained with DAPI (Thermo Fisher Scientific) and viability sorted on the BD FACS Fusion (BD Biosciences) system into 5 ml low-bind Eppendorf

tubes containing BSA with RNase inhibitor. The sorted nuclei were centrifuged at 500$g$ for 5 min at 4 °C and the supernatant was discarded. Nuclei were resuspended in 50 µl of resuspension buffer. The nuclei pellet was resuspended in diluted nucleus buffer (1× Nuclei Buffer Multiome kit, 10x Genomics), 1 mM DTT, 1 U µl$^{-1}$ RNasin Plus RNase Inhibitor (Promega)). A total of 9 µl of sample was mixed with 1 µl of AO/PI stain, loaded onto a LUNA-FL slide and visualized using the LUNA-FL Automated cell counter for nucleus yield, morphology and the presence of clumps/debris.

**Library preparation.** Single-nucleus libraries were generated using the 10x Chromium Single-Cell Instrument and NextGEM Single Cell Multiome ATAC + Gene Expression kit (10x Genomics) according to the manufacturer's protocol. In brief, the nuclei were incubated for 60 min at 37 °C with a transposase that fragments the DNA in open regions of the chromatin and adds adapter sequences to the ends of the DNA fragments. After generating nanolitre-scale GEMs, GEMs were incubated in a C1000 Touch Thermal Cycler (Bio-Rad) under the following program: 37 °C for 45 min; 25 °C for 30 min; and hold at 4 °C. Incubation of the GEMs produced 10x barcoded DNA from the transposed DNA (for ATAC) and 10x barcoded, full-length cDNA from poly-adenylated mRNA (for GEX). Next, quenching reagent (Multiome 10x kit) was used to stop the reaction. After quenching, single-cell droplets were dissolved and the transposed DNA and full-length cDNA were isolated using the clean-up mix containing silane Dynabeads. To fill gaps and generate sufficient mass for library construction, the transposed DNA and cDNA were amplified by PCR: 72 °C for 5 min; 98 °C for 3 min; 7 cycles of 98 °C for 20 s, 63 °C for 30 s and 72 °C for 1 min; 72 °C for 1 min; and hold at 4 °C. The pre-amplified product was used as input for both ATAC library construction and cDNA amplification for gene expression library construction. Illumina P7 sequence and a sample index were added to the single-strand DNA during ATAC library construction by PCR: 98 °C for 45 s; 7–9 cycles of 98 °C for 20 s, 67 °C for 30 s and 72 °C for 20 s; 72 °C for 1 min; and hold at 4 °C. The sequencing-ready ATAC library was cleaned up with SPRIselect beads (Beckman Coulter). Barcoded, full-length pre-amplified cDNA was further amplified by PCR: 98 °C for 3 min; 6–9 cycles of 98 °C for 15 s, 63 °C for 20 s and 72 °C for 1 min; 72 °C for 1 min; and hold at 4 °C. Subsequently, the amplified cDNA was fragmented, end-repaired, A-tailed and index-adapter-ligated, with SPRIselect bead (Beckman Coulter) clean-up between steps. The final gene expression library was amplified by PCR: 98 °C for 45 s; 5–16 cycles of 98 °C for 20 s, 54 °C for 30 s and 72 °C for 20 s; 72 °C for 1 min; and hold at 4 °C. The sequencing-ready GEX library was cleaned up using SPRIselect beads (Beckman Coulter).

**Sequencing.** Final libraries were quantified using the Qubit dsDNA HS Assay Kit (Life Technologies). The fragment size of every library was analysed using the Bioanalyzer high-sensitivity chip. All 10x Multiome ATAC libraries were sequenced on the NovaSeq 6000 instruments (Illumina) with the following sequencing parameters: 50 bp read 1, 8 bp index 1 (i7), 16 bp index 2 (i5), 49 bp read 2. All 10x Multiome gene expression libraries were sequenced on the NovaSeq 6000 instruments with the following sequencing parameters: 28 bp read 1, 10 bp index 1 (i7), 10 bp index 2 (i5), 75 bp read 2.

**Multiome (snATAC-seq and scRNA-seq) read mapping.** The generated fastq files were processed with cellranger-arc (v.2.0.0) count function, with the include introns =True option. Reads were aligned to the *M. musculus* reference genome (ata-cellranger-arc-mm10-2020-A-2.0.0).

### Single-cell data analysis
**Transcriptome analysis.** 10x snRNA-seq and 10x multiome (gene expression) runs were analysed first independently using VSN-pipelines (v.0.27.0)[65]. In brief, cells with at least 350 genes expressed and a percentage of mitochondrial reads below 10% were retained. Scanpy

(v.1.8.2)[66] was run with the default parameters, using the number of principal components automatically selected by VSN-Pipelines and using Leiden clustering with resolutions 0.4, 0.6 and 0.8. Hepatocyte clusters with low gene expression and a high percentage of mitochondrial reads were removed, as well as doublets called with Scrublet (v.0.2.3)[67]. The samples were merged, obtaining 29,798 high-quality cells, and reanalysed using VSN-Pipelines. To correct for batch effects, we used Harmony on the selected principal components (34), using Leiden clustering with resolution 0.6, resulting 15 clusters. The VEC and DC subpopulations were identified according to marker genes. This resulted in the identification of 14 cell types.

**Epigenome analysis.** 10× snATAC–seq samples were processed with cisTopic (v.0.3.0)[68], using the cells called by Cell Ranger (v.1.2.0, 5,628 cells) and mm10 SCREEN regions (1,212,823 regions). For topic modelling, we used cisTopic's WarpLDA[69] with the default parameters, using 500 iterations and inferring models with 2, 5, 10 to 30 (by a step of 1), 35, 40, 45 and 50. This resulted in a model with 19 topics. After correcting sample effects with Harmony[70] (v.1.0, applied on the scaled topic distributions), we performed Leiden clustering with resolution 0.6, obtaining 11 clusters. Gene activity was calculated by aggregating the probabilities of regions ±10 kb from the TSS (including the gene body). Cluster annotation was performed based on motif enrichment, gene activity and label transfer from the annotated transcriptome with Seurat[71] (v.4.0.3, using cisTopic's gene activity matrix, cca as reduction and the first 10 dimensions). The labelled 10x snATAC–seq and multiome cells (annotated based on the transcriptome labels) and the snATAC–seq fragments were used as input for pycisTopic (v.1.0.1.dev75 + g3d3b721)[7]. In brief, we first created pseuobulks per cell type and performed peak calling using MACS2[72] (v.2.2.7.1, with --format BEDPE --keep-dup all --shift 73 --ext_size 146 as parameters, as recommended for single-cell ATAC–seq data). To derive a set of consensus peaks, we used the iterative overlap peak merging procedure described previously[73], as implemented in pycisTopic. First, each summit is extended a 'peak_half_width' (by default, 250 bp) in each direction and then we iteratively filtered out less significant peaks that overlap with a more significant one. During this procedure, peaks are merged and, depending on the number of peaks included into them, different processes will happen: (1) 1 peak: the original peak will be retained; (2) 2 peaks: the original peak region with the highest score will be retained; and (3) 3 or more peaks: the original region with the most significant score will be taken, and all of the original peak regions in this merged peak region that overlap with the significant peak region will be removed. The process is repeated with the next most significant peak (if it was not removed already) until all of the peaks are processed. This procedure will happen twice, first in each pseudobulk peak, and after peak score normalization to process all of the peaks together. This resulted in 486,888 regions. We further filtered the dataset on the basis of the snATAC–seq quality as well, retaining cells with at least 1,000 fragments, FRiP > 0.4 and TSS enrichment > 7, resulting in 22,600 high-quality cells. Topic modelling was performed using Mallet (v.2.0), using 500 iterations and models with 2 topics and from 5 to 100 by an increase of 5. Additional models between 75 and 85 (by an increase of 1) were added, as we observed that the best model should be on that area based on the model selection metrics, and we selected a model with 82 topics. Batch effects between samples were corrected using harmonypy[70] (v.0.0.6) on the scaled topic distributions, and Leiden clustering with a resolution of 0.6 resulted in 11 clusters, corresponding to 14 cell types based on previous labelling. Drop-out imputation was performed by multiplying the region-topic and topic-cell probabilities. The imputed accessibility matrix was multiplied by 10[6]. DARs were calculated between all cell populations and specifically within hepatocytes, HSC and LSEC subgroups using the default parameters and topics were binarized using Otsu thresholding[74]. Hepatocyte DARs and shared hepatocyte topics were curated by

performing hierarchical clustering on the pseudobulk probabilities, removing a small fraction of lowly accessible and generally accessible regions, and defining non-overlapping groups between the different gradient groups. Gene Ontology analysis was performed using GREAT (v.4)[75]. We also ran MACS2 (v.2.2.7.1) bdgdiff between hepatocytes, LSECs and HSCs zonated states using the default parameters. The number of shared regions across mice was calculated as the regions in the shared curated topics. To identify enriched motifs and infer TF cistromes (that is, sets of regions in which a TF motif is present), pycisTarget (v.1.0.1.dev42+gb6707ee) was run using a custom database with the consensus regions on DARs, binarized topics (with Otsu thresholding), curated DARs and topics and MACS2 (v.2.2.7.1) bdgdiff regions, with and without promoters, and using pycisTarget and DEM[7].

**Multiome analysis.** The gene expression matrix, the imputed accessibility from pycisTopic and the TF cistromes previously identified by motif enrichment analysis on DARs and topics with pycisTarget were used as input for SCENIC+ (v 0.1.dev411+gf4bcae5.d20220810)[7], using only the multiome cells for eGRN inference. SCENIC+ was run with the default parameters on the complete dataset and using only hepatocytes, using http://nov2020.archive.ensembl.org/ as Biomart host. In brief, a search space of a maximum between either the boundary of the closest gene or 150 kb and a minimum of 1 kb upstream of the TSS or downstream of the end of the gene was considered for calculating region-to-gene relationships using gradient boosting machine regression. TF-to-gene relationships were calculated using gradient boosting machine regression between all TFs and all genes. Final eRegulons were constructed using the GSEA approach in which region-to-gene relationships were binarized based on gradient boosting machine regression importance scores using the 85th, 90th and 95th quantile; the top 5, 10 and 15 regions per gene and using the BASC method for binarization[76]. eRegulons between the two runs (with all cells and only hepatocytes) were merged. Gene-based and region-based eRegulons were scored in the relevant datasets (multiome, all snRNA-seq and snATAC–seq and spatial templates) using AUCell (v.1.22.0)[77]. eRegulons with positive region-to-gene relationships, at least 20 target genes and a correlation between gene-based and region-based AUC scores above 0.4 were retained, obtaining 180 high-quality eRegulons.

**Hi-C and ChIP–seq data analysis.** To validate these eRegulons, we used publicly available Hi-C and ChIP–seq data[28,29]. In brief, the Hi-C data were processed using Juicer (v.1.9.9), extracting values using KR for normalization by 5 kb windows, and retaining only links with a score of >10 and involving a bin that overlaps at least one of the consensus peaks and a TSS (±1,000 bp), resulting in 890,488 region–gene links. For the ChIP–seq data processing, reads were mapped to the mm10 genome using Bowtie2 (v.2.3.5.1)[78], peaks were called using MACS2 (v.2.2.7.1, with --format BAM --gsize mm --qvalue 0.05 --nomodel --keep-dup all --call-summits --nolambda as options) and bigwig files were generated using deepTools[79] bamCoverage function (v.3.5.0, with --normalizeUsing CPM --binSize 1 as parameters). Coverage on the eRegulon regions was obtained using deepTools computeMatrix.

**Downstream analyses.** Pseudotime order was calculated using the DPT() function of destiny (v.3.2.0)[80] per cell type using as input the harmony corrected PCs and topics from the snRNA-seq and snATAC–seq analyses, respectively. To assess the number of regions and regions affected by zonation, we took the shared regions in hepatocytes, LSECs and HSCs (based on topics) and marker genes, respectively, and a GAM was fitted on the basis of their accessibility and expression over pseudotime (representing zonation). After filtering for genes fitted with adjusted $P$ < 0.01. We identified 275, 220 and 2,697 genes and 281, 475 and 8,805 regions that vary along the portocentral axis in HSCs, LSECs and hepatocytes, respectively. To rank eRegulons (or signatures) based on how affected they are by zonation and/or sample, we performed

ANOVA over the AUC values along the pseudotime per sample, and calculated Bonferroni adjusted $P$ values. We performed PCA dimensionality reduction on the AUC eRegulon matrix (with regulons as rows, hepatocyte cells as columns and the AUC per eRegulon and cell as values), using prcomp from the stats R package (v.3.6.2) with center=TRUE and scale=TRUE. We found that the first and second principal components largely explained the variance due to zonation and sample biases, respectively, based on the distribution of the ANOVA $P$ value over the PCs. ANOVA was also used to identify pathways affected by zonation and/or sample, derived from a previous study[6], using the AUC values after scoring the signatures with AUCell (v.0.11.2 + 19.gfaa0216)[77] on the cells (snRNA-seq or gene activities from snATAC–seq). To obtain the circadian rhythm signatures, we used the scRNA-seq data from the mouse liver at different timepoints of the circadian rhythm from a previous study[81], performing differential expression analysis between the different timepoints with Seurat (v.4.0.03)[71]. To further validate the circadian rhythm effects, we generated two additional multiome datasets on unstarved mice (using the 10x protocol). Data were processed as previously described and combined with the previous samples and analysed using VSN-pipelines (v.0.27.0, for the snRNA-seq layer) and pycisTopic (v.1.0.1.dev75 + g3d3b721, for the snATAC–seq layer, using 100 topics). Combined genome coverage and gene expression plots were performed using Signac (v.1.10.0)[82].

#### Molecular cartography in the mouse liver

**Gene panel selection.** In total, 100 genes were selected on the basis of their gene expression patterns (marker genes for a cell type or group of cell types) on our in-house mouse liver dataset and literature (Supplementary Table 1). Moreover, we performed dimensionality reduction using only these 100 genes to ensure that all cell types could be distinguished with this gene panel.

**Tissue sections.** Livers from three different mice were used in this experiment. Mouse liver samples were fixed with PAXgene Tissue FIX solution (Resolve Biosciences) for 24 h at room temperature followed by 2 h in the PAXgene Tissue Stabilizer (Resolve Biosciences) at room temperature. The samples were cryoprotected in a 30% sucrose solution (w/v) overnight at 4 °C and frozen in 2-methylbutane (Sigma-Aldrich, 106056) on dry ice. Frozen samples were sectioned with a cryostat (Leica, CM3050) and 10 μm thick sections were placed within the capture areas of cold Resolve Biosciences slides. The samples were then sent to Resolve BioSciences on dry ice for analysis. After arrival, the tissue sections were thawed and rehydrated with isopropanol, followed by 1 min washes in 95% ethanol and 70% ethanol at room temperature. The samples were used for molecular cartography (100-plex combinatorial smFISH) according to the manufacturer's instructions (protocol v.1.3; available for registered users), starting with the aspiration of ethanol and the addition of buffer DST1 followed by tissue priming and hybridization. In brief, tissues were primed for 30 min at 37 °C followed by overnight hybridization of all probes specific for the target genes (see below for probe design details and target list). The samples were washed the next day to remove excess probes and fluorescently tagged in a two-step colour development process. Regions of interest were imaged as described below and fluorescent signals removed during decolorization. Colour development, imaging and decolorization were repeated for multiple cycles to build a unique combinatorial code for every target gene that was derived from raw images as described below.

**Probe design.** The probes for the 100 selected genes were designed using Resolve's proprietary design algorithm. In brief, probe design was performed at the gene level. For every targeted gene, all full-length protein-coding transcript sequences from the ENSEMBL database were used as design targets if the isoform had the GENCODE annotation tag 'basic'[83]. To speed up the process, the calculation of computationally expensive parts, especially the off-target searches, the selection of probe sequences was not performed randomly, but limited to sequences with high success rates. To filter highly repetitive regions, the abundance of $k$-mers was obtained from the background transcriptome using Jellyfish[84]. Every target sequence was scanned once for all $k$-mers, and those regions with rare $k$-mers were preferred as seeds for full probe design. A probe candidate was generated by extending a seed sequence until a certain target stability was reached. A set of simple rules was applied to discard sequences that were found experimentally to cause problems. After these fast screens, every kept probe candidate was mapped to the background transcriptome using ThermonucleotideBLAST[85] and probes with stable off-target hits were discarded. Specific probes were then scored based on the number of on-target matches (isoforms), which were weighted by their associated APPRIS level[86], favouring principal isoforms over others. A bonus was added if the binding site was inside the protein-coding region. From the pool of accepted probes, the final set was composed by greedily picking the highest scoring probes. Gene names and catalogue numbers for the specific probes designed by Resolve BioSciences are included in Supplementary Table 1.

**Imaging.** Samples were imaged on a Zeiss Celldiscoverer 7, using the ×50 Plan Apochromat water-immersion objective with an NA of 1.2 and the ×0.5 magnification changer, resulting in a ×25 final magnification. Standard CD7 LED excitation light source, filters and dichroic mirrors were used together with customized emission filters optimized for detecting specific signals. The excitation time per image was 1,000 ms for each channel (DAPI was 20 ms). A $z$ stack was taken at each region with a distance per $z$ slice according to the Nyquist-Shannon sampling theorem. The custom CD7 CMOS camera (Zeiss Axiocam Mono 712, 3.45 μm pixel size) was used. For each region, a $z$ stack per fluorescent colour (two colours) was imaged per imaging round. A total of eight imaging rounds was performed for each position, resulting in 16 $z$ stacks per region. The completely automated imaging process per round (including water-immersion generation and precise relocation of regions to image in all three dimensions) was realised by a custom Python script using the scripting API of the Zeiss ZEN software (open application development, v.2023.02.27).

**Spot segmentation.** The algorithms for spot segmentation were written in Java and are based on the ImageJ library functionalities. Only the iterative closest point algorithm is written in C++ based on the libpointmatcher library (https://github.com/ethz-asl/libpointmatcher).

**Preprocessing.** As a first step, all of the images were corrected for background fluorescence. A target value for the allowed number of maxima was determined based on the area of the slice in μm² multiplied by the factor 0.5. This factor was empirically optimized. The brightest maxima per plane were determined, based on an empirically optimized threshold. The number and location of the respective maxima was stored. This procedure was performed for every image slice independently. Maxima that did not have a neighbouring maximum in an adjacent slice (called $z$-group) were excluded. The resulting maxima list was further filtered in an iterative loop by adjusting the allowed thresholds for $(B_{abs} - B_{back})$ and $(B_{peri} - B_{back})$ to reach a feature target value ($B_{abs}$, absolute brightness; $B_{back}$, local background; $B_{peri}$, background of periphery within 1 pixel). These feature target values were based on the volume of the 3D image. Only maxima still in a $z$-group of at least 2 after filtering passed the filter step. Each $z$-group was counted as one hit. The members of the $z$-groups with the highest absolute brightness were used as features and written to a file. They resemble a 3D point cloud. For the final signal segmentation and decoding, to align the raw data images from different imaging rounds, images had to be corrected. To do so, the extracted feature point clouds were used to find the transformation matrices. For this purpose, an iterative closest point cloud algorithm was used to minimize the error between two point clouds. The point

clouds of each round were aligned to the point cloud of round one (reference point cloud). The corresponding point clouds were stored for downstream processes. On the basis of the transformation matrices, the corresponding images were processed by a rigid transformation using trilinear interpolation. The aligned images were used to create a profile for each pixel consisting of 16 values (16 images from two colour channels in 8 imaging rounds). The pixel profiles were filtered for variance from zero normalized by total brightness of all pixels in the profile. Matched pixel profiles with the highest score were assigned as an ID to the pixel. Pixels with neighbours with the same ID were grouped. The pixel groups were filtered by group size, number of direct adjacent pixels in group, number of dimensions with size of two pixels. The local 3D maxima of the groups were determined as potential final transcript locations. Maxima were filtered by number of maxima in the raw data images where a maximum was expected. The remaining maxima were further evaluated by the fit to the corresponding code. The remaining maxima were written to the results file and considered to resemble transcripts of the corresponding gene. The ratio of signals matching to codes used in the experiment and signals matching to codes not used in the experiment were used as estimation for specificity (false positives).

**Visualization and nucleus segmentation.** Final image analysis was performed in ImageJ (v.2.3.0/1.53f) using the Polylux tool plugin (v.1.6.1) from Resolve BioSciences to examine specific molecular cartography signals. Nucleus segmentation was performed using QuPATH (v.4.2.1)[87] based on the DAPI signal, setting pixel size to 0.25, minimum area to 10, maximum area to 400, sigma to 1.7 and cell expansion 8. Data were analysed using Seurat (v.4.0.3)[71]. Using 14 PCs, we performed Leiden clustering, resulting in 19 clusters that corresponded to 11 cell types that were annotated on the basis of marker gene expression.

**Single-cell data mapping.** The liver lobule representation was used to generate the virtual liver lobule template coordinates. The template was reduced to a size of 100 × 100 pixels and was split into one image per cell type (in red colour). Each image was read using the jpeg (v.0.1-8) R package, and the background (in white colour) was removed using *k*-means clustering on the RGB pixel values. Cells in the bile duct were labelled as BECs; in the portal vein we included the periportal VECs and fibroblasts and we mapped the pericentral VECs around the central vein. The remaining cell types were spread in the lobule randomly based on the proportions of the cell types in the snRNA-seq data. Multiome cells (with gene expression, chromatin accessibility, regulatory topics and eGRNs) were mapped into a template of the liver lobule and the smFISH spatial map using ScoMAP (v.0.1.0)[12]. In brief, zonated cell types (hepatocytes, LSECs and HSCs) were first ordered by pseudotime using the DPT() function of the destiny R package (v.3.2.0)[80], using as input the harmony-corrected PCs from the snRNA-seq layer. The pseudotime order represents the distance along the portocentral axis. Each cell type was divided into ten bins based on their pseudotime order. In the liver lobule template, we calculated the distance of each metacell (that is pixel) from the central vein, and divide the cells in ten bins. The zonated cell types were ordered on the basis of pseudotime using PCs calculated by Seurat[71] (v.4.0.3, which represented the distance along the portocentral axis as well) and divided into ten bins. For each cell type, we assigned a real profile from the matching bin to each virtual cell randomly (for example, the cells in the first bin of a pseudotime ordered cell type are assigned to the virtual cells in the first bin of that cell type based on the distance to the central vein in the liver lobule template, or to the same bin in the pseudotime order for the smFISH template). For non-spatially-located cell types, cells were sampled randomly without binning based on the annotations between the templates and the single-cell data. If there are more real cells than virtual ones, random sampling is done without repetition; if there are more virtual cells than real ones, real profiles are assigned more than once. The gene expression, region accessibility, topic contribution

and eRegulon enrichment values of the virtual cells are those of their matching real cell. These approaches are included in the ScoMAP R package, with detailed tutorials available at GitHub (https://github.com/aertslab/ScoMAP).

### Analysis of snRNA-seq data from male and female livers

snRNA data from male and female livers was obtained from a previous study[88]. The fastq files were downloaded from the SRA Project PRJNA779049 (Vehicle_Female_liver**:** SAMN23009762 and Vehicle_Male_liver: SAMN23009760). The fastq files were processed with Cell Ranger's (v.7.0.1) count function. Reads were aligned to the *M. musculus* reference genome (mm10-2020-A), including intronic reads in the counting. The gene expression matrix for each sample was analysed first independently using VSN-pipelines (v.0.27.0). In brief, cells with at least 350 genes expressed and a percentage of mitochondrial reads below 10% were retained. Scanpy (v.1.8.2) was run with the default parameters, using the number of principal components automatically selected by VSN-Pipelines and using Leiden clustering with resolution 1. Hepatocyte clusters with low gene expression and high percentage of mitochondrial reads were removed, as well as doublets called with Scrublet (v.0.2.3). After filtering, the female sample contained 5,342 high-quality cells and the male sample contained 4,860 high-quality cells. Differentially expressed genes were calculated using Seurat's FindMarkers function (v.4.0.3), retaining genes with log[FC] > 0.75 and adjusted *P* < 0.05.

### TBX3 ChIP–seq analysis of mouse hepatocytes

Mouse hepatocytes were freshly isolated from a mouse according to the same procedure described in the 'FACS MPRA in vivo' section. In brief, C57BL/6 mice were anaesthetized and the livers were perfused with SC-1 and SC-2 medium. The liver lobes were dissected and treated with collagenase P. Hepatocytes were collected by centrifugation for 2 min at 50*g*, washed with PBS and resuspended in DMEM + 10% FBS + 10 mM HEPES. ChIP–seq was performed by following the Myers Lab ChIP–seq Protocol v011014 on 2 × 10⁷ hepatocytes. A total of 5 µl of bethyl rabbit anti-TBX3 antibody (1 µg per µl, Sanbio, A303-098A, 1) was used for ChIP. A total of 10 ng of immunoprecipitated DNA was used to perform library preparation according to the Illumina TruSeq DNA Sample preparation guide. In brief, the immunoprecipitated DNA was end-repaired, A-tailed and ligated to diluted sequencing adapters (1/100). After PCR amplification (18 cycles) and bead purification (Agencourt AmpureXP, Analis), the libraries with fragment size of 300–500 bp were sequenced using the NextSeq 2000 (Illumina) system. Reads were mapped to the mm10 genome using Bowtie2 (v.2.3.5.1)[78], peaks were called using MACS2 (v.2.2.7.1, with --format BAM --gsize mm --qvalue 0.01 --call-summits as options) and bigwig files were generated using deepTools[79] bamCoverage function (v.3.5.0, with --normalizeUsing CPM --binSize 1 as parameters). Motif enrichment was performed using pycisTarget (v.1.0.1.dev42+gb6707ee) with the default parameters on the top 1,000 ChIP–seq regions. Coverage heat maps were created using deepTools[79] (v.3.5.0).

### Culture of HepG2, Hepa1-6 and AML12 cells

The HepG2, Hepa1-6 and AML12 cell lines were purchased from ATCC (HB-8065, CRK-1830 and CRL-2254, respectively). HepG2 cells were cultured in Eagle's minimum essential medium (EMEM, Thermo Fisher Scientific), Hepa1-6 cells were cultured in Dulbecco's Modified Eagle's Medium (DMEM, Gibco), and AML12 in DMEM-F12 medium (Gibco). All media were supplemented with 10% fetal bovine serum (Thermo Fisher Scientific) and 50 µg ml⁻¹ penicillin–streptomycin (Gibco). Cell cultures were kept at 37 °C, with 5% CO₂.

### HepG2, Hepa1-6 and AML12 OmniATAC–seq

Omni-assay for transposase-accessible chromatin using sequencing (OmniATAC–seq) was performed as described previously[89]. In brief,

50,000 cells obtained after FACS were resuspended in 50 μl of cold ATAC−seq resuspension buffer (RSB; 10 mM TrisHCl (pH 7.4), 10 mM NaCl, and 3 mM MgCl$_2$ in water) containing 0.1% IGEPAL CA-63, 0.1% Tween-20 and 0.01% digitonin by pipetting up and down three times. This cell lysis reaction was incubated on ice for 3 min. After lysis, 1 ml of ATAC−seq RSB containing 0.1% Tween-20 was added, and the tubes were inverted to mix. Nuclei were then centrifuged for 10 min at 500*g* in a prechilled (4 °C) fixed-angle centrifuge. The supernatant was removed and nuclei were resuspended in 50 μl of transposition mix (25 μl 2× TD buffer, 2.5 μl transposase (Nextera Tn5 transposase, Illumina), 16.5 μl PBS, 0.5 μl 1% digitonin, 0.5 μl 10% Tween-20 and 5 μl water) by pipetting up and down six times. Transposition reactions were incubated at 37 °C for 30 min in a thermoblock. Reactions were cleaned-up by MinElute (Qiagen). Transposed DNA was amplified with primers i5_Indexing_For and i7_Indexing_Rev (Supplementary Table 4). The number of PCR cycles was based on quantitative PCR as described previously[90]. All libraries were sequenced on the NextSeq 2000 instrument (Illumina) with the following sequencing parameters: 51 bp read 1, 8 bp index 1, 8 bp index 2, 51 bp read 2. Adapters were removed with fastq-mcf (ea-utils, v.1.12), cleaned reads were mapped to the hg19 (HepG2) or mm10 (AML12 and Hepa1-6) genome using Bowtie2 (v.2.3.5.1)[78] and bigwig files were generated using deepTools[79] bamCoverage function (v.3.5.0, with --normalizeUsing CPM --binSize 1 as parameters). Coverage heat maps were created using deepTools[79] (v.3.5.0).

## MPRAs

**Library design.** For the first library (hereafter, the 12K library), we selected 10,845 candidate regions based only on accessibility in hepatocytes. This set also includes shared regions (accessible in hepatocytes and at least one other cell type, 4,163, out of which 1,386 are accessible in hepatocytes and BECs); regions specifically accessible across all hepatocytes (4,357); and regions that are only accessible in periportal, intermediate and pericentral hepatocytes (795, 527 and 656, respectively). Note that, in these experiments, we assess the capacity of the sequences to activate a minimal promoter (that is, enhancer activity), rather than their activity as promoters. Moreover, we included 795 shuffled regions as a negative control, and 360 positive controls that previously showed activity in HepG2, and for which a mouse orthologous region can be found that is also accessible in the mouse liver. For this latter subset, we included both the human and the corresponding mouse sequences[60,91,92]. For each region, we selected a 258 bp sequence centred at the peak summit, to which we added a 12 bp barcode selected from https://github.com/hawkjo/freebarcodes, excluding those with repeats (more than the same nucleotide 6 times in a row). The sequences were flanked with the adaptors CCAGTGCAAGTGCAG and GGCCTAACTGCCGG in 5′ and 3′, respectively, resulting in 300 bp sequences. The final library was synthesized by Twist Bioscience as an oligo pool. For the second library (455 library), we selected 13 periportal enhancers, 21 pericentral enhancers, 2 positive controls (*Aldob* and *LTV1*[41]) and 44 shuffled regions. For each enhancer, we manually introduced mutations affecting activity and/or zonation based on the saturation mutagenesis of DeepLiver. We then selected 259 bp windows, based on the information content from DeepExplainer. For enhancers in which the 259 bp window did not cover all of the relevant nucleotides, we selected more than one window. This resulted in 16 periportal and 25 pericentral windows, with 370 sequence variants in total. For each region, we added an 11 bp barcode selected from https://github.com/hawkjo/freebarcodes, excluding those with repeats (more than the same nucleotide 6 times in a row). The sequences were flanked with the adaptors CCAGTGCAAGTGCAG and GGCCTAACTGGCCGG in 5′ and 3′, respectively, resulting in 300 bp sequences. The final library was synthesized by Twist Bioscience as an oligo pool.

**Enhancer library cloning.** The pSA293-CHEQseq plasmid (Addgene, 174669), containing a SCP1 promoter, a chimeric intron and the Venus

cDNA, was used as a reporter plasmid for MPRA. Two different versions of that plasmid were used for the cloning of the 12k library: pSA293-CHEQseq-5′BC contains a random 17 bp barcode (BC) upstream of the chimeric intron and pSA293-CHEQseq-3′BC-1 contains a random 17 bp barcode between the Venus and the poly(A) tail[36]. The 455 library was cloned in a newly generated pSA293-CHEQseq-3′BC-2 containing an 18 bp barcode-optimized for Oxford Nanopore Technologies sequencing with the following pattern: NNNYRNNNYRNNNYRNNN. The oligonucleotide libraries were resuspended according to the manufacturer's recommendation and amplified by PCR with the primers CHEQ_liver_For and CHEQ_liver_Rev (Supplementary Table 4). To clone the amplified enhancer library upstream of the SCP1 promoter, the vectors were linearized by inverse PCR with primers CHEQ_lin_For and CHEQ_lin_Rev (Supplementary Table 4). Amplified libraries and the corresponding linearized vector were combined in an NEBuilder reaction with a vector to insert ratio of 1:5. The NEBuilder reactions were dialysed against water in a 6 cm Petri dish with a membrane filter MF-Millipore 0.05 μm (Merck) for 1 h. The reactions were recovered from the membrane, and 2.5 μl of the reaction was transformed into 25 μl of Lucigen Endura ElectroCompetent Cells (Biosearch Technologies). Before culture for maxiprep, 1:100,000 of the transformed bacteria was plated onto an LB-agar dish with carbenicillin to estimate the complexity of the cloned library. A volume of bacteria corresponding to a complexity of 500 barcodes per enhancer was put in culture for maxiprep. Maxiprep was performed using the Nucleobond Xtra endotoxin-free maxiprep kit (Macherey-Nagel).

**Enhancer-barcode assignment.** For the liver library cloned in the pSA293-CHEQseq-5′BC plasmid, a PCR amplification (12 cycles) of the enhancer, together with the random barcode, was performed with the primers Enh_BC_5′_For and Enh_BC_5′_Rev (Supplementary Table 4). Illumina sequencing adaptors were added during a second round of PCR with the primers i5_Indexing_For and i7_Indexing_Rev (Supplementary Table 4). Before sequencing, the fragment size of every library was analysed using the Bioanalyzer high-sensitivity chip. All libraries were sequenced on a NextSeq 2000 instrument (Illumina) with the following sequencing parameters: 51 bp read 1, 8 bp index 1, 8 bp index 2, 51 bp read 2. Reads were first processed with fastqc (v.0.11.8) to assess their quality, and then trimmed using cutadapt (v.1.18)[93] with the options -g TGTCCCCAGTGCAAGTGCAG --discard-untrimmed -m 12 -l 12 for read 1 to extract the enhancer barcode and options -g AATTAATTCGGGCCCCGGTCC…GATCGGCGCGCCTGCTCG -j 10 --discard-untrimmed -m 17 -M 17 for read 2 to extract the plasmid barcode. For read 2, we then used seqkit (v.0.10.2)[94], with the options seq -r -p, to get the reverse complement sequence. Reads were filtered to retain only those with quality > 30 using fastp (v.0.20.0)[95]. This resulted in 8,835,050 enhancer-barcode assignments for the 5′ 12K library, with 78.3% barcodes assigned to a unique enhancer. For the liver libraries cloned in the pSA293-CHEQseq-3′BC plasmids, we performed Nanopore sequencing as follows. A total of 1.5 μg of the library was linearized by digestion with NcoI according to the manufacturer's protocol. We next processed 200 ng of the cleaned up and linearized plasmid with an Oxford Nanopore Technologies Q20+ ligation sequencing kit early-access SQK-LSK112 according to the manufacturer's protocol (Genomic DNA by Ligation; revision GDE_9141_v112_revC_01Dec2021). A total amount of 10 fmol of the prepared library was loaded onto a MinION R10.4 flow cell and sequenced for at least 72 h (MinKNOW, v.21.11.09 or later). Raw signal fast5 files were rebasecalled using Guppy (v.6.0.7) using the super-accuracy model (dna_r10.4_e8.1_sup). To process the reads, we first generated a synthetic genome consisting of the plasmid sequence (with 100 bp flanks) with all possible enhancers inserted. Reads were mapped using minimap2 (v.2.22)[96] with the options -ax map-ont --secondary=no -N1 -f 500 and a bam file was generated using SAMTools (v.1.11)[97]. Next, we used the R package GenomicAlignments (v.1.24.0) to extract the sequences overlapping the enhancer sequence

and the enhancer barcodes from the reads and calculate the number of mismatches. For the 455 library, we retained only those assignments with no mismatches, as many enhancer sequences differ in few base pairs only, and those assignments in which the plasmid barcode sequence matches with NNNNYRNNNYRNNNYRNNN. This resulted in libraries with 723,805 and 71,658 unique enhancer-barcode assignments, with 85.5% and 97.8% barcodes assigned to a unique enhancer, respectively.

**Bulk MPRA in vitro.** The MPRA libraries were transfected in HepG2 and AML12 cells using the Lipofectamine 3000 reagent (Thermo Fisher Scientific). In brief, 4 million cells were seeded in a 10 cm cell culture dish. The next day, when cells reach 70–90% confluency, a tube A with 500 μl opti-MEM (Thermo Fisher Scientific) and 25 μl Lipofectamine 3000 reagent and a tube B with 500 μl opti-MEM and 15 μg of the liver enhancer library were prepared and incubated for 5 min at room temperature. Tube B was mixed carefully with tube A and incubated for 15 min at room temperature. The medium of the cells was also changed to opti-MEM medium and finally the mixture was added dropwise to the cells. Then, 48 h after transfection, cells were detached from the plate using trypsin (Thermo Fisher Scientific). One-fifth of the cells was used for plasmid DNA extraction (Qiagen). The remaining cells were used for RNA extraction using the innuPREP RNA Mini Kit 2.0 (Analytik Jena), followed by mRNA isolation using the Dynabeads mRNA purification kit (Ambion) and cDNA synthesis using the GoScript RT Kit with oligo dT primer (Promega). To amplify the random 5′ and 3′ barcode from the plasmid DNA or cDNA sample, a PCR was performed for 16 cycles using the CHEQseq_barcode_5′_For, CHEQseq_barcode_5′_Rev and CHEQseq_barcode_3′_For and CHEQseq_barcode_3′_Rev primers (Supplementary Table 4), respectively. To add Illumina sequencing adaptors, all of the samples were finally amplified by PCR for six cycles with the primers i5_Indexing_For and i7_Indexing_Rev. For the 12K library, two experiments were performed in HepG2 and AML12 cells, respectively; for the 455 library, three experiments were performed in HepG2 cells.

**Bulk MPRA in vivo.** For intrahepatic delivery of the liver MPRA libraries, mice (aged 8 to 10 weeks) were secured and hydrodynamically injected with 20 μg of the libraries through the lateral tail vein. All of the libraries were diluted in sterile filtered 0.9% NaCl, and the total volume was adjusted to 10% (in ml) of the total body weight (in grams). Then, 24 and 48 h after injection, for the 12K library and the 455 library, respectively, mice were anaesthetized by intraperitoneal injection of sodium pentobarbital (Nembutal, 50 mg per kg) and whole livers were isolated. Liver tissues were homogenized by using M tubes (Miltenyi Biotec) and the GentleMACS Dissociator (Miltenyi Biotec). RNA and plasmid DNA extraction, mRNA purification and cDNA preparation, and barcode amplification were performed as described for MPRA in vitro. In total, 3 and 4 mice were used for the 3′ and 5′ 12k library experiments, respectively; and 8 mice were used for the 455 library experiments.

**FACS MPRA in vivo.** For intrahepatic delivery of the liver MPRA libraries, four mice (aged 8–10 weeks) were secured and hydrodynamically injected with 20 μg of the libraries through the lateral tail vein. All of the libraries were diluted in sterile filtered 0.9% NaCl, and the total volume was adjusted to 10% (in ml) of the total body weight (in grams). Then, 48 h after injection, the mice were anaesthetized by intraperitoneal injection of sodium pentobarbital (Nembutal, 50 mg per kg). The livers were perfused for 5 min with 40 ml of perfusion medium SC-1 (8 g l⁻¹ NaCl, 400 mg l⁻¹ KCl, 75.5 mg l⁻¹ NaH$_2$PO$_4$, 120.5 mg l⁻¹ Na$_2$HPO$_4$, 2.38 g l⁻¹ HEPES, 350 mg l⁻¹ NaHCO$_3$, 190 mg l⁻¹ EGTA, 900 mg l⁻¹ D-(+)-glucose, 1.2 ml phenol red solution) to remove the blood, followed by perfusion with 40 ml of SC-2 medium (8 g l⁻¹ NaCl, 400 mg l⁻¹ KCl, 75.5 mg l⁻¹ NaH$_2$PO$_4$, 120.5 mg l⁻¹ Na$_2$HPO$_4$, 2.38 g l⁻¹ HEPES, 350 mg l⁻¹ NaHCO$_3$, 560 mg l⁻¹ CaCl$_2$·2H$_2$O, 1.2 ml phenol red solution) containing 10 mg of collagenase P (Merck) for 5 min. Each lobe was dissected off and minced into small pieces in a beaker containing 39 ml SC-2 supplemented with 1 ml DNase I (Sigma-Aldrich) and 20 mg collagenase P, followed by rotating incubation for 15 min at 37 °C. Hepatocytes were centrifuged for 2 min at 50g, washed with PBS, centrifuged again for 2 min at 50g, resuspended in 3 ml Hoechst buffer (DMEM + 10% FBS + 10 mM HEPES) and filtered through a 70 μm strainer. The protocol for hepatocyte staining was adapted from a previous study[43]. After counting the cells on the LUNA cell counter, the concentration was adjusted to 5 million cells in 1 ml of Hoechst buffer. To determine the ploidy of hepatocytes, DNA was stained with Hoechst (Thermo Fisher Scientific) (15 μg ml⁻¹). Reserpine (5 μM) was also supplemented to prevent Hoechst expulsion from the cells. Cells were incubated for 30 min at 37 °C. Hepatocytes were centrifuged for 5 min at 1,000 rpm at 4 °C and the supernatant was discarded. Cells were resuspended in cold PBS in a concentration of 1 million cells in 100 μl. After centrifuging (1,000 rpm for 5 min at 4 °C), cells were resuspended in FACS buffer (2 mM EDTA, pH 8, and 0.5% BSA in 1× PBS) at a concentration of 1 million cells in 100 μl. Cells were stained with the following antibodies (BioLegend) at a dilution of 1:300: PE anti-mouse/human CD324 E-cadherin (147304) and APC anti-mouse CD73 (127210). FcX blocking solution (BioLegend, 101319) was added at a dilution of 1:50. Cells were sorted using the FACS-Aria-Fusion (BD Biosciences) system using a 100 μm nozzle and analysed with FACSDiva (v.9.0.1). FSC-A and SSC-A were used for hepatocyte size selection. Cells containing the library were selected based on GFP. Tetraploid hepatocytes were selected based on Hoechst stain. CD73 and ECAD were used to select hepatocyte bins along the portocentral axis, obtaining 100,000–200,000 cells per bin. RNA extraction, mRNA purification and cDNA preparation were performed as described for MPRA in vitro. To amplify the enhancer barcode on the cDNA, small modifications were made. To amplify the random 3′ barcode, a PCR with 24 cycles was performed. To add Illumina sequencing adaptors, a PCR with 10 cycles was performed. In total, four mice were used.

**MPRA data analysis.** CHEQ-seq barcodes were extracted from the plasmid and cDNA samples (read 2) using cutadapt (v.1.18)[93] with parameters with options -g TTATCATGTCTGCTCGAAGC…GATCGG CGCGCCTGCTCG --discard-untrimmed -m 17 -M 17 for the 12K libraries and g GTATCTTATCATGTCTGCTCGAAGC…GATCGGC -j 10 --discard-untrimmed -m 18 -M 18 for the 455 library, and seqkit (v.0.10.2)[94], with options seq -r -p, was used to get the reverse complement sequence. Reads were filtered to retain only those with quality > 30 using fastp (v.0.20.0)[95]. Reads were assigned to enhancers based on the corresponding enhancer-barcode assignments, resulting in a count matrix with number of reads per enhancer and sample. Samples were processed using DESeq2 (v.1.37.6)[98], comparing the corresponding cDNA replicates versus their plasmid samples. For the FACS fractions, as we did not extract plasmid DNA from the samples, we used the plasmid replicates from the in vivo bulk experiment. To assess enhancer activity, we used the log[FC] calculated using DESeq2 (v.1.37.6)[98]. To distinguish active and inactive enhancers, a Gaussian fit of the shuffled negative control values was performed using robustbase (v.0.93-6), and a P value and Benjamini–Hochberg-adjusted P value was calculated based on that Gaussian fit for all enhancers. An enhancer is considered to be active if its adjusted $P < 0.1$.

**FACS ATAC–seq.** Hepatocytes were isolated and stained as described in the 'FACS MPRA in vivo' section, with minor modifications. Cells were additionally stained with Alexa Fluor 488 Zombie Green (BioLegend) to enable the detection of viable cells by FACS. Zombie Green was added at a dilution of 1:500 and cells were kept in a rotator in the dark at room temperature for 15 min. Cells were sorted on the FACS-Aria-Fusion (BD Biosciences) system using a 100 μm nozzle. FSC-A and SSC-A were used for hepatocyte size selection. Viable cells were selected based on the Zombie Green signal. Tetraploid hepatocytes were selected based

on Hoechst stain. CD73 and ECAD were used to select hepatocyte bins along the portocentral axis, obtaining 20,000–50,000 cells per bin. OmniATAC–seq was performed as described previously (see the 'HepG2, AML12 and Hepa1-6 OmniATAC–seq' section)[89], using 20,000–50,00 cells obtained after FACS as input. Adapters were removed with fastq-mcf (ea-utils, v.1.12) and cleaned reads were mapped to the mm10 genome using Bowtie2 (v.2.3.5.1)[78]. A fragment count matrix was generated using the liver snATAC–seq consensus peaks using SubRead (v.1.6.3)[99]. AUCell (v.1.22.0)[77] was used to assess the enrichment of the core signatures (general, periportal, pericentral-intermediate and pericentral) in each of the fractions, using the default parameters.

## DeepLiver

**Model training.** The top 3,000 regions in each topic (based on the region-topic distributions) were used as the input for a deep learning model, whereby 500 bp DNA sequences were used to predict the topic set to which the region belongs (topic-CNN). The model is a hybrid CNN–recurrent-neural-network multiclass classifier and its architecture was adopted from earlier studies[55,56,58], trained with Tensorflow (v.1.15) with minor adaptations. In brief, we used 1,024 filters and a filter size of 24. To initialize the filters, we used 725 PWMs derived from running differentially enriched motifs between selected cell-type-specific topics (topics 48 (hepatocytes), 66 (periportal hepatocytes), 58 (pericentral hepatocytes), 38 (Kupffer cells), 71 (LSECs), 32 (HSCs), 27 (fibroblasts) and 42 (BECs)), with $\log_2[FC] > 1.5$ and adjusted $P < 0.0001$ as thresholds. The zonation-CNN was trained using regions derived from the curated shared hepatocytes topics and DARs (classified as general, pericentral and periportal after hierarchical clustering, resulting in 12,122, 4,181 and 1,372 regions, respectively), while the MPRA-CNN was trained using the binarized log[FC] distributions from the 12K in vivo MPRA (with adjusted $P < 0.01$, resulting in 1,232 and 2,983 high-confidence active and inactive regions, respectively). The weights derived from the topic-CNN model were used to initialize these two models, an approach known as transfer learning. The zonation-CNN and the MPRA-CNN, which have the same architecture as the topic-CNN, were trained with identical parameters, except for the learning rate, which was set to 0.00001 instead of 0.001.

**Model performance.** To assess the performance of the models, we performed ninefold cross validation. In brief, the data were divided into ten groups and, in each iteration, we used eight groups for training (80% of the regions), one group as a validation set (10% of the regions) and one group as a test set (10% of the regions). To increase the sample size for the deep learning model, we augmented the regions by extending them to 700 bp and used a sliding window of 500 bp with a 50 bp stride, increasing the sample size five times. During the training, the validation set was used for early stopping and the 12th, 66th and 122nd epochs were chosen to evaluate the performance of cross-validation models for the topic-CNN, the zonation-CNN and the MPRA-CNN, respectively. After training, we assessed the performance of the models on the non-augmented test set by scoring the test set regions with the models. Then, using the prediction scores and the labels (topics, zonation class or activity pattern for the topic-CNN, zonation-CNN and MPRA-CNN, respectively), we calculated the AUPR and AUROC using the average_precision_score and roc_auc_score functions from the scikit-learn package (v.0.21.3). To validate DeepLiver predictions, we calculated the correlation between the predictions of DeepLiver and a previously published MPRA dataset performed on synthetic sequences in vivo[38]. These sequences were designed by adding different number of instances and combinations of motifs corresponding to TFs that are relevant to hepatocytes, including HNF1A, HNF4A (COUPTF), CEBPA, ONECUT1 (HNF6) and FOXA1 (HNF3), among others.

**Nucleotide contributions.** To find the nucleotides that contribute the most to the topic prediction, we used a DeepExplainer, included in the SHAP package (v.0.37.0)[39], using the default parameters and 500 random sequences for initialization. The importance score obtained from the DeepExplainer analysis was multiplied by the one-hot encoded DNA sequence and visualized as the height of the nucleotide letters as in earlier work[100]. In addition to the nucleotide importance plots, we performed in silico saturation mutagenesis in which we calculated the effect of each variant of a region on its model prediction score. The sequences with all possible single mutations were generated and the delta prediction score for each topic was calculated. To validate DeepLiver in silico mutagenesis, we calculated the correlation between the effect of the mutations predicted by DeepLiver and experimental saturation mutagenesis data on six enhancers from earlier studies (three enhancers from in vivo studies and three enhancers from HepG2 cells)[41,42].

**TF-binding site predictions.** High nucleotide importance on DeepExplainer plots represents potential binding sites for TFs. We used TF-Modisco (v.0.5.14.1)[40] to identify the most common patterns along the zonation classes (general, pericentral and periportal, using the zonation-CNN) and active versus inactive enhancers (using the MPRA-CNN). To run TF-Modisco, we used MEME initialization, a sliding window of 15 bp, 10 as flank size and a false-discovery-rate threshold of 0.15. We next used TF-Modisco patterns and selected PWMs to score the sequences. In brief, we trimmed them using trim_by_ic=0.25, and the sum score was calculated using compute_sum_scores on the nucleotide importance scores. We converted the patterns and PWMs to convolutional filters and calculated pattern activation scores using the tf.nn.conv1d function of TensorFlow (v.1.15). Global motif instances were calculated for the curated shared hepatocyte regions (general, pericentral and periportal). Optimal thresholds were selected manually. To validate the predicted binding hits, we used the ChIP–seq data for HNF4A, ONECUT1, CEBPA and FOXA1[29], as described for the SCENIC+ eRegulons.

## TBX3 and TCF7L1 perturbation simulation

TBX3 and TCF7L1 computational perturbations were performed using SCENIC+ (v 0.1.dev411+gf4bcae5.d20220810)[7]. In brief, based on the inferred eGRN, we first trained a GBM model per gene, in which we predict the gene expression using its predicted regulators (that is TFs). To simulate knockdowns, we set the expression of the selected TF to 0 across all cells and recalculated the predicted gene expression matrix using the previously trained models. To simulate overexpression, we set the expression of the TFs to the maximum expression value in the dataset on hepatocyte cells. Predictions are updated over several iterations, to account for downstream effects.

## Luciferase reporter assay

Synthetic liver sequences were ordered as gBlocks from Integrated DNA Technologies. The pGL4.23-GW luciferase reporter vector (Promega) was linearized by inverse PCR with primers LUC_lin_For and LUC_lin_Rev (Supplementary Table 4). The synthetic sequences and the linearized vector were combined in an NEBuilder reaction and 2 µl of the reaction was transformed into 25 µl of Stellar chemically competent bacteria. HepG2 cells were seeded into 24-well plates at a density of 100,000 cells per well and transfected with 400 ng pGL4.23-enhancer vector + 40 ng pRL-TK *Renilla* vector (Promega) with Lipofectamine 3000 reagent. Then, 1 day after transfection, the luciferase activity was measured using the Dual-Luciferase Reporter Assay System (Promega) according to the manufacturer's protocol. In brief, cells were lysed with 100 µl of Passive Lysis Buffer for 15 min at 500 rpm. A total of 20 µl of the lysate was transferred in duplicate in a well of an OptiPlate-96 HB (PerkinElmer) and 100 µl of luciferase assay reagent II was added in each well. Luciferase-generated luminescence was measured on the Victor X luminometer (PerkinElmer). A total of 100 µl of the Stop & Glo Reagent was added to each well, and the luminescence was measured

again to record *Renilla* activity. Luciferase activity was estimated by calculating the ratio luciferase/*Renilla*. This value was normalized to the ratio calculated on blank wells containing only reagents. Four biological replicates were performed per condition.

### Human liver data analysis

Human liver data were obtained from a previous study[44]. The author labels and the snATAC–seq fragments were used as an input for pycisTopic (v.1.0.1.dev75 + g3d3b721)[7]. In brief, we first inferred consensus peaks as previously described, resulting in a dataset with 121,593 regions and 6,366 cells. Topic modelling was performed using Mallet (v.2.0), using 500 iterations and models with 2 topics and from 5 to 100 by an increase of 5. Drop-out imputation was performed by multiplying the region-topic and topic-cell probabilities. The imputed accessibility matrix was multiplied by $10^6$. The mouse region-based eRegulons were transformed to hg38 coordinates using liftOver (https://genome.ucsc.edu/cgi-bin/hgLiftOver). The imputed accessibility matrix and the liftovered signatures were used as input for AUCell (v.1.22.0) to assess eRegulon enrichment.

### Statistics and reproducibility analysis

No statistical methods were used to predetermine sample size. In the single-cell experiments, low-quality cells were removed for downstream analyses. The experiments were not randomized. The investigators were not blinded to allocation during experiments and outcome assessment. For the single-cell experiments we performed at least two experiments per technique (snRNA-seq, snATAC–seq or single-cell multiomics). For the smFISH experiment we performed three replicates. For the MPRA experiments we performed at least two replicates per condition (system (HepG2, AML12, mouse) and library (3′, 5′)). For the luciferase experiments, we performed four replicates per condition. All statistical analyses were performed using one or two-sided rank-sum Wilcoxon tests (nonparametric) or implemented in external algorithms (such as the binomial test used in GREAT, or the two-sided Wilcoxon test performed using Seurat's FindMarkers). Details such as the statistical tests, multiple-testing correction method, and experimental replicates are indicated in the figures or figure legends, and exact *P* values are provided in the source data.

### Reporting summary

Further information on research design is available in the Nature Portfolio Reporting Summary linked to this article.

### Data availability

Data generated in this study (scRNA-seq, scATAC–seq, single-cell multiome, TBX3 ChIP–seq, MPRAs and bulk ATAC–seq) are available at the Gene Expression Omnibus (GEO: GSE218472). Signatures for RAS signalling, WNT signalling pituitary response and hypoxia were obtained from a previous study[6]. scRNA-seq data of the mouse liver at different timepoints of the circadian rhythm[81] were downloaded from the GEO (GSE145197). ChIP–seq data for HNF4A, CEBPA, FOXA1 and ONECUT1[29] were downloaded from the European Nucleotide Archive (PRJEB1571). Hi-C data[28] were obtained from the GEO (GSE65126). Raw snRNA-seq data from male and female livers[88] were downloaded from the SRA Project PRJNA779049 (Vehicle_Female_liver: SAMN23009762; and Vehicle_Male_liver: SAMN23009760). Bulk RNA-seq data of HepG2, Hepa1-6 and AML12 cells were obtained from ENCODE and GEO (ENCFF790EGR, GSE167316, GSE146053). Data for MPRA positive controls were retrieved from ENCODE (ENCFF288HIT, ENCFF032RDN)[92], GEO (GSE71279)[91] and a previous study[60]. Saturation mutagenesis data were downloaded from https://mpra.gs.washington.edu/satMutMPRA/ and obtained from previous studies[41,42]. DeepExplainer plots for each of the wild-type zonated enhancers selected for the library design in Fig. 7 and Fig. 8 are available at FigShare (https://doi.org/10.6084/m9.figshare.24115986). Human liver snATAC–seq data[44] were downloaded from the GEO (GSE184462).

Processed data can be explored in Scope (http://scope.aertslab.org/#/Bravo_et_al_Liver; Supplementary Note) and the UCSC genome browser (https://genome.ucsc.edu/s/cbravo/Bravo_et_al_Liver; Supplementary Note). Source data for the Supplementary Figs. 1–7 are available at FigShare (https://doi.org/10.6084/m9.figshare.24532951). All other data supporting the findings of this study are available from the corresponding author on reasonable request. Source data are provided with this paper.

### Code availability

VSN-Pipelines (https://vsn-pipelines.readthedocs.io/), pycisTopic (https://pycistopic.readthedocs.io/), pycistarget (https://pycistarget.readthedocs.io/), SCENIC+ (https://scenicplus.readthedocs.io/), ScoMAP (https://github.com/aertslab/ScoMAP) are available online. Notebooks to reproduce the main figures are available at GitHub (https://github.com/aertslab/Bravo_et_al_Liver). DeepLiver is available at Zenodo (https://zenodo.org/record/8139953#) and is available at Kipoi (https://kipoi.org/docs/).

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

## Acknowledgements

Computing was performed at the Vlaams Supercomputer Center (VSC). This work is funded by the following grants to S.A.: ERC Consolidator Grant (724226_cis-CONTROL), ERC Proof of Concept (963884), Special Research Fund (BOF) KU Leuven (grant C14/18/092), Foundation Against Cancer (2020-1396) and FWO (grants G0B5619N and G094121N) and a PhD fellowship from the FWO to C.B.G.-B. (11F1519N). We thank the members of various groups that make curated PWMs publicly available, including T. Hughes, M. Bulyk, A. Mathelier, V. Makeev and many others; the staff at Resolve Biosciences, specially J. Aerts, for performing the molecular cartography experiments in the mouse liver; the staff at Janssen Pharmaceutica, VIB Tech Watch and the VIB single-cell accelerator for their help and funding for generating the mouse liver single-cell data; and the staff at the VIB FACS expertise center for their assistance during the FACS–MPRA and FACS–ATAC experiments. Supplementary Fig. 9a was created using BioRender (agreement number: BV24NAUKOX).

## Author contributions

C.B.G.-B., G. Halder and S.A. conceived the study. C.B.G.-B. performed the computational analyses, with assistance of I.I.T. and G. Hulselmans. I.M., V.C., L.S.-G., E.V. and S.P. performed the single-cell experiments. H.H., R.V., V.C., L.S.-G., J.D., N.P. and D.M. performed the MPRA experiments. V.C. performed the luciferase experiments. C.B.G.-B. and S.A. wrote the manuscript.

## Competing interests

The authors declare no competing interests.

## Additional information

**Extended data** is available for this paper at https://doi.org/10.1038/s41556-023-01316-4.

**Correspondence and requests for materials** should be addressed to Stein Aerts.

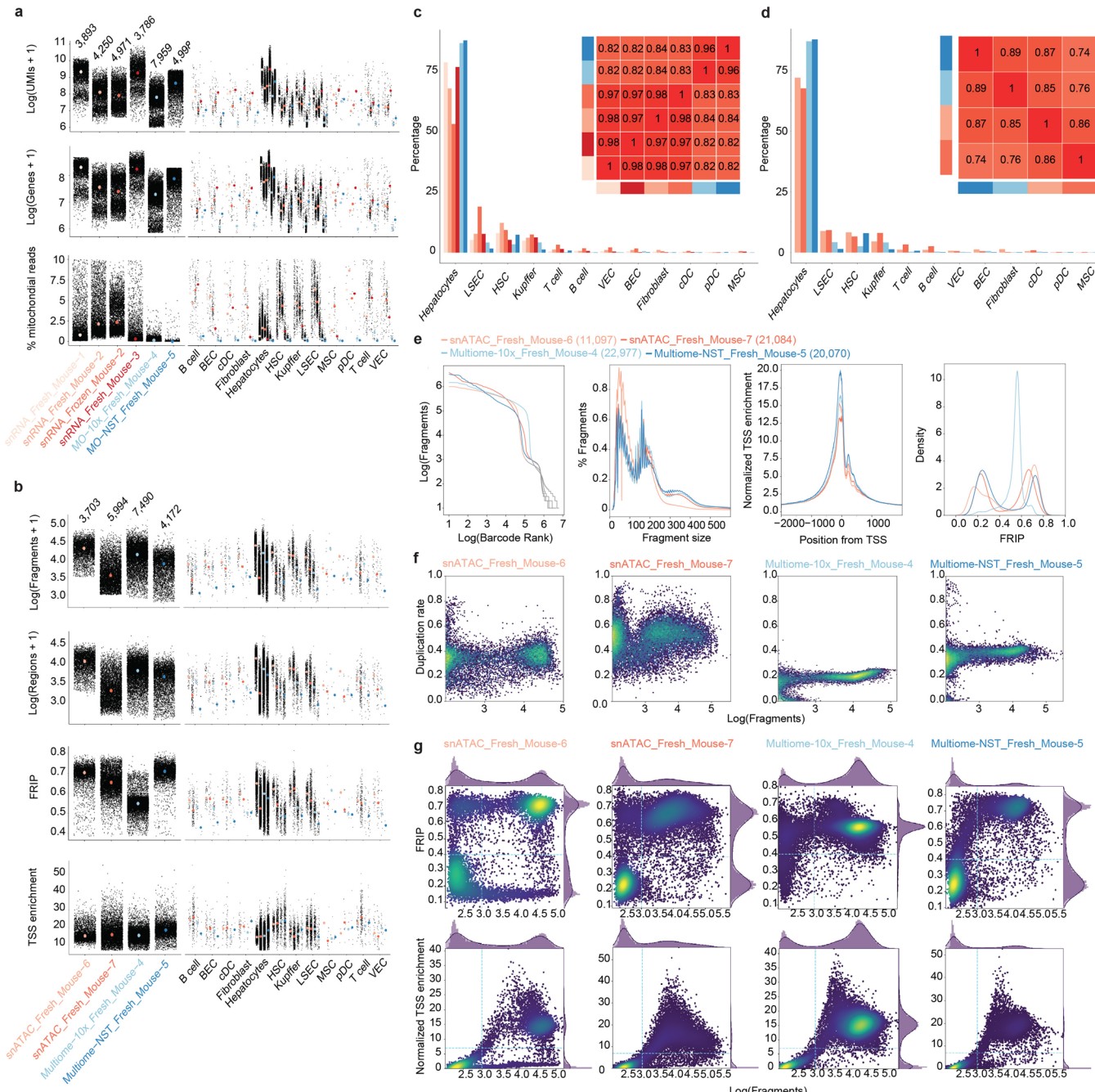

**Extended Data Fig. 1 | Single cell omics data quality control. a.** Dot plot showing the log number of UMIs (before normalization), the log number of expressed genes and the ratio of mitochondrial reads per sample and cell type for the snRNA-seq and multiome samples. The number of cells per sample is indicated on top of the dot plots. The median of each distribution is indicated with a coloured dot. **b.** Dot plot showing the $\log_{10}$(number of fragments), the $\log_{10}$(number of accessible regions), the Fraction of Reads in Peaks (FRIP), and the normalized TSS enrichment per sample and cell type for the snATAC-seq and multiome samples. The number of cells per sample is indicated on top of the dot plots. The median of each distribution is indicated with a coloured dot. **c.** Bar plot showing the percentage of cells corresponding to each cell type across samples and correlation between normalized gene expression values (as bulk) across samples for the snRNA-seq and multiome samples. **d.** Bar plot

showing the percentage of cells corresponding to each cell type across samples and correlation between normalized region accessibility values (as bulk) across samples for the snATAC-seq and multiome samples. **e.** Sample-level epigenome quality control, including (in order top to bottom): barcode rank plot, insert size distribution, TSS enrichment and Fraction of Reads In Peaks (FRIP). **f.** Duplication rate per barcode versus log number of fragments per sample. **g.** Fraction of Reads in Peaks (FRIP, top) and Normalized TSS enrichment (bottom) per barcode per sample. The blue dotted lines indicate the minimum threshold in FRIP, number of fragments and TSS enrichment to select high quality cells. BEC: biliary epithelial cells, cDC: conventional dendritic cell, HSC: hepatic stellate cells, MSC: mesothelial cells, pDC: plasmacytoid dendritic cell, VEC: vascular endothelial cells. Source numerical data are available in source data.

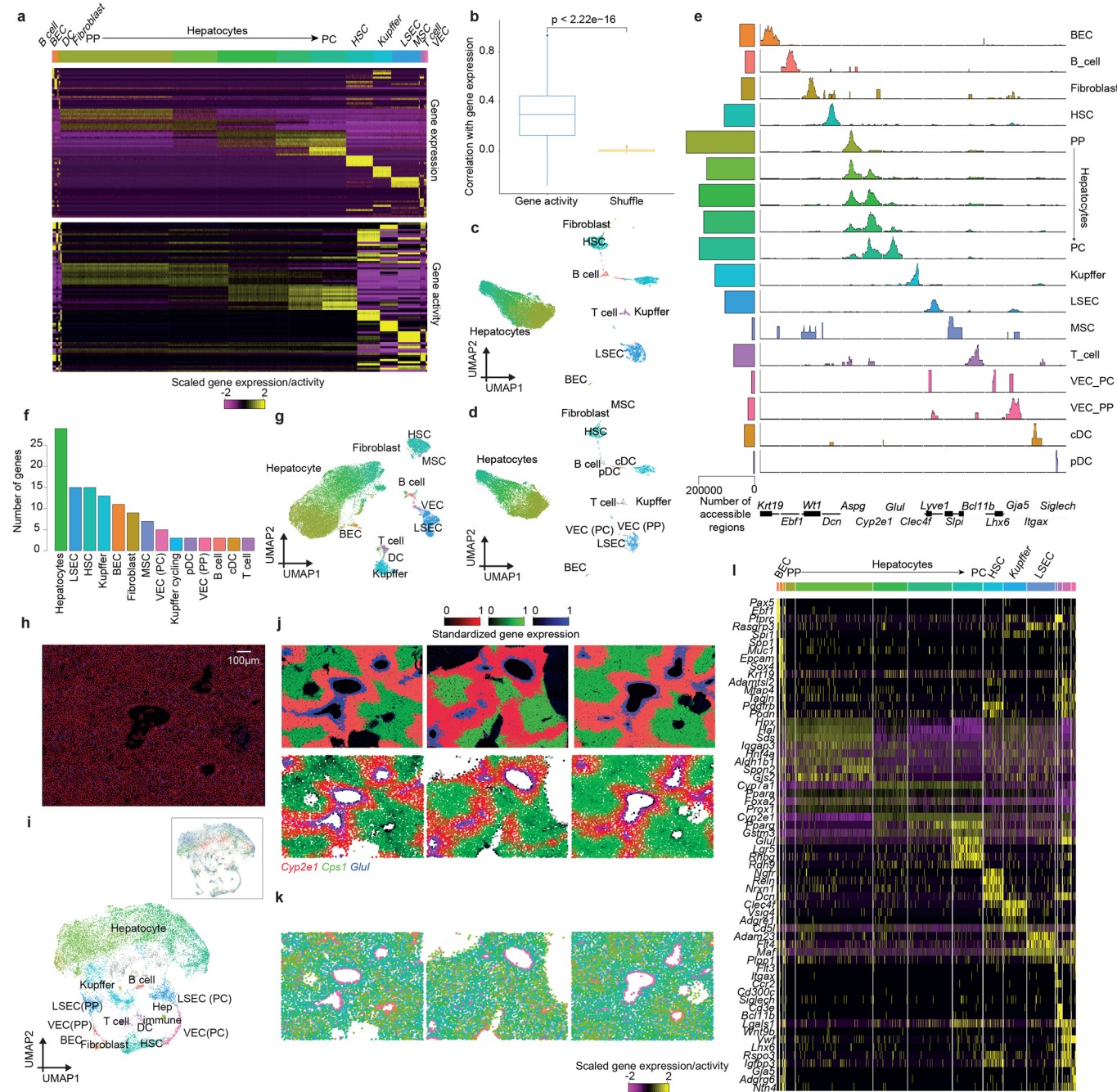

**Extended Data Fig. 2 | Validation of snATAC-seq topic model with gene activity and label transfer and Molecular cartography of the mouse liver.**
**a.** Scaled gene expression (top) and scaled gene activity inferred from the snATAC-seq layer (bottom). **b.** Correlation between gene activity and gene expression versus a random control (shuffled gene activity) across 1,141 marker genes (log₂[FC]> 1.5 and adjusted P < 0.05 in at least one cell type). The top/lower hinge represents the upper/lower quartile and whiskers extend from the hinge to the largest/smallest value no further than 1.5 × interquartile range from the hinge, respectively. The median is used as the centre. A one-sided rank-sum Wilcoxon test was performed to assess if the correlation based on gene activity was greater than random (P: < 10⁻¹⁶). Gene activity values are derived from 4 independent biological replicates. **c.** snATAC-seq UMAP (22,600 cells) coloured by the label transfer cell type annotation (from snRNA-seq cells to snATAC-seq cells). For the UMAP, cells from 4 biological replicates were combined. **d.** snATAC-seq UMAP (12,898 cells) coloured by the label given to the cell based on the snRNA-seq clustering. **e.** Pseudobulk accessibility profiles on representative Differentially Accessible Regions per cell type. The bar plot indicates the number

of peaks called by MACS in each pseudobulk. **f.** Number of selected genes per cell type within the Molecular Cartography gene panel (100 genes). **g.** snRNA-seq (29,798 cells) UMAP based on the selected 100 genes. **h.** Cell segmentation using the DAPI signal on the sample with QuPath. Three independent experiments were performed, with similar results. **i.** UMAP (15,522 spots/cells) of the segmented nuclei based on the number of transcripts measured per gene per spot coloured by their assigned cell type. Within the rectangle, UMAP (15,522 spots/cells) of the segmented nuclei based on the number of transcripts measured per spot coloured by their sample of origin. For the UMAP, cells/spots from 3 biological replicates were combined. **j.** Molecular Cartography maps (3 replicates, 15,522 spots) coloured by aggregated gene expression using RGB encoding. **k.** Molecular Cartography maps (3 replicates, 15,522 spots) coloured by assigned cell type. **l.** Heat map showing the scaled gene expression across cells (grouped by cell type) of selected genes. BEC: biliary epithelial cells, cDC: conventional dendritic cell, HSC: hepatic stellate cells, MSC: mesothelial cells, PC: pericentral, pDC: plasmacytoid dendritic cell, PP: periportal, VEC: vascular endothelial cells. Source numerical data are available in source data.

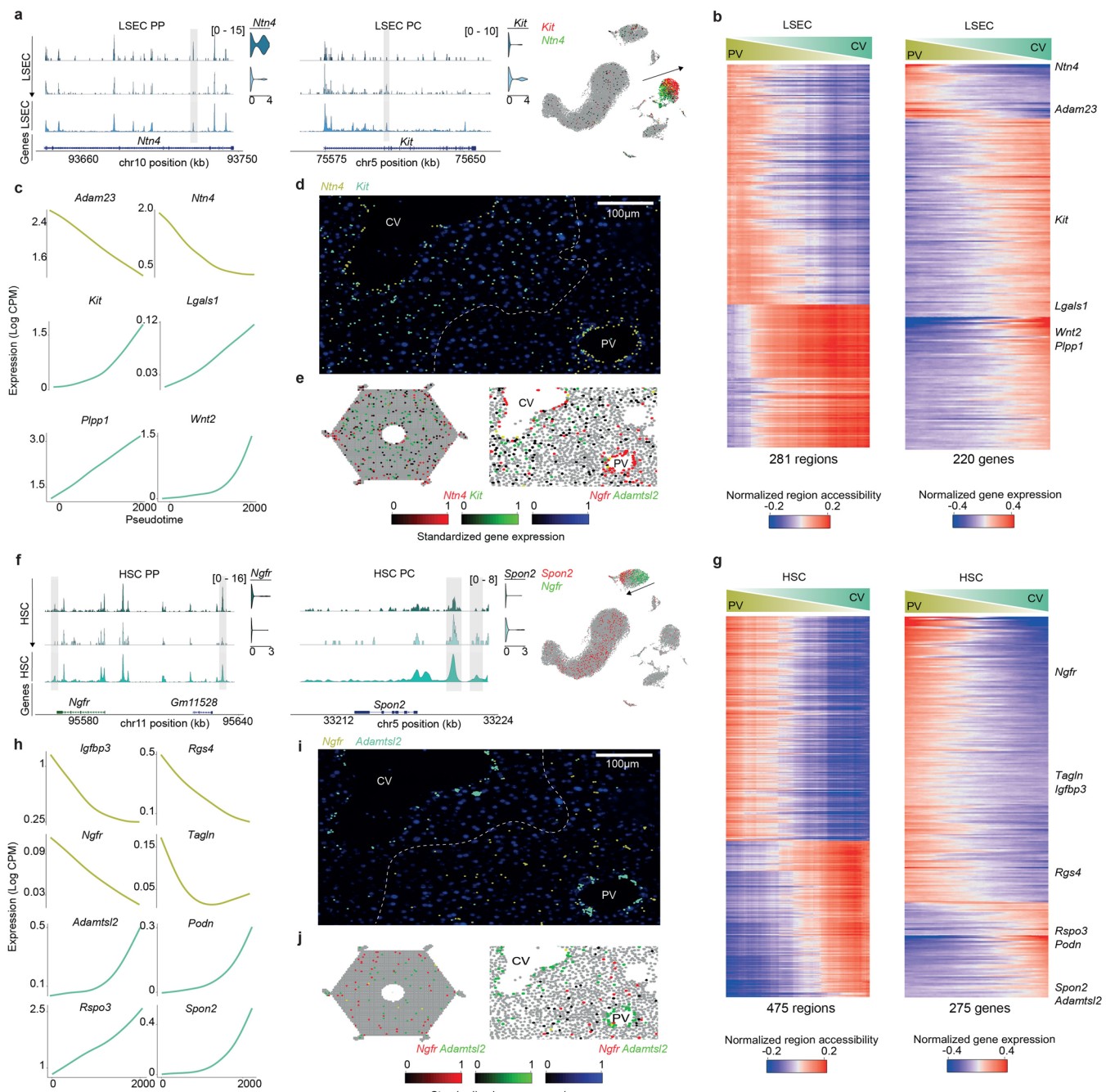

**Extended Data Fig. 3 | Zonation of Liver Sinusoidal Endothelial Cells (LSEC) and Hepatocellular Stellate Cells (HSC). a.** Pseudobulk chromatin profiles at different gene loci for LSEC zonation states, accompanied by violin plots representing the normalized gene expression of the relevant gene in each class. UMAPs show the gene expression of the relevant genes with RGB encoding. **b.** Normalized region accessibility and gene expression zonation heat maps. LSECs are ordered by pseudotime (from periportal to pericentral) and regions and genes affected by zonation are shown (281 regions and 220 genes). **c.** GAM fitted gene expression profiles for selected genes along the zonation pseudotime for LSECs. **d.** Liver section image showing smFISH profiles for *Ntn4* (PP LSEC marker) and *Kit* (PC LSEC marker). **e.** ScoMAP liver lobule and smFISH coloured by gene expression using RGB encoding. **f.** Pseudobulk chromatin profiles at different gene loci for HSC zonation states, accompanied by violin plots

representing the normalized gene expression of the relevant gene in each class. UMAPs show the gene expression of the relevant genes with RGB encoding. **g.** Normalized region accessibility and gene expression zonation heat maps. HSCs are ordered by pseudotime (from periportal to pericentral) and regions and genes affected by zonation are shown (475 regions and 275 genes). **h.** GAM fitted gene expression profiles for selected genes along the zonation pseudotime for HSCs. **i.** Liver section image showing smFISH profiles for *Ntgr* (HSC PP marker) and *Adamtsl2* (PC HSC marker). **j.** ScoMAP liver lobule and smFISH coloured by gene expression using RGB encoding. For the transcriptome and epigenome data, cells from 5 and 4 biological replicates were combined, respectively. For i and h, three independent experiments were performed, with similar results. HSC: hepatic stellate cells, LSEC: liver sinusoidal endothelial cells, PC: pericentral, PP: periportal. Source numerical data are available in source data.

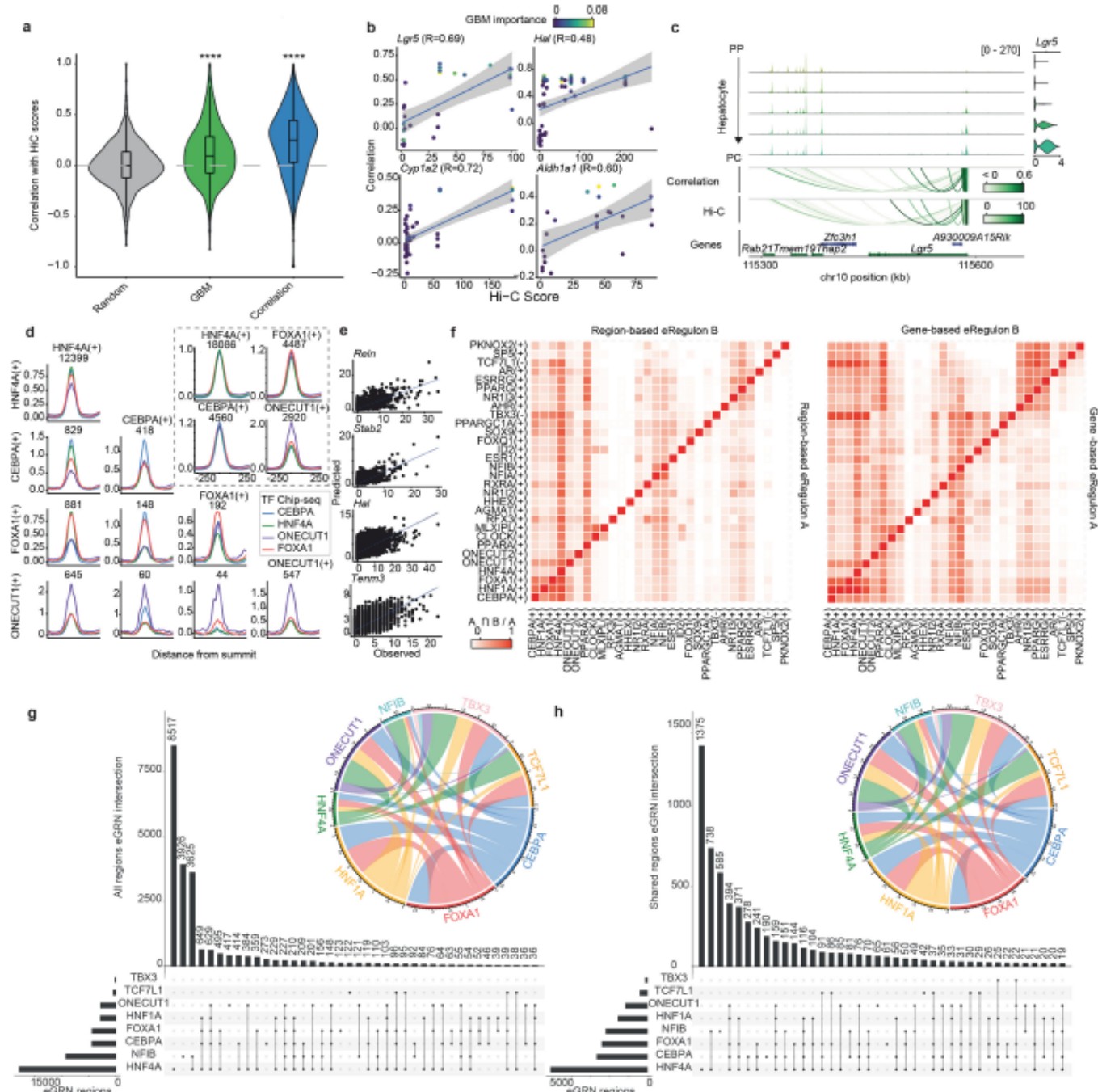

**Extended Data Fig. 4 | Validation of SCENIC+ regulons. a.** Violin plot showing the distribution of the correlations between SCENIC+ predicted region to gene links per gene (by the Gradient Boosting Machine (GBM) and correlation methods) with Hi-C scores (between the same regions and the Transcription Start Site (TSS) of the linked gene) for 559 hepatocyte markers genes ($\log_2[FC] > 1$ and adjusted p-value < 0.05). The random control distribution consists of shuffled correlation values. In the boxplots, the top/lower hinge represents the upper/lower quartile and whiskers extend from the hinge to the largest/smallest value no further than 1.5 × interquartile range from the hinge, respectively. The median is used as the centre. SCENIC+ was trained using transcriptome and epigenome data from 5 and 4 biological replicates, respectively. **b.** Examples showing the correlation between SCENIC+ region to gene links correlation scores and Hi-C scores, coloured by their GBM importance. The blue line represents the fitted linear regression line and the grey bands represent the 95% confidence interval

bands. **c.** Example on the *Lgr5* locus depicting chromatin accessibility profiles and gene expression across hepatocyte subpopulations and the region to gene correlation and Hi-C scores. For the transcriptome and epigenome data, cells from 5 and 4 biological replicates were combined, respectively. **d.** ChIP-seq coverage profiles for HNF4A, CEBPA, FOXA1 and ONECUT1 on their unique and shared predicted regulon regions. **e.** Example showing the correlation between observed gene expression values (on left-out-data) and gene expression predicted using a GBM model (per gene) trained using the expression of the predicted TF regulators as features. **f.** Heat maps showing the overlap between selected region-based and gene-based regulons. **g.** Overlap between all regions included in selected regulons. **h.** Overlap between core regions (that is accessible across all mice) included in selected regulons. Source numerical data are available in source data.

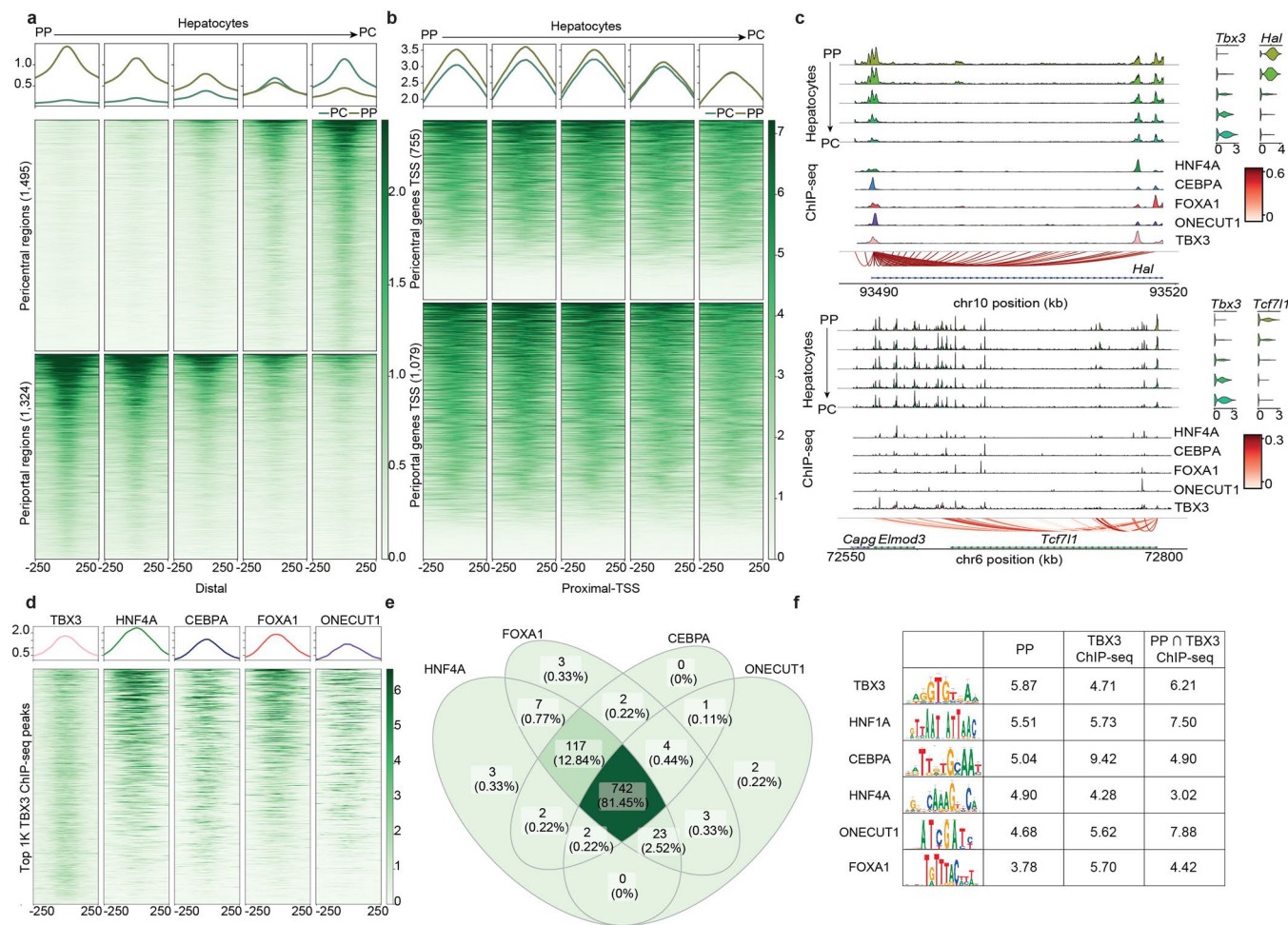

**Extended Data Fig. 5 | Chromatin accessibility profiles of regulatory regions of zonated genes inferred with SCENIC+ and at their TSS and validation and intersection of TBX3 and other hepatocyte TFs in periportal regions.** **a**, **b**. Coverage plots on zonated pericentral and periportal regions (**a**.) and at the Transcription Start Site (TSS) of zonated genes linked to these regions (**b**.) on mouse hepatocytes snATAC-seq pseudobulk, ordered by cluster from periportal to pericentral. **c**. Pseudobulk accessibility profiles, ChIP-seq coverage (for HNF4A, CEBPA, FOXA1, ONECUT1 and TBX3), SCENIC+ region to gene links coloured by correlation score and gene and *Tbx3* expression across the zonated hepatocytes classes (from periportal (PP) to pericentral (PC)) are shown. The gene loci showed are *Hal* and *Tcf7l1*. For the transcriptome and epigenome data, cells from 5 and 4 biological replicates were combined, respectively. **d**. Coverage plot on the top 1,000 TBX3 ChIP-seq regions on HNF4A, CEBPA, FOXA1, ONECUT1, and TBX3 ChIP-seq data. **e**. Overlap between HNF4A, FOXA1, CEBPA and ONECUT1 regions that overlap the top 1,000 Tbx3 regions. **f**. Cistarget motif enrichment in periportal and TBX3 ChIP-seq regions and their overlap. Values indicate the cisTarget Normalized Enrichment Scores (NES) in the different regions sets for the indicated motifs (top motifs found in periportal regions).

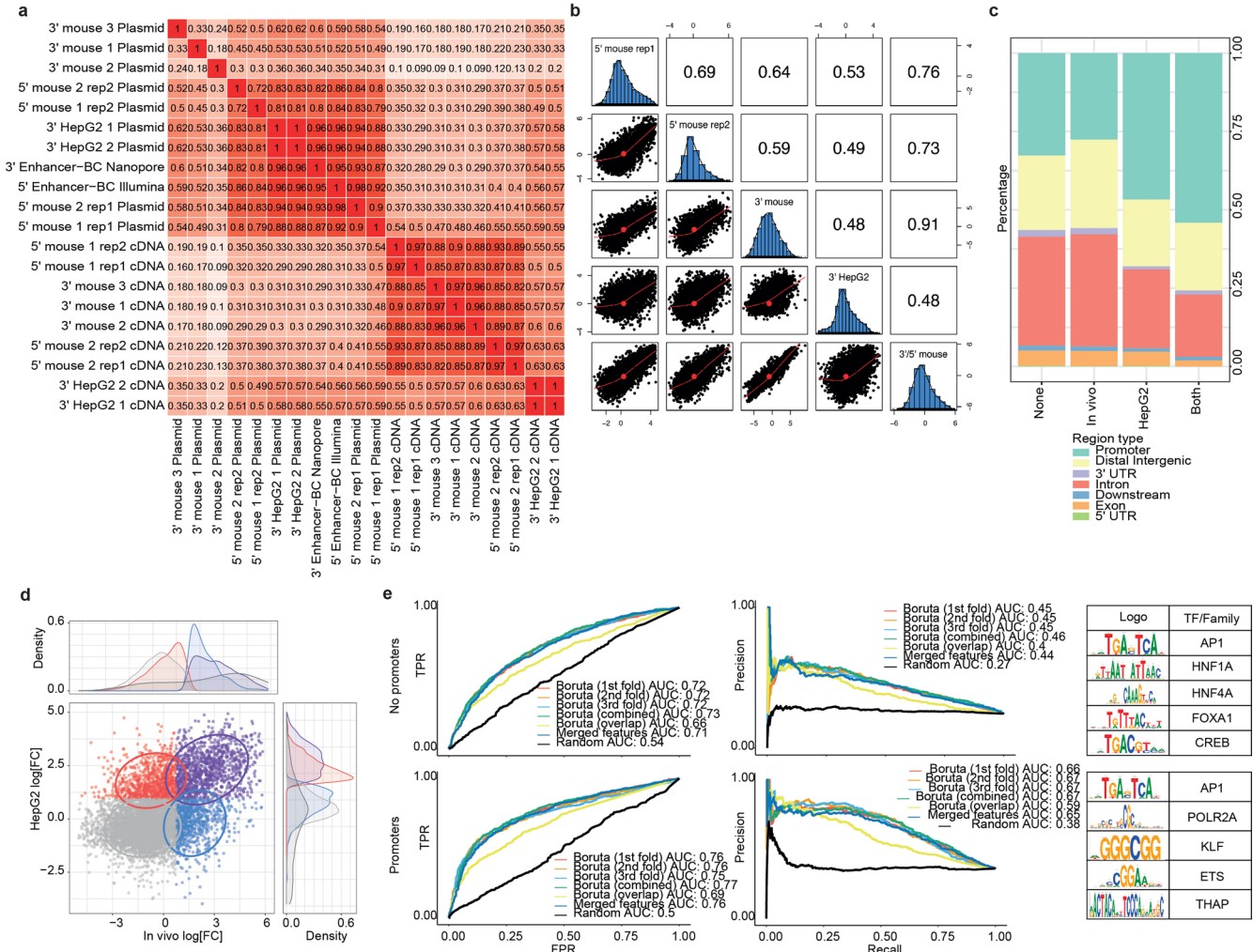

**Extended Data Fig. 6 | Random Forest models allow to identify sequence features driving enhancer activity. a.** Correlation between plasmid and cDNA measurements across MPRA experiments in the mouse liver and HepG2. Both 5' experiments were performed in vivo, while one of the 3' experiments was perfomed in vivo and the other in HepG2, respectively. **b.** Correlation of the MPRA log₂[FC] values per enhancer across experiments. **c.** Proportion of enhancer classes based on genomic annotation per high confidence activity class. None: Not active (n = 4,285), In vivo: Active only in vivo (n = 806), HepG2: Active only in HepG2 (n = 921), Both: Active in HepG2 and in vivo (n = 1,186). **d.** Correlation between log₂[FC] for high confidence enhancers (n = 7,198) in Hepg2 and in vivo coloured by enhancer type, with data ellipses per activity

group: Not active (grey), active in HepG2 (red), active in vivo (blue), active both *in vivo* and HepG2 (purple). **e.** Receiver operating characteristic and precision-recall curves for the trained activity models (with and without promoters). Boruta was run using 3-fold validation, and models were trained per fold, using all features found in at least one fold or only overlapping features. Merged features are derived by using all features found in at least one fold and merged based on their CRM score correlation and motif similarity. Top selected features per model (ordered by importance) are shown on the table on the right. For the MPRA experiments, 2 and 7 biological replicates were used in vitro and in vivo, respectively. Source numerical data are available in source data.

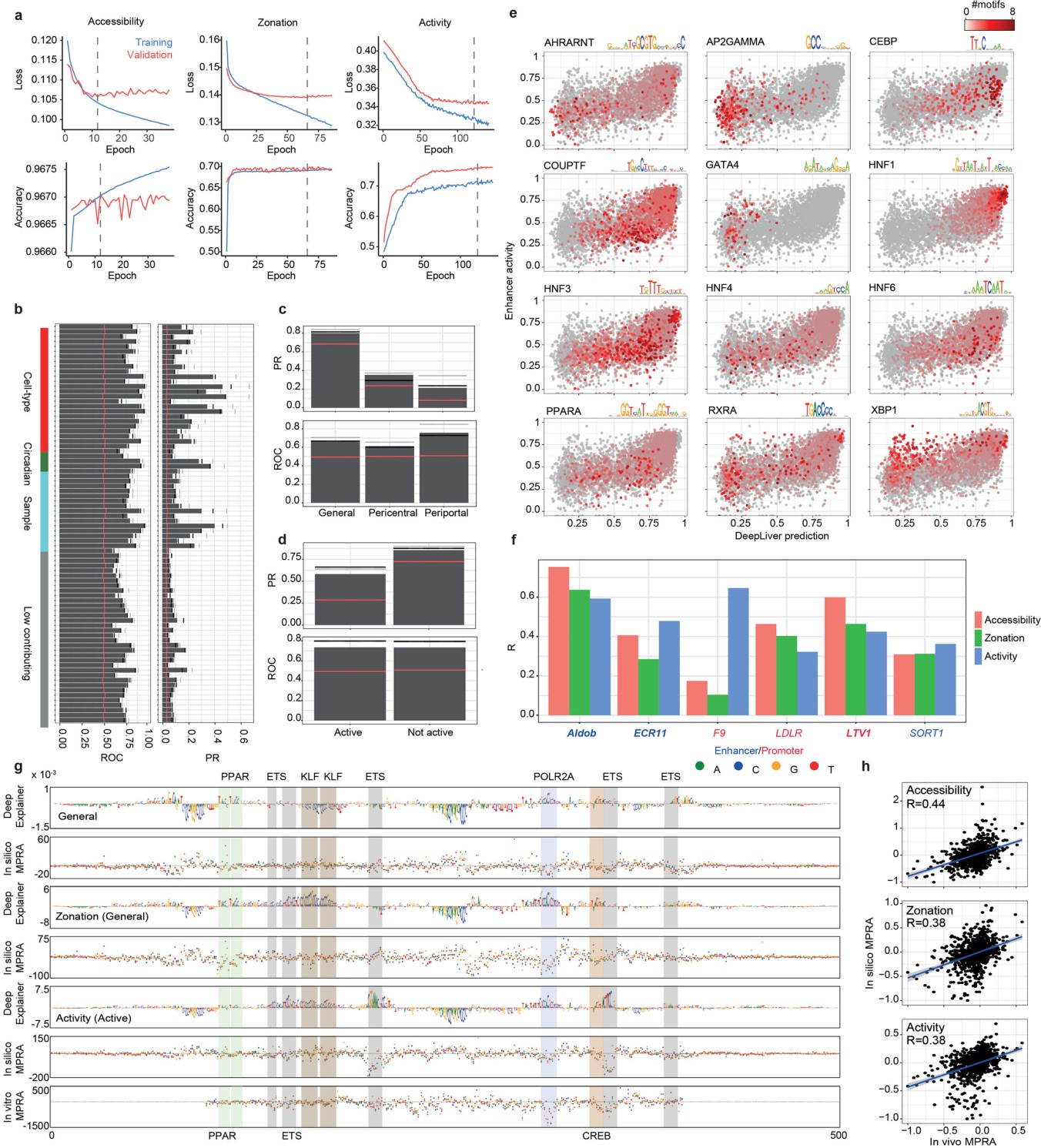

**Extended Data Fig. 7 | Overview of DeepLiver models and their predictability on previously tested enhancers. a.** Loss and accuracy curves for the DeepLiver models. The grey dashed lines indicate the selected epochs per model. **b.** ROC (Receiver Operating Characteristic) and PR (Precision-Recall) values on test data per topic for the DeepLiver accessibility model. The red line shows the values for a random classifier; the grey line, on the training data; and the black line on the validation data. **c.** ROC and PR values per topic for the DeepLiver zonation model. The red line shows the values for a random classifier; the grey line, on the training data; and the black line on the validation data. **d.** ROC and PR values per topic for the DeepLiver activity model. The red line shows the values for a random classifier; the grey line, on the training data; and the black line on the validation data. **e.** Correlation plot between Smith *et al.*[38] enhancer activity and DeepLiver activity predictions (n = 4,966) coloured by the number of motif instances in the sequences (red scale). **f.** Correlation between in silico and experimental saturation mutagenesis for different sequences tested in vivo by Patwardhan et al.[41] (*AldoB*, *ECR11* and *LTV1*) and in HepG2 by Kircher et al.[42] (*F9*, *LDLR* and *SORT1*). **g.** DeepExplainer and saturation mutagenesis plots for the accessibility, zonation and activity models on the *LTV1* promoter (mm9: chr7:29161343-29161843), with motifs highlighted. Saturation mutagenesis, shown below, was performed in this enhancer by Patwardhan et al.[41]. **h.** Correlation between DeepLiver in silico mutagenesis and experimental saturation mutagenesis in the *LTV1* promoter. The blue line represents the fitted linear regression and the grey bands represent the 95% confidence interval bands. Source numerical data are available in source data.

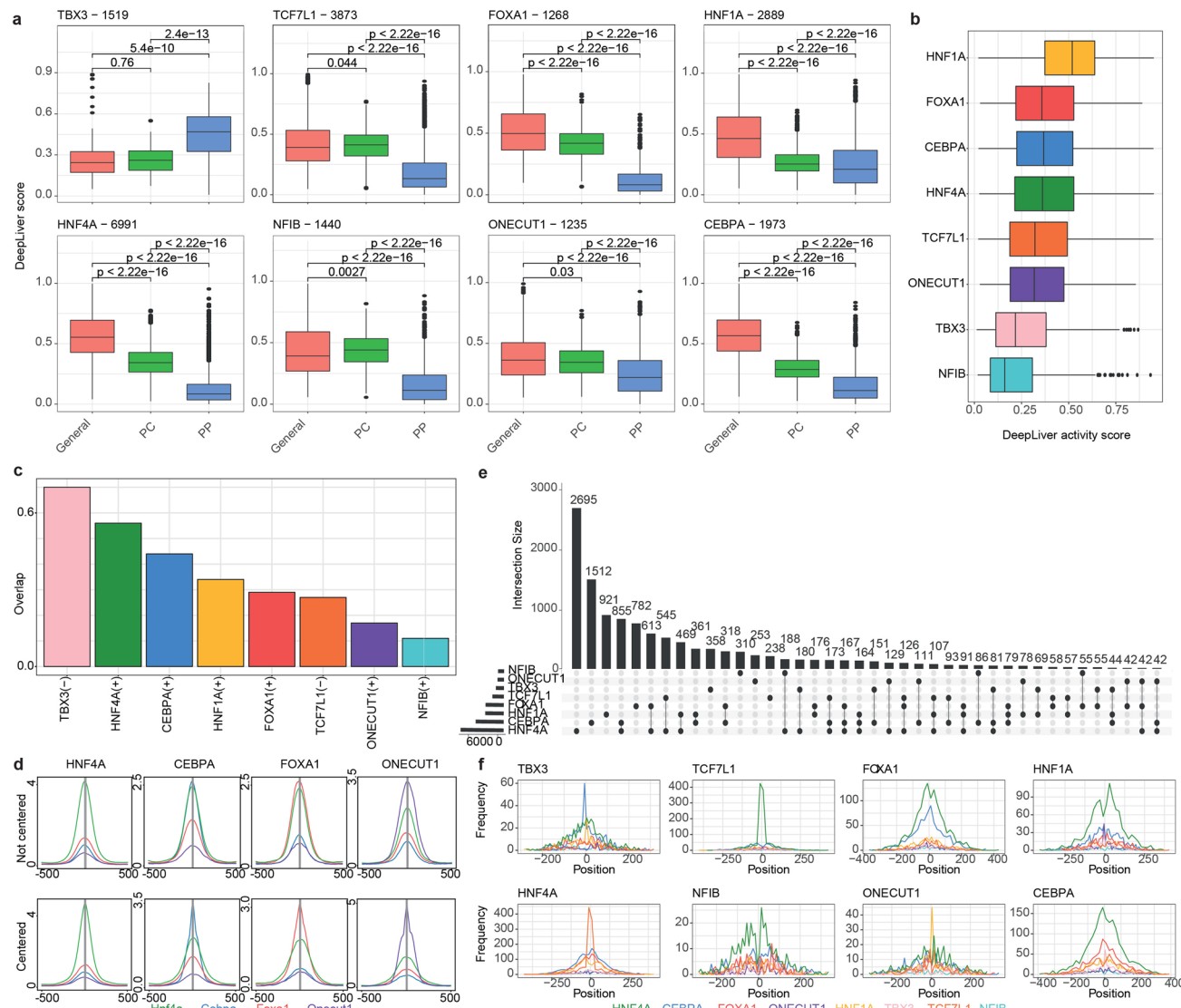

**Extended Data Fig. 8 | DeepLiver predictions in SCENIC+ regulons and validation of predicted target sites. a.** DeepLiver zonation predictions on DeepLiver predicted target regions for different transcription factors. The number of target regions identified per transcription factors is indicated in each plot. Two-sided rank-sum Wilcoxon tests were performed to compare the distributions between the groups. Bonferroni adjusted p-values are reported on top. DeepLiver was trained on data from 4 biological replicates. **b.** DeepLiver activity predictions on DeepLiver predicted target regions for different transcription factors. **c.** Percentage of TF target regions predicted by SCENIC+ found by DeepLiver. **d.** ChIP-seq coverage on TF target regions predicted by DeepLiver, without centreing (that is ATAC peak coordinates) or centering on the predicted binding site. **e.** Overlap between target regions predicted by DeepLiver for different transcription factors. **f.** Distances between binding sites for a TF and binding sites of other TFs in overlapping regions. In the boxplots in a and b, the top/lower hinge represents the upper/lower quartile and whiskers extend from the hinge to the largest/smallest value no further than 1.5 × interquartile range from the hinge, respectively. The median is used as the centre. Source numerical data are available in source data.

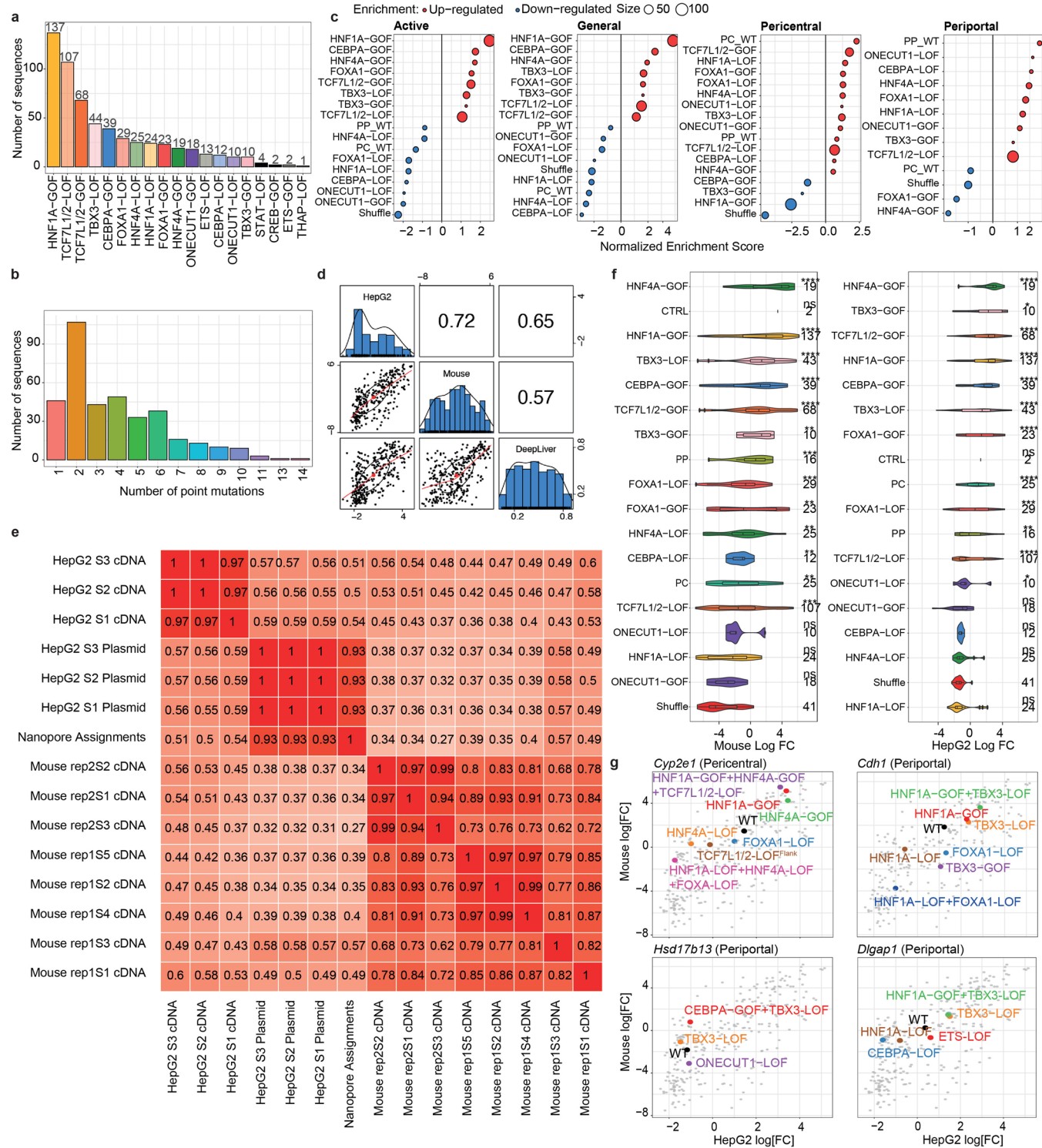

**Extended Data Fig. 9 | Library design and validation on wild-type zonated sequences and their activity and zonation variants. a.** Number of sequences containing each variant type. **b.** Number of point mutations across library sequences. **c.** Normalized Enrichment Scores (NES) for variant types along the predicted DeepLiver scores for activity and zonation (general, pericentral and periportal). **d.** Correlation between MPRA log₂[FC] in vivo and in HepG2 and DeepLiver activity predictions. **e.** Correlation between number of counts per enhancer across samples and replicates. **f.** MPRA log₂[FC] per variant type in vivo and in HepG2. In the boxplots, the top/lower hinge represents the upper/lower quartile and whiskers extend from the hinge to the largest/smallest value no

further than 1.5 × interquartile range from the hinge, respectively. The median is used as the centre. One-sided rank-sum Wilcoxon tests were performed to assess if the log₂[FC] values of each group were greater than those of the shuffled regions. The asterisks represent the Bonferroni adjusted p-values of the comparisons.****, $P <= 0.0001$; ***, $P <= 0.001$; **, $P <= 0.01$; *, $P <= 0.05$; ns, $P > 0.05$. **g.** In vivo MPRA log₂[FC] versus DeepLiver activity score with highlighted sequence variants for each enhancer. For the MPRA experiments, 3 and 8 biological replicates were performed in HepG2 and in vivo, respectively. GOF: Gain-Of-Function, LOF: Loss-of-function. Source numerical data are available in source data.

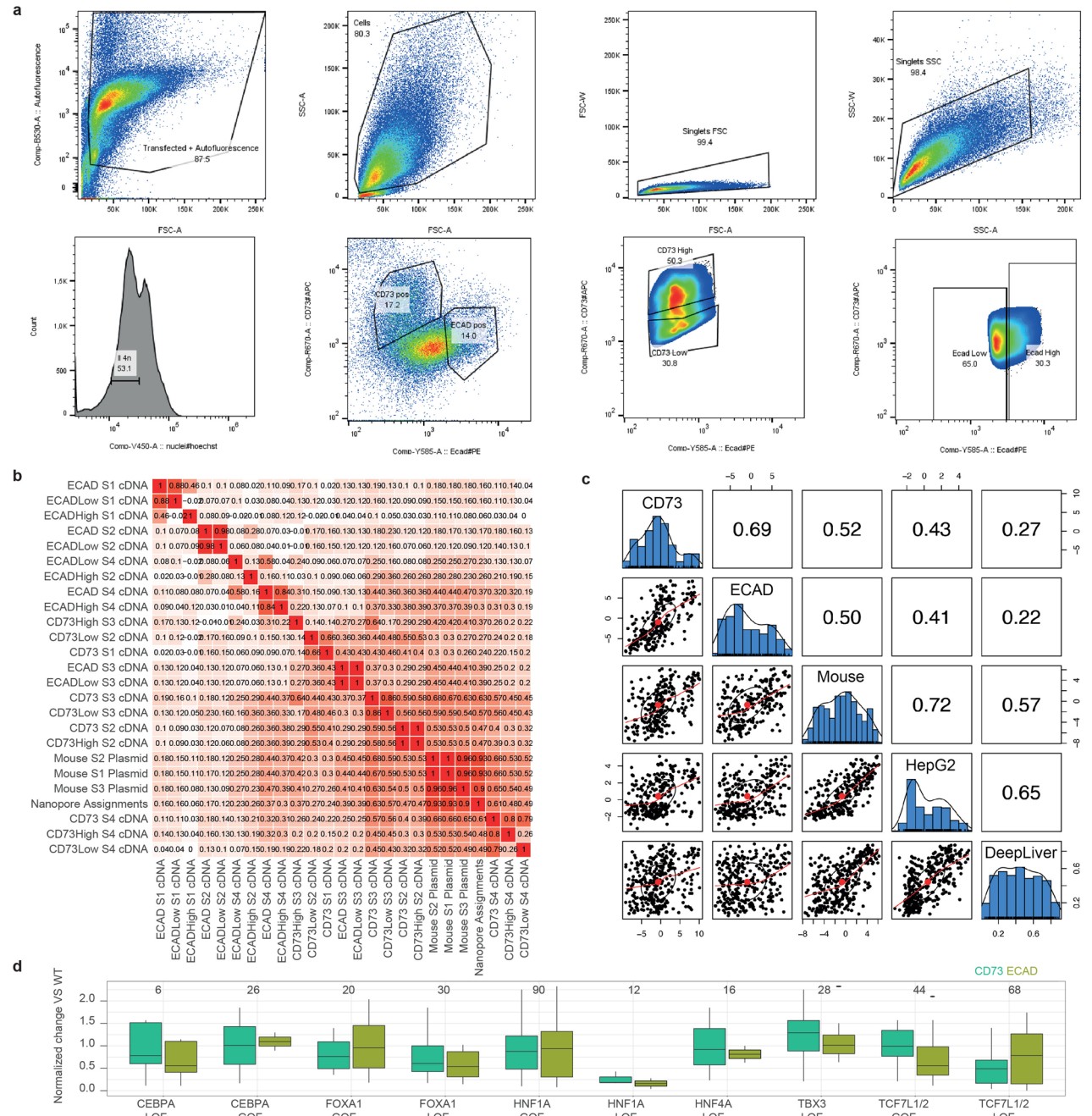

**Extended Data Fig. 10 | FACS MPRA reveals activity differences between pericentral and periportal enhancers. a**. FACS gating strategy. FSC-A and SSC-A were used for hepatocytes size selection. For the FACS MPRA experiment, cells containing the library were selected based on GFP. For the FACS ATAC experiment, viable cells were selected using the Zombie Green Viability kit. Tetraploid hepatocytes were selected based on Hoechst stain. CD73 and ECAD were used to select hepatocytes bins along the porto-central axis. **b**. Correlation between number of counts per enhancer across samples and replicates. **c**. Correlation between CHEQ-seq log$_2$[FC] in the CD73$^+$ and ECAD$^+$ fractions,

*in vivo* bulk and in HepG2 and DeepLiver activity predictions. **d**. Normalized change between enhancers (grouped by variant type) in the CD73$^+$ and ECAD$^+$ fractions. The number of enhancers used in each comparison is indicated over each mutational group. The top/lower hinge represents the upper/lower quartile and whiskers extend from the hinge to the largest/smallest value no further than 1.5 × interquartile range from the hinge, respectively. The median is used as the centre. Four biological replicates were used. GOF: Gain-Of-Function, LOF: Loss-of-function. PC: Pericentral, PP: Periportal, WT: Wild-type. Source numerical data are available in source data.

| | |
|---|---|

# Reporting Summary

## Statistics

For all statistical analyses, confirm that the following items are present in the figure legend, table legend, main text, or Methods section.

| n/a | Confirmed | |
|---|---|---|
| ☐ | ☒ | The exact sample size (*n*) for each experimental group/condition, given as a discrete number and unit of measurement |
| ☐ | ☒ | A statement on whether measurements were taken from distinct samples or whether the same sample was measured repeatedly |
| ☐ | ☒ | The statistical test(s) used AND whether they are one- or two-sided<br>*Only common tests should be described solely by name; describe more complex techniques in the Methods section.* |
| ☐ | ☒ | A description of all covariates tested |
| ☐ | ☒ | A description of any assumptions or corrections, such as tests of normality and adjustment for multiple comparisons |
| ☐ | ☒ | A full description of the statistical parameters including central tendency (e.g. means) or other basic estimates (e.g. regression coefficient) AND variation (e.g. standard deviation) or associated estimates of uncertainty (e.g. confidence intervals) |
| ☐ | ☒ | For null hypothesis testing, the test statistic (e.g. *F*, *t*, *r*) with confidence intervals, effect sizes, degrees of freedom and *P* value noted<br>*Give P values as exact values whenever suitable.* |
| ☐ | ☒ | For Bayesian analysis, information on the choice of priors and Markov chain Monte Carlo settings |
| ☒ | ☐ | For hierarchical and complex designs, identification of the appropriate level for tests and full reporting of outcomes |
| ☐ | ☒ | Estimates of effect sizes (e.g. Cohen's *d*, Pearson's *r*), indicating how they were calculated |

*Our web collection on statistics for biologists contains articles on many of the points above.*

## Software and code

Policy information about availability of computer code

| Data collection | 1. Transcriptome analysis: The 10x snRNA-seq fatsq files were processed with cellranger (v1.0.0) count function. Reads were aligned to a pre-mRNA Mus musculus reference genome (mm10), that listed each gene transcript locus as an exon, and included intronic reads in the counting (10x Genomics, see https://support.10xgenomics.com/single-cell-gene-expression/software/pipelines/latest/advanced/references#premrna). The 10x multiome fastq files were processed with cellranger-arc (v2.0.0) count function, with include introns =True option. Reads were aligned to the Mus musculus reference genome (ata-cellranger-arc-mm10-2020-A-2.0.0). 10x snRNA-seq and 10x multiome (gene expression) runs were analyzed first independently using VSN-pipelines (v0.27.0). Briefly, cells with at least 350 genes expressed and a percentage of mitochondrial reads below 10% were kept. Scanpy (v1.8.2) was run with default parameters, using the number of principal components automatically selected by VSN-Pipelines and using Leiden clustering with resolutions 0.4, 0.6 and 0.8. Hepatocyte clusters with low gene expression and high percentage of mitochondrial reads were removed, as well as doublets called with Scrublet (v0.2.3). The samples were merged, obtaining 29,798 high-quality cells, and reanalyzed with VSN-Pipelines. To correct for batch effects, we used Harmony on the selected principal components (34), using Leiden clustering with resolution 0.6, resulting in 15 clusters. The VEC and DC subpopulations were identified according to marker genes. This resulted in the identification of 14 cell types.<br>2. Epigenome analysis: 10x scATAC-seq samples were processed with cisTopic (v0.3.0), using the cells called by cellRanger (v1.2.0, 5,628 cells) and mm10 SCREEN regions (1,212,823 regions). For topic modelling, we used WarpLDA with default parameters, using 500 iterations and inferring models with 2, 5, 10 to 30 (by a step of 1), 35, 40, 45 and 50. This resulted in a model with 19 topics. After correcting sample effects with harmony (v1.0, applied on the scaled topic distributions), we performed Leiden clustering with resolution 0.6, obtaining 11 clusters. Gene activity was calculated by aggregating the probabilities of regions +/-10kb from the TSS (including the gene body). Cluster annotation was done based on motif enrichment, gene activity and label transfer from the annotated transcriptome with Seurat (v4.0.3, ,using cisTopic's gene activity matrix, cca as reduction and the first 10 dimensions). The labelled 10x scATAC-seq and multiome cells (annotated based on the transcriptome labels) and the scATAC-seq fragments were used as input for pycisTopic (v1.0.1.dev75+g3d3b721). Briefly, we first created pseudobulks per cell type and performed peak calling using MACS2 (v2.2.7.1, with –format BEDPE –keep-dup all –shift 73 –ext_size 146 as |

parameters, as recommended for single-cell ATAC-seq data). To derive a set of consensus peaks, we used the iterative overlap peak merging procedure describe in Corces et al. (2018). First, each summit is extended a 'peak_half_width' (by default, 250bp) in each direction and then we iteratively filter out less significant peaks that overlap with a more significant one. During this procedure peaks are merged and depending on the number of peaks included into them, different processes will happen: 1) 1 peak: The original peak will be kept, 2) 2 peaks: The original peak region with the highest score will be kept and 3) 3 or more peaks: The original region with the most significant score will be taken, and all the original peak regions in this merged peak region that overlap with the significant peak region will be removed. The process is repeated with the next most significant peak (if it was not removed already) until all peaks are processed. This procedure will happen twice, first in each pseudobulk peaks, and after peak score normalization to process all peaks together. This resulted in 486,888 regions. We further filtered the data set based on the scATAC-seq quality as well, keeping cells with at least 1,000 fragments, FRiP > 0.4 and TSS enrichment > 7, resulting in 22,600 high-quality cells. Topic modelling was performed using Mallet (v2.0), using 500 iterations and models with 2 topics and from 5 to 100 by an increase of 5. Additional models between 75 and 85 (by an increase of 1) were added as we observed that the best model should be on that area based on the model selection metrics, and we selected a model with 82 topics. Batch effects between samples were corrected using harmonypy (v0.0.6) on the scaled topic distributions, and Leiden clustering with resolution 0.6 resulted in 11 clusters, corresponding to 14 cell types based on previous labelling. Drop-out imputation was performed by multiplying the region-topic and topic-cell probabilities. The imputed accessibility matrix was multiplied by 10^6. Differentially Accessible Regions (DARs) were calculated between all cell populations and specifically within hepatocytes, HSC and LSEC subgroups, using default parameters and topics were binarized using Otsu thresholding. Hepatocyte DARs and shared hepatocyte topics were curated by performing hierarchical clustering on the pseudobulk probabilities, removing a small fraction of lowly and generally accessible regions, and defining non-overlapping groups between the different gradient groups. Gene Ontology analysis was performed using GREAT (v4). We additionally run MACS2 (v2.2.7.1) bdgdiff between hepatocytes, LSEC and HSC zonated state using default parameters. The number of shared regions across mice was calculated as the regions in the shared curated topics. PycisTarget (v1.0.1.dev42+gb6707ee) was run using a custom database with the consensus regions, on DARs, binarized topics (with Otsu thresholding), curated DARs and topics and MACS2 (v2.2.7.1) bdgdiff, with and without promoters, and using pycisTarget and DEM.

3. Multiome analysis: The gene expression matrix, the imputed accessibility from pycisTopic and the motif enrichment results were used as input for SCENIC+ (v 0.1.dev411+gf4bcae5.d20220810), using only the multiome cells for eGRN inference. SCENIC+ was run with default parameters, on the complete data set and only using hepatocytes, using http://nov2020.archive.ensembl.org/ as Biomart host. Briefly, a search space of a maximum between either the boundary of the closest gene or 150 kb and a minimum of 1 kb upstream of the TSS or downstream of the end of the gene was considered for calculating region-to-gene relationships using gradient boosting machine regression. TF-to-gene relationships were calculated using gradient boosting machine regression between all TFs and all genes. Final eRegulons were constructed using the GSEA approach in which region-to-gene relationships were binarized based on gradient boosting machine regression importance scores using the 85th, 90th and 95th quantile; the top 5, 10 and 15 regions per gene and using the BASC method for binarization. Regulons between the two runs (with all cells and only hepatocytes) were merged. Gene-based and region-based regulons were scored in the relevant data sets (multiome, all scRNA-seq and scATAC-seq and spatial templates) using AUCell (v1.22.0). Regulons with positive region-to-gene relationships, at least 20 target genes and a correlation between gene-based and region-based AUC scores above 0.4 were kept, obtaining 180 high quality regulons.

4. smFISH data analysis: The completely automated imaging process per round (including water immersion generation and precise relocation of regions to image in all three dimensions) was realized by a custom python script using the scripting API of the Zeiss ZEN software (Open application development, v2023.02.27). Final image analysis was performed in ImageJ (v2.3.0/1.53f) using the Polylux tool plugin (v1.6.1) from Resolve BioSciences to examine specific Molecular CartographyTM signals. Nuclei segmentation was performed using QuPATH (v4.2.1) based on the DAPI signal, setting pixel size to 0.25, minimum area to 10, maximum area to 400, sigma to 1.7 and cell expansion 8. Data was analyzed using Seurat. Using 14 PCs, we performed Leiden clustering, resulting in 19 clusters that corresponded to 11 cell types, that were annotated based on marker gene expression.

5. Hi-C and ChIP-seq data analysis: To validate regulons, we used publicly available Hi-C and ChIP-seq data, and generated new Tbx3 ChIP-seq data. Briefly, the Hi-C data was processed using Juicer (v1.9.9), extracting values using KR for normalization by 5kb windows, and keeping only links with score > 10 and involving a bin that overlaps at least one of the consensus peaks and a TSS (+/-1000bp), resulting in 890,488 region-gene links. For the ChIP-seq data processing, reads were mapped to the mm10 genome using Bowtie2 (v2.3.5.1), peaks were called with MACS2 (v2.2.7.1, with --format BAM --gsize mm --qvalue 0.05 --nomodel --keep-dup all --call-summits –nolambda as options) bigwig files were generated using deepTools bamCoverage function (v3.5.0, with --normalizeUsing CPM --binSize 1 as parameters). Coverage on the regulon regions was obtained with deepTools computeMatrix.

6. MPRA analysis: CHEQ-seq barcodes were extracted from the plasmid and cDNA samples (read 2) using cutadapt (v1.18) with parameters with options  -g TTATCATGTCTGCTCGAAGC...GATCGGCGCGCCTGCTCG --discard-untrimmed -m 17 -M 17 for the 12K libraries and g GTATCTTATCATGTCTGCTCGAAGC...GATCGGC  -j 10 --discard-untrimmed -m 18 -M 18 for the 455 library and seqkit (v0.10.2), with options seq -r -p, was used to get the reverse complement sequence. Reads were filtered to keep only those with quality > 30 using fastp (v0.20.0). Reads were assigned to enhancers based on the corresponding enhancer-barcode assignments, resulting in a count matrix with number of reads per enhancer and sample. Samples were processed using DESeq2 (v1.37.6), comparing the corresponding cDNA replicates versus their plasmid samples. For the FACS fractions, since we did not extract plasmid DNA from the samples, we used the plasmid replicates from the in vivo bulk experiment. To assess enhancer activity, we used the LogFC calculated by DESeq2 (v1.37.6). To distinguish active and inactive enhancers, a Gaussian fit of the shuffled negative control values was performed with robustbase (v0.93-6), and a p-value and Benjamini–Hochberg adjusted p-value was calculated based on that Gaussian fit for all enhancers. An enhancer is considered active if its adjusted p-value is < 0.1. For the FACS experiments, FACSDiva (v9.0.1) was used.

7. ATAC-seq analysis: Adapters we removed with fastq-mcf (ea-utils v1.12) and cleaned reads were mapped to the mm10 (or hg19 for HepG2) genome using Bowtie2 (v2.3.5.1). For the FACS-ATAC experiment, a fragment count matrix was generated using the liver scATAC-seq consensus peaks using SubRead (v1.6.3). AUCell was used to assess the enrichment of the `core` signatures (general, periportal, pericentral-intermediate and pericentral) in each of the fractions, using default parameters.

8. Human data analysis: Human liver data was obtained from Zhang et al (2021).  The authors labels and the scATAC-seq fragments were used as input for pycisTopic (v1.0.1.dev75+g3d3b721). Briefly, we first inferred consensus peaks as previously described, resulting in a data set with 121,593 regions and 6,366 cells. Topic modelling was performed using Mallet (v2.0), using 500 iterations and models with 2 topics and from 5 to 100 by an increase of 5. Drop-out imputation was performed by multiplying the region-topic and topic-cell probabilities. The imputed accessibility matrix was multiplied by 10^6. The mouse region-based regulons were transformed to hg38 coordinates using liftOver (https://genome.ucsc.edu/cgi-bin/hgLiftOver). The imputed accessibility matrix and the liftovered signatures were used as input for AUCell (v1.22.0) to assess regulon enrichment.

| Data analysis | VSN-Pipelines as available at https://vsn-pipelines.readthedocs.io/. pycisTopic is available at https://pycistopic.readthedocs.io/. pycistarget is available at https://pycistarget.readthedocs.io/. SCENIC+ is available at https://scenicplus.readthedocs.io/. ScoMAP is available at https://github.com/aertslab/ScoMAP. Notebooks to reproduce the main figures are available at https://github.com/aertslab/Bravo_et_al_Liver. |

DeepLiver is available in Zenodo (https://zenodo.org/record/8139953#) and is available in Kipoi (https://kipoi.org/docs/).

For manuscripts utilizing custom algorithms or software that are central to the research but not yet described in published literature, software must be made available to editors and reviewers. We strongly encourage code deposition in a community repository (e.g. GitHub). See the Nature Portfolio guidelines for submitting code & software for further information.

## Data

Policy information about availability of data

All manuscripts must include a data availability statement. This statement should provide the following information, where applicable:
- Accession codes, unique identifiers, or web links for publicly available datasets
- A description of any restrictions on data availability
- For clinical datasets or third party data, please ensure that the statement adheres to our policy

Data generated in this manuscript (single cell RNA-seq, single cell ATAC-seq, single cell multiome, Tbx3 ChIP-seq, MPRAs and bulk ATAC-seq) is available at GEO (GSE218472). Signatures for Ras signalling, Wnt signalling pituitary response and hypoxia were obtained from Halpern et al. (2017). Single cell RNA-seq data of the mouse liver at different time points of the circadian rhythm was downloaded from GEO (GSE145197). ChIP-seq data for Hnf4a, Cebpa, Foxa1 and Onecut was downloaded from ENA (PRJEB1571). Hi-C data was obtained from GEO (GSE65126). Raw snRNA data from male and female livers was downloaded from the SRA Project PRJNA779049 (Vehicle_Female_liver: SAMN23009762 and Vehicle_Male_liver: SAMN23009760). Bulk RNA-seq data of HepG2, Hepa1-6 and AML12 was obtained from GEO (ENCFF790EGR, GSE167316, GSE146053). Data for MPRA positive controls was retrieved from ENCODE (ENCFF288HIT, ENCFF032RDN), GEO (GSE71279) and Klein et al. (2018). Saturation mutagenesis data was downloaded from https://mpra.gs.washington.edu/satMutMPRA/ and Patwardhan et al. (2012). DeepExplainer plots for each of the wild-type zonated enhancers selected for the library design in Figure 5 are available in FigShare (DOI: 10.6084/m9.figshare.24115986). DeepExplainer plots for each of the wild-type zonated enhancers selected for the library design in Figure 5 are available in FigShare (DOI: 10.6084/m9.figshare.24115986). Human liver scATAC-seq data was downloaded from GEO (GSE184462). Processed data can be explored in Scope (http://scope.aertslab.org/#/Bravo_et_al_Liver, see Supplementary Note 4) and the UCSC genome browser (https://genome.ucsc.edu/s/cbravo/Bravo_et_al_Liver, see Supplementary Note 5). Source data are provided with this study.

## Human research participants

Policy information about studies involving human research participants and Sex and Gender in Research.

| | |
|---|---|
| Reporting on sex and gender | NA |
| Population characteristics | NA |
| Recruitment | NA |
| Ethics oversight | NA |

Note that full information on the approval of the study protocol must also be provided in the manuscript.

# Field-specific reporting

Please select the one below that is the best fit for your research. If you are not sure, read the appropriate sections before making your selection.

☒ Life sciences ☐ Behavioural & social sciences ☐ Ecological, evolutionary & environmental sciences

For a reference copy of the document with all sections, see nature.com/documents/nr-reporting-summary-flat.pdf

# Life sciences study design

All studies must disclose on these points even when the disclosure is negative.

| | |
|---|---|
| Sample size | No statistical method was used to predetermine sample size. For each analysis the sample size was sufficient to derive statistically meaningful results passing multiple testing procedures. |
| Data exclusions | - Transcriptome analysis: 10x snRNA-seq and 10x multiome (gene expression) runs were analyzed first independently using VSN-pipelines (v0.27.0). Briefly, cells with at least 350 genes expressed and a percentage of mitochondrial reads below 10% were kept. Scanpy (v1.8.2) was run with default parameters, using the number of principal components automatically selected by VSN-Pipelines and using Leiden clustering with resolutions 0.4, 0.6 and 0.8. Hepatocyte clusters with low gene expression and high percentage of mitochondrial reads were removed, as well as doublets called with Scrublet (v0.2.3). The samples were merged, obtaining 29,798 high-quality cells, and reanalyzed with VSN-Pipelines<br>- Epigenome analysis: We filtered the data set based on the scATAC-seq quality, keeping cells with at least 1,000 fragments, FRiP > 0.4 and TSS enrichment > 7, resulting in 22,600 high-quality cells. |
| Replication | For the single-cell experiments we performed at least two experiments per technique (snRNA-seq, snATAC-seq or single cell multiomics). For the smFISH we performed three replicates. For the MPRA experiments we performed at least two replicates per condition (system [HepG2, AML12, Mouse] and library). For the luciferase experiments we performed tfour replicates per condition. In all experiments, results between |

experiments were similar.

| Randomization | Not relevant, in each experiment groups were independent. |
|---|---|
| Blinding | Blinding was not relevant to the study as the computational analyses are fully reproducible. |

# Reporting for specific materials, systems and methods

We require information from authors about some types of materials, experimental systems and methods used in many studies. Here, indicate whether each material, system or method listed is relevant to your study. If you are not sure if a list item applies to your research, read the appropriate section before selecting a response.

## Materials & experimental systems

| n/a | Involved in the study |
|---|---|
| ☐ | ☒ Antibodies |
| ☐ | ☒ Eukaryotic cell lines |
| ☒ | ☐ Palaeontology and archaeology |
| ☐ | ☒ Animals and other organisms |
| ☒ | ☐ Clinical data |
| ☒ | ☐ Dual use research of concern |

## Methods

| n/a | Involved in the study |
|---|---|
| ☐ | ☒ ChIP-seq |
| ☐ | ☒ Flow cytometry |
| ☒ | ☐ MRI-based neuroimaging |

## Antibodies

| Antibodies used | Cells used for the FACS experiments were stained with the following antibodies (BioLegend) at a dilution of 1:300: PE anti-mouse/human CD324 E-cadherin (catalogue no. 147304) and APC anti-mouse CD73 (catalogue no. 127210). For ChIP-seq, Bethyl rabbit anti-TBX3 Antibody was used (Sanbio A303-098A, lot number #`1). |
|---|---|
| Validation | More information about the antibodies is provided at https://www.biolegend.com/en-us/products/pe-anti-mouse-human-cd324-e-cadherin-antibody-9276 and https://www.biolegend.com/en-us/products/apc-anti-mouse-cd73-antibody-7893. For the ChIP-seq antibody, more information is available at https://www.sanbio.nl/catalog/product/view/id/569891 and https://www.thermofisher.com/antibody/product/TBX3-Antibody-Polyclonal/A303-098A. The TBX3 antibody was tested for western blot and immunoprecipitation, giving similar results as the rabbit anti-TBX3 antibody BL8058, which recognizes a dowstream epitope (see website). |

## Eukaryotic cell lines

Policy information about cell lines and Sex and Gender in Research

| Cell line source(s) | The HepG2, Hepa1-6 and AML12 cell lines were purchased from ATCC (HB-8065, CRK-1830 and CRL-2254, respectively). |
|---|---|
| Authentication | Cell lines were not authentificated in-house, as they were obtained from a commercial source. |
| Mycoplasma contamination | HepG2, Hepa1-6 and AML12 cells tested negative for mycoplasma contamination. |
| Commonly misidentified lines (See ICLAC register) | We confirm that none of the cell lines used in this study is a Commonly Misidentified line. |

## Animals and other research organisms

Policy information about studies involving animals; ARRIVE guidelines recommended for reporting animal research, and Sex and Gender in Research

| Laboratory animals | Adult male mice (8 to 10 weeks old) were used in this study. All mice were C57BL/6JaxCrl except for mouse 1 in the single-cell experiments, which was Crl:CD-1. Mice were maintained under standard housing conditions, with continuous access to food and water; except for mice 4 and 5 in the single-cell experiments, for which food was removed approximately 10 hours before the experiments. |
|---|---|
| Wild animals | No wild animals were used in this study. |
| Reporting on sex | All mice used in this study were male. We used publically available data on female and male livers (Goldfarb et al. 2022) to assess the relevance of the eGRN networks inferred from male liver data. |
| Field-collected samples | No field samples were collected. |
| Ethics oversight | All animal experiments were conducted according to the KU Leuven ethical guidelines and approved by the KU Leuven Ethical |

| Ethics oversight | Committee for Animal Experimentation (approved protocol number ECD P007/2021). |

Note that full information on the approval of the study protocol must also be provided in the manuscript.

# ChIP-seq

## Data deposition

☒ Confirm that both raw and final processed data have been deposited in a public database such as GEO.

☒ Confirm that you have deposited or provided access to graph files (e.g. BED files) for the called peaks.

| Data access links<br>*May remain private before publication.* | Tbx3 ChIP-seq raw data, processed bigwig files and MACS called peaks are available at GEO (GEO218472). |
| Files in database submission | Fastq, bigwig and narrowPeak files. |
| Genome browser session<br>(e.g. UCSC) | Data can be visualized at https://genome.ucsc.edu/s/cbravo/Bravo_et_al_Liver |

## Methodology

| Replicates | Only 1 replicate per experiment (Tbx3 ChIP-seq and input) was performed |
| Sequencing depth | Libraries with fragment size of 300-500bp were sequenced using NextSeq2000 (paired end, 50bp per reads). In the Tbx3 ChIP sample, 74,325,203 reads were sequenced, out of which 93.1% were properly paired. In the input ChIP sample, 370,359,282 reads were sequenced, out of which 93.63% were properly paired. |
| Antibodies | Bethyl rabbit anti-TBX3 Ab (Sanbio A303-098A). |
| Peak calling parameters | Peaks were called with MACS2 (v2.2.7.1, with --format BAM --gsize mm --qvalue 0.01 --call-summits as options). |
| Data quality | Using MACS2, we identified 23,951 peaks (q-value < 0.01), out of which 19,812 overlap with regions accessible in the mouse liver. Motif enrichment was performed using pycisTarget with default parameters on the top 1,000 ChIP-seq regions, detecting the Tbx3 motif as expected. |
| Software | Reads were mapped to the mm10 genome using Bowtie2 (v2.3.5.1), peaks were called with MACS2 (v2.2.7.1, with --format BAM --gsize mm --qvalue 0.01 --call-summits as options) and bigwig files were generated using deepTools bamCoverage function (v3.5.0, with --normalizeUsing CPM --binSize 1 as parameters). |

# Flow Cytometry

## Plots

Confirm that:

☒ The axis labels state the marker and fluorochrome used (e.g. CD4-FITC).

☒ The axis scales are clearly visible. Include numbers along axes only for bottom left plot of group (a 'group' is an analysis of identical markers).

☒ All plots are contour plots with outliers or pseudocolor plots.

☒ A numerical value for number of cells or percentage (with statistics) is provided.

## Methodology

| Sample preparation | FACS MPRA in vivo: For intrahepatic delivery of the liver MPRA libraries, 8 to 10-week-old mice were secured and hydrodynamically injected with 20 µg of the libraries via the lateral tail vein. All libraries were diluted in sterile filtered 0.9% NaCl, and the total volume was adjusted to 10% (in mL) of the total body weight (in grams). 48 hours post-injection, mice were anesthetized by intraperitoneal injection of sodium pentobarbital (Nembutal, 50 mg/kg). Livers were perfused for 5 minutes with 40 mL of perfusion medium SC-1 (8 g/L Nacl, 400 mg/L KCl, 75.5 mg/L NaH2PO4, 120.5 mg/L Na2HPO4, 2,38 g/L HEPES, 350 mg/L NaHCO3, 190 mg/L EGTA, 900 mg/L D-(+)-Glucose, 1.2 mL Phenol Red solution) to remove the blood, followed by perfusion with 40 mL of SC-2 medium (8 g/L Nacl, 400 mg/L KCl, 75.5 mg/L NaH2PO4, 120.5 mg/L Na2HPO4, 2,38 g/L HEPES, 350 mg/L NaHCO3, 560 mg/L CaCl2.2H2O, 1.2 mL Phenol Red solution) containing 10 mg of collagenase P (Merck) for 5 minutes. Each lobe was dissected off and minced into small pieces in a beaker containing 39 mL SC-2 supplemented with 1 mL DNase I (Sigma-Aldrich) and 20 mg collagenase P, followed by rotating incubation for 15 min at 37°C. Hepatocytes were centrifuged for 2 min at 50 g, washed with PBS, centrifuged again for 2 min at 50 g, resuspended in 3 ml Hoechst buffer (DMEM + 10% FBS + 10 mM HEPES) and filtered through a 70 µm strainer. The protocol for hepatocyte staining was adapted from Ben-Moshe et al. (2019)61. After counting the cells on a LUNA cell counter, the concentration was adjusted to 5 million cells in 1 mL of Hoechst buffer. To determine the ploidy of hepatocytes, DNA was stained with Hoechst (Thermo Fisher Scientific) (15 µg/mL). Reserpine (5 µM) was also supplemented to prevent Hoechst expulsion from the cells. Cells were incubated for 30 min at 37°C. Hepatocytes were centrifuged for 5 min at 1,000 rpm at 4°C and the supernatant was discarded. Cells were resuspended in cold PBS in a concentration of 1 million cells in 100 µL. After spinning down (1,000 rpm for 5 min at 4°C), cells were resuspended in FACS buffer (2 mM EDTA, pH 8, and 0.5% BSA in 1x PBS) at a concentration of 1 |

million cells in 100 μL. Cells were stained with the following antibodies (BioLegend) at a dilution of 1:300: PE anti-mouse/human CD324 E-cadherin (catalogue no. 147304) and APC anti-mouse CD73 (catalogue no. 127210). FcX blocking solution (BioLegend catalogue no. 101319) was added at a dilution of 1:50.

FACS ATAC-seq: Hepatocytes were isolated and stained as described in the FACS MPRA section, with minor modifications. Cells were additionally stained with Alexa Fluor 488 Zombie Green (BioLegend) to enable the detection of viable cells by FACS. Zombie Green was added at a dilution of 1:500 and cells were kept in a rotator in the dark at room temperature for 15 min.

| Instrument | Cells were sorted by FACS-Aria-Fusion (BD Biosciences) using a 100 μm nozzle. |
|---|---|
| Software | FACSDiva (v9.0.1) |
| Cell population abundance | OmniATAC-seq was performed in the FAC-sorted fractions. AUCell was used to assess the enrichment of the `core` signatures (general, periportal, pericentral-intermediate and pericentral) in each of the fractions, using default parameters. |
| Gating strategy | FACS MPRA in vivo: FSC-A and SSC-A were used for hepatocytes size selection. Cells containing the library were selected based on GFP. Tetraploid hepatocytes were selected based on Hoechst stain. CD73 and Ecad were used to select hepatocytes bins along the porto-central axis, obtaining 100,000-200,000 cells per bin.<br>FACS ATAC-seq: FSC-A and SSC-A were used for hepatocytes size selection. Viable cells were selected based on the Zombie Green signal. Tetraploid hepatocytes were selected based on Hoechst stain. CD73 and Ecad were used to select hepatocytes bins along the porto-central axis, obtaining 20,000-50,000 cells per bin. |

☒ Tick this box to confirm that a figure exemplifying the gating strategy is provided in the Supplementary Information.

