## [Peer Review File · Nature Cell Biology]

Peer Review Information

Journal: Nature Cell Biology

Manuscript Title: Single-cell spatial multi-omics and deep learning dissect enhancer-driven gene regulatory networks in liver zonation

Corresponding author name(s): Professor Stein Aerts

Editorial Notes:

Reviewer Comments & Decisions:

Decision Letter, initial version:
--

*Please delete the link to your author homepage if you wish to forward this email to co-authors.

Dear Stein,

Your manuscript, "Enhancer grammar of liver cell types and hepatocyte zonation states", has now been seen by 4 referees, who are experts in liver zonation, scRNAseq-spatial transcriptomics, ATACseq (referee 1); enhancers-gene regulatory networks, massively parallel reporter assay, multiomics (referee 2); liver, computational methods (referee 3); and liver atlas (referee 4). As you will see from their comments (attached below) they find this work of potential interest, but have raised substantial concerns, which in our view would need to be addressed with considerable revisions before we can

consider publication in Nature Cell Biology. Please note that we are willing to further consider your manuscript as a Resource, rather than an article.

Nature Cell Biology editors discuss the referee reports in detail within the editorial team, including the chief editor, to identify key referee points that should be addressed with priority, and requests that are overruled as being beyond the scope of the current study. To guide the scope of the revisions, I have listed these points below. I should stress that the referees' concerns point to a premature dataset and these points would need to be addressed with experiments and data, and reconsideration of the study for this journal and re-engagement of referees would depend on strength of these revisions.

In particular, it would be essential to:

(A) Address the issues noted concerning the dietary background of mice used to generate the datasets, but also the claims about a link between circadian rhythm and batch effects, as indicated by:

Referee #1:

"Although the authors managed to create a convincing UMAP after removal of batch effects using Harmony, some conclusions are based on comparisons between cells that stem from different experimental conditions. For example, both Multiome samples were collected from starved mice. The snRNA and snATAC data are derived from mice that had continuous access to food. The authors noticed batch effects which they allocate to differences in circadian rhythm between these samples. However, it remains unclear if some of the observed differences could be rather attributed to the different experimental setups. The conclusion that certain GRNs depend on the animal's physiological state (including circadian rhythm) is therefore based on rather superficial data. The authors may consider removing this small part from their manuscript (page 6, lines 171-180, page 6, line 191). If the authors rather prefer to strengthen this idea, they should add samples from the same experimental setup (eg multiome) but derived from mice with distinct feeding procedures/from different times of the diurnal cycle (which could easily become a separate manuscript). Related to that topic: Some cell types are exclusively derived from one type of dataset (eg VECs only from Multiome vs LSECs only from snRNA/snATAC) at least according to FigureS7. The authors should comment on that and highlight this in the main text. More in general, an overall discussion about the limitations of their tools and experimental approaches should be included".

Referee #3:

"It is not clear why the authors included the circadian rhythm signatures from Droin et al. (2021) - this part is not well elaborated and I could not find the link to the rest of analysis, especially to zonation".

"The circadian data should be either better embedded in the liver zonation cell-and -space fate story eGRNs, or removed. I could not find how it contributed to the overall conclusions. The authors claim that circadian rhythm was among the batch effect differences in hepatocytes based on physiological states of the mice and that (Topic 17 and Topic 75) from supplementary Figure S8 represent different phases of the circadian rhythm. How can this be justified? It is also not clear how the publicly available scRNA-seq data on the mouse liver during different phases of the circadian rhythm from Droin et al. 2021 contributed to better understand the enhancer grammar of liver cell types and hepatocyte zonation states (Figs.S9-S10). In light of this, the summary statement needs modification "In summary, our spatial single cell

multiome atlas of the mouse liver reveals that both cell type identity and cell states, such as zonation and circadian rhythm, are congruently encoded at both the transcriptome and chromatin accessibility level".

(B) Address the issues noted by the referees regarding the use of HepG2 cells in the study.

Referee #1 says:

"In order to validate their findings in human, the authors perform LOF of Tbx3 in HepG2 cells using bulk MPRA. However, HepG2 cells cannot be used as an appropriate model for zonation in humans, even if they express Tbx3. Hepatocellular carcinoma, from which HepG2 cells are derived, generally show loss of proper liver zonation, and cells in culture in general anyway lose zoned gene expression. HepG2 cells should therefore not be used to study mechanisms related to liver zonation. These experiments and the corresponding conclusions should be completely removed from the MS since they might be misleading in this context. The manuscript is also very exciting without this data".

Referee #3 says:

"A bit confusing is the use of human HepG2 cells and not a mouse hepatic cell line, such as Hepa1. A mouse cell line would make it easier to answer some questions, also if the distal enhancer activation is more difficult in immortalized hepatic cell lines compared to the cells in the liver – a very important question to be addressed. Translation to the human remains difficult for the topic of zonation. Data from 2D cell cultures are useful but with limiting biological relevance that we have to acknowledge. On top of that is also the importance of the liver sex/ gender that was not even mentioned in the manuscript".

Referee #4 says:

"The authors observed an obvious disproportion of active regions in the enhancer/promoter regions between mouse and human hepatocytes, and further stated that 'either that distal enhancer activation is more difficult to recapitulate in HepG2 cells, or that distal enhancers are less conserved than promoters between mouse and human'. Not enough evidence was provided to support this conclusion. Only 7,198 valid regions were captured in total, which represents a small portion of the actual regulatory regions in vivo. Thus, this conclusion is not well supported due to insufficient coverage of the regulatory regions".

(C) Address the issue noted by referee #3 regarding the use of only the male gender and either discuss the drawbacks or address with experiments if possible:

"A drawback is to investigate the only the male gender – HepG2 cells are of male origin (HepG2), and only male mice were used. The data and conclusions should thus not be generalized for both sexes. Liver is, after the gonads, the most sexually dimorphic organ (Lefebvre P, Steals B, Nature Endocrinol, 2021, <https://www.nature.com/articles/s41574-021-00538-6>, Cvitanovic et al., Hepatology 2017 <https://pubmed.ncbi.nlm.nih.gov/28520105/> and other relevant references). Vandel J et al., Hepatology 2021 (<https://aasldpubs.onlinelibrary.wiley.com/doi/10.1002/hep.31312>) recently described how the large-scale analysis of transcriptomic profiles from human livers emphasized the

sexually dimorphic nature of NASH (nonalcoholic steatohepatitis) as a liver disease state and its link with fibrosis. They call for the integration of sex as a major determinant of liver responses to liver disease progression and the responses to drugs".

(D) Further investigate and substantiate claims about the relationship and effects of Tbx3 and Tcf7l1, as indicated by:

Referee #1:

"Although this is a resource manuscript, in vivo validation showing that Tbx3 and Tcf7l1 act as key repressors controlling liver zonation should be performed. Besides validating this interesting finding, this would also highlight overall translational value of the resource data. The analyses performed by the authors are based on chromatin accessibility and expression, and although this can be a strong indicator, only ChIP assays allow a proper statement on target regions/binding. The authors could perform a Cut&Run approach (or any other similar preferred sequencing approach)".

"The authors state that Tbx3 and Tcf7l1 binding sites are located predominantly within hepatocyte enhancers, and in close proximity to binding sites of core transcription factors. It would be interesting to validate this predicted proximity between these repressors and the core factors, eg by a proximity ligation assay".

Referee #4:

"The authors found that Tbx3 and Tcf7l1 directly repress each other by using SCENIC+ eGRN. This is very interesting, but more supportive evidence is needed. For example, did the author check the public ChIP-seq data of TBX3 or TCF7L1 in hepatocytes. Does TBX3 directly bind to the enhancers of Tcf7l1, and vice versa?"

"In the discussion, the authors proposed a feedback loop between Tbx3 and Tcf7l1 in hepatocyte zonation. There might be several issues of this model. First, not enough evidence was provided for the direct binding of Tbx3 and Tcf7l1. Second, according to the illustration, Tcf7l2 also directly regulates Tbx3, but no data in this paper supports this conclusion. Third, according to Fig 2a, Tbx3 was highly expressed in pericentral hepatocytes. But the number of potential binding sites for Tbx3 is very small compared with that in periportal hepatocyte. More evidence is needed to support that Tbx3 directly binds to Tcf7l1 with such limited binding options".

(E) Please ensure that all datasets are available to the referees, so that they can properly evaluate them.

Referee #3 says:

"The authors promise availability to explore the resource at http://scope.aertslab.org/#/Bravo_et_al_Liver, however, the datasets and their visualisation seem not to be yet available publically (maybe we can see them for review purposes?). At the UCSC Genome Browser27 (https://genome.ucsc.edu/s/cbravo/Bravo_et_al_Liver) one can find the useful Chip-Seq data with transcription factor binding sites in different liver cell types, but it was not clear to me how this relates to zonation - some explanatory sentences are lacking for broader understanding".

(F) Please clarify and address any issues raised regarding the sequencing approaches and processes, as indicated by:

Referee #3:

"The epigenome analysis on scATAC-seq samples is written in a less understandable manner. It is not clear in which step this data were integrated to provide finally the active regulons which should show also open chromatin structures?"

"Table S2. 12K MPRA metadata and measurements – there are multiple blank lines in the column AN, "activity expression". What is a difference between "none" or a blank section? From 12000 reporter probes, how many were not found active in vivo nor in vitro in HepG2 cells? How many were "blank"?"

"Chr1: 194610309-194610809 - TF labels are missing for: The enhancer accessibility only in hepatocytes, What makes the enhancer periportal and What makes the enhancer to be active".

Referee #4:

"In Fig S1, the number of UMIs and genes vary significantly even within each individual cell type. Was normalization (e.g., sequencing depth normalization) properly performed for the data?"

(G) All other referee concerns pertaining to strengthening existing data, providing controls, methodological details, clarifications and textual changes, should also be addressed.

(H) Finally please pay close attention to our guidelines on statistical and methodological reporting (listed below) as failure to do so may delay the reconsideration of the revised manuscript. In particular please provide:

- a Supplementary Figure including unprocessed images of all gels/blots in the form of a multi-page pdf file. Please ensure that blots/gels are labeled and the sections presented in the figures are clearly indicated.
- a Supplementary Table including all numerical source data in Excel format, with data for different figures provided as different sheets within a single Excel file. The file should include source data giving rise to graphical representations and statistical descriptions in the paper and for all instances where the figures present representative experiments of multiple independent repeats, the source data of all repeats should be provided.

We would be happy to consider a revised manuscript that would satisfactorily address these points, unless a similar paper is published elsewhere, or is accepted for publication in Nature Cell Biology in the meantime.

- ensure that it conforms to our format instructions and publication policies (see below and

<https://www.nature.com/nature/for-authors>).

- provide a point-by-point rebuttal to the full referee reports verbatim, as provided at the end of this letter.
- provide the completed Reporting Summary (found here <https://www.nature.com/documents/nr-reporting-summary.pdf>). This is essential for reconsideration of the manuscript will be available to editors and referees in the event of peer review. For more information see <http://www.nature.com/authors/policies/availability.html> or contact me. Please also make sure to provide an explicit statement regarding the use of Commonly Misidentified Lines in the Reporting Summary.

When submitting the revised version of your manuscript, please pay close attention to our [href="https://www.nature.com/nature-portfolio/editorial-policies/image-integrity">Digital Image Integrity Guidelines](https://www.nature.com/nature-portfolio/editorial-policies/image-integrity). and to the following points below:

Nature Cell Biology is committed to improving transparency in authorship. As part of our efforts in this direction, we are now requesting that all authors identified as 'corresponding author' on published papers create and link their Open Researcher and Contributor Identifier (ORCID) with their account on the Manuscript Tracking System (MTS), prior to acceptance. ORCID helps the scientific community achieve unambiguous attribution of all scholarly contributions. You can create and link your ORCID from the home page of the MTS by clicking on 'Modify my Springer Nature account'. For more information please visit www.springernature.com/orcid.

This journal strongly supports public availability of data. Please place the data used in your paper into a public data repository, or alternatively, present the data as Supplementary Information. If data can only be shared on request, please explain why in your Data Availability Statement, and also in the correspondence with your editor. Please note that for some data types, deposition in a public repository is mandatory - more information on our data deposition policies and available repositories appears below.

[Redacted]

We would like to receive a revised submission within six months.

We hope that you will find our referees' comments, and editorial guidance helpful. Please do not hesitate to contact me if there is anything you would like to discuss.

Best wishes,

Stelios

Stylios Lefkopoulos, PhD
He/him/his
Associate Editor
Nature Cell Biology
Springer Nature
Heidelberger Platz 3, 14197 Berlin, Germany

E-mail: stylios.lefkopoulos@springernature.com
Twitter: @s_lefkopoulos

Reviewers' Comments:

Reviewer #1:

Remarks to the Author:

In their manuscript entitled "Enhancer grammar of liver cell types and hepatocyte zonation states" Bravo González-Blas et al. uncover enhancer-driven gene regulatory networks (eGRN) that are involved in liver zonation by combining single cell multiomics, spatial omics, GRN inference, and deep learning. Specifically, the authors performed combined single cell RNA- and ATAC-seq, as well as spatial transcriptomics on the mouse liver and found that cell type identity and cell states (e.g. zonation) are encoded by the transcriptome and chromatin accessibility. Applying a tool called SCENIC+, Bravo González-Blas and colleagues identify several core general hepatocyte transcription factors, including Hnf4a, Hnf1a, Cebpa, Onecut1, Foxa1 and Nfib, and importantly, highlight Tbx3 and Tcf7l1 as key repressors of periportal and pericentral gene expression, respectively. The authors then performed a Massively Parallel Reporter Assay (MPRA) to determine enhancer sequence activity and found that around 40% of the accessible regions in hepatocytes are active. By training a hierarchical Deep Learning model, called DeepLiver, the authors provide a tool to decode and predict hepatocyte enhancer accessibility, activity, and zonation. Finally, the authors performed computational validation of the zoned transcription factors by simulated KD and OE, but also experimental validation by MPRA assays, thus, highlighting the importance of Tbx3 and Tcf7l1/2 binding sites within hepatocyte enhancers to drive zonation, while Hnf1a and Hnf4a binding was crucial for enhancer activity. I really enjoyed reading this exciting manuscript that is of high importance, well written, has beautiful and informative illustrations and Figures, elegant bioinformatics tools, and uncovers interesting and novel aspects of mouse liver zonation regulation by eGRNs. However, some aspects of this work should be clarified/improved, as detailed below.

Major comments:

1. Although the authors managed to create a convincing UMAP after removal of batch effects using Harmony, some conclusions are based on comparisons between cells that stem from different experimental conditions. For example, both Multiome samples were collected from starved mice. The snRNA and snATAC data are derived from mice that had continuous access to food. The authors noticed batch effects which they allocate to differences in circadian rhythm between these samples.

However, it remains unclear if some of the observed differences could be rather attributed to the different experimental setups. The conclusion that certain GRNs depend on the animal's physiological state (including circadian rhythm) is therefore based on rather superficial data. The authors may consider removing this small part from their manuscript (page 6, lines 171-180, page 6, line 191). If the authors rather prefer to strengthen this idea, they should add samples from the same experimental setup (eg multiome) but derived from mice with distinct feeding procedures/from different times of the diurnal cycle (which could easily become a separate manuscript). Related to that topic: Some cell types are exclusively derived from one type of dataset (eg VECs only from Multiome vs LSECs only from snRNA/snATAC) at least according to FigureS7. The authors should comment on that and highlight this in the main text. More in general, an overall discussion about the limitations of their tools and experimental approaches should be included.

2. Although this is a resource manuscript, *in vivo* validation showing that Tbx3 and Tcf7l1 act as key repressors controlling liver zonation should be performed. Besides validating this interesting finding, this would also highlight overall translational value of the resource data. The analyses performed by the authors are based on chromatin accessibility and expression, and although this can be a strong indicator, only ChIP assays allow a proper statement on target regions/binding. The authors could perform a Cut&Run approach (or any other similar preferred sequencing approach).

3. In order to validate their findings in human, the authors perform LOF of Tbx3 in HepG2 cells using bulk MPRA. However, HepG2 cells cannot be used as an appropriate model for zonation in humans, even if they express Tbx3. Hepatocellular carcinoma, from which HepG2 cells are derived, generally show loss of proper liver zonation, and cells in culture in general anyway lose zoned gene expression. HepG2 cells should therefore not be used to study mechanisms related to liver zonation. These experiments and the corresponding conclusions should be completely removed from the MS since they might be misleading in this context. The manuscript is also very exciting without this data.

4. The authors state that Tbx3 and Tcf7l1 binding sites are located predominantly within hepatocyte enhancers, and in close proximity to binding sites of core transcription factors. It would be interesting to validate this predicted proximity between these repressors and the core factors, eg by a proximity ligation assay.

5. Last year it was published (PMID: 34129813) that hepatocytes have an open chromatin configuration for both periportal and pericentral genes, regardless in which zone they reside. The work by Bravo González-Blas et al. provides important insights into how these genes may be repressed to confer zonation but does not mention how their findings relate to such previous work. In addition, the authors did not cite any of the seminal papers by the Zaret lab, dissecting the epigenome/GRNs defining hepatic cell identity. While the authors are certainly experts in GRNs/multi-omics and cool bioinformatics, and adequately cite such work, they should be more inclusive in discussing existing liver literature.

6. Spatial positioning of cells from sc/sn profiling studies is a central tool for studying zoned effects in the liver. It would be important that the authors discuss how their approach differs from what the Itzkovitz lab has developed.

Minor comments:

1. In general, all figures are too small. Especially, the font size makes many figures hard to read.

2. Fig 1e: It is not clear to which lines exactly the highlighted genes are assigned to. Can the authors clarify this in the figure?

3. Fig 2b-e: figure legends missing

4. In Figure S7b it looks like mouse 1 was the CD-1 strain, however, in the methods section the authors state that this is mouse 3.
5. In FigS9/S10 all legends/scales are missing for color coding. Also, for Figures S16a.
6. Typo line 236: "assess" to "assessed"
7. Typo line 476: "FAC" to "FACS"
8. Typo line 587: ",," to "."
9. Figure S19 is blurry.
10. Can the authors state how they defined the terms "promoters" and "enhancers"? Regulatory elements may have both enhancer and promoter functions and distinct factors may determine these activities (see <https://doi.org/10.1038/s41576-019-0173-8>). In this regard, the authors should discuss the limitations that comes with their chosen MPRA assay.
11. Page 6, line 182: the Halpern et al paper is not a correct reference here. This is "just" a scRNAseq resource paper without functional validation. The mechanistic role of these pathways in zonation have been published earlier elsewhere. Better use a recent review article about liver zonation that covers the original papers or cite them directly.

Reviewer #2:

Remarks to the Author:

In the manuscript entitled "Enhancer grammar of liver cell types and hepatocyte zonation states", Bravo and colleagues generated single-cell multiomics (scRNA-seq and scATAC-seq) and spatial omics data (single molecule FISH) to reveal gene regulatory network across mouse liver cell types. They found that zonation states of the liver are regulated by transcription factors (Hnf4a, Cebpa, Hnf1a, Onecut1 and Foxa1) and repressors (Tcf7l1 and Tbx3). Furthermore, they performed in vivo massively parallel reporter assays (MPRAs) to examine >10,000 candidate regulatory elements for their enhancer activity in mouse liver and HepG2, and identified 2,913 active enhancers. They developed a deep learning model (DeepLiver) to dissect the function of these TF binding motifs and predict their regulatory grammar. Their omics approach associating gene regulation and spacial information, as well as machine learning approach based on in vivo MPRA that allows to reveal regulatory code at base pair resolution, are impactful broadly in the gene regulatory genomics field. Their datasets, computational tools, and browsers (Scope and UCSC genome browser) are robust and useful as resources in the research community.

Minor comments:

1. They termed a TF with its set of predicted target enhancers and regions "eRegulon" (lines 238-239) but used "regulons" instead in the following sentences (e.g., line 240, "This analysis revealed 180 regulons"). Please check if these should be "eRegulons" or not.
2. MPRA reproducibility (Figure S14a, b) was not clear to me. Why 3' mouse plasmids were not reproducible between replicates, while 5' mouse plasmids reproducible? Please add some explanation about how the MPRA data are reproducible in the text.

3. I am curious whether DeepLiver is useful for de novo functional motif discovery. For example, in figure 4c, around the 310bp region in the *Cdh1* enhancer seems to associate negatively with the accessibility but not highlighted here. Does this sequence overlap with any known TF motifs? Any other regions that are potentially interesting as functional motifs found in this analysis?

Reviewer #3:

Remarks to the Author:

A. Summary of the key results

The manuscript of C.B. Gonzalez-Blas represents a complex systems biology paper representing the spatial multiomics atlas of the mouse liver sinusoid, with characteristic zonation from periportal to pericentral regions, at the single cell level. The major added value is the prediction and validation of cell-type specific gene regulatory networks (eGRNs) through analysis of active enhancers, and also the cell state specific eGRNs that depend also on the cell location. The massive parallel reporter assay (MPRA) in vitro and in vivo aided in defining active regulons that are characteristic for each cell type or for groups of cells. A DeepLiver deep learning model was applied to validate experimental data, especially to predict enhancer accessibility and activity, as well as zonation state of a cell. An interesting observation is that in the mouse more distal enhancers are active compared to promoters, while in HepG2 immortal cells situation was the opposite. The authors propose that either distal enhancer activation is more difficult in HepG2 cells, or that distal enhancers are less conserved than promoters between mouse and human.

While it is not completely novel that the cell state changes in transcription and chromatin accessibility in hepatocytes, liver sinusoidal endothelial cells and hepatic stellate cells depend on zonation, the novelty lies in determining transcription repressors (by eGRN mapping) that define the periportal zonation (*Tbx3*), and pericentral zonation (*Tcf7l1*). *Tbx3* is a transcriptional repressor essential during early embryonic development, in the formation other organ systems, and in tumorigenesis while *Tcf7l1* predominantly acts as a repressor of Wnt target gene expression, (while together with *Lef1* can act as transcriptional activator). The five transcription factors that were determined to control the core hepatocyte gene regulatory networks (*Hnf4a*, *Cebpa*, *Hnf1a*, *Onecut1* and *Foxa1*) were confirmed also in validation experiments where data from MPRA were applied to train the DeepLiver model. The above transcription factors were identified as drivers of enhancer specificity. It is interesting that *Tbx3* and *Tcf7l1* expression profiles are anti-correlated with the accessibility of their potential target regions, i. e. *Tbx3* is expressed only in pericentral hepatocytes, while its candidate target regions are only accessible periportal. Novel is also the finding that *Tbx3* and *Tcf7l1* repress each other and in this manner control the zonation of downstream gene expression.

B. Originality and significance

Previous published single-cell and spatial transcriptomics studies have shown that not only hepatocyte function, but also the transcriptome, varies along the periportal-pericentral liver lobule axis, described also as zonation. The novelty of this paper is to elucidate how zonation interacts with the gene regulatory networks (GRNs) of hepatic cells and to apply single cell data to predict whether a regulatory region is active. This is a significant work and might represent a good data resource for the liver scientists interested in zonation of the liver metabolism, where not only the cell type but also location of the cells defines its metabolic state, described herein by multiomics single cell data. The challenge of how to infer GRNs from single cell data to predict whether a regulatory region is active was also solved in this work by combination of experimental and deep learning modelling, that allowed also mutagenesis in silico.

A drawback is to investigate the only the male gender – HepG2 cells are of male origin (HepG2), and only male mice were used. The data and conclusions should thus not be generalized for both sexes. Liver is, after the gonads, the most sexually dimorphic organ (Lefebvre P, Steels B, Nature Endocrinol,

2021, <https://www.nature.com/articles/s41574-021-00538-6>, Cvitanovic et al., *Hepatology* 2017 <https://pubmed.ncbi.nlm.nih.gov/28520105/> and other relevant references). Vandel J et al., *Hepatology* 2021 (<https://aasldpubs.onlinelibrary.wiley.com/doi/10.1002/hep.31312>) recently described how the large-scale analysis of transcriptomic profiles from human livers emphasized the sexually dimorphic nature of NASH (nonalcoholic steatohepatitis) as a liver disease state and its link with fibrosis. They call for the integration of sex as a major determinant of liver responses to liver disease progression and the responses to drugs.

C. Data & methodology: validity of approach, quality of data, quality of presentation

The paper is relatively easy to read despite comprehensive methodology. Experimental and computational approaches are appropriate for such complex questions. The methodology is up-to date. The number of replicates is stated. I focus on main experimental approaches and their presentation. In brief:

- From snRNAseq (10x) and multi-ome gene expression 14 hepatic cell types were identified.
- The epigenome analysis on scATAC-seq samples was performed.
- E-regulons represent a crucial part of the manuscript. Gene-based and region-based regulons were scored based on other datasets that led to 180 high quality regulons.
- smFISH image analysis resulted in identification of 19 clusters with 11 cell types that were annotated based on marker gene expression with a panel of 100 selected genes across cell types and cell states in the liver that represents a crucial reagent.
- Hi-C and Chip-Seq publically available data were used to validate the regulons.
- MPRA was applied to measure the regulatory function of DNA sequences.
- Downstream analyses included the pseudotime order which represents the distance along the portal-central axis and identifies numbers of genes and regions in hepatic stellate cells (HSC) Liver sinusoidal endothelial cells (LSEC) and in hepatocytes (that hold about 10 times more genes and 20 – 40 times more regions compared to the other two cell types). Regulons are then stratified by zonation and the sample by PCA.

It is not clear why the authors included the circadian rhythm signatures from Droin et al. (2021) - this part is not well elaborated and I could not find the link to the rest of analysis, especially to zonation.

D. Appropriate use of statistics and treatment of uncertainties

I am not a computation specialist. From the approaches described, the computation is comprehensive, multi-level and statistically sound (adjusted p values) and up to date data integration techniques have been applied.

E. Conclusions: robustness, validity, reliability

Conclusions based on experimental data and computational predictions are largely concordant. Data were proven from different angles. DeepLiver proved to be a good prediction tool.

A bit confusing is the use of human HepG2 cells and not a mouse hepatic cell line, such as Hepa1. A mouse cell line would make it easier to answer some questions, also if the distal enhancer activation is more difficult in immortalized hepatic cell lines compared to the cells in the liver – a very important question to be addressed. Translation to the human remains difficult for the topic of zonation. Data from 2D cell cultures are useful but with limiting biological relevance that we have to acknowledge. On top of that is also the importance of the liver sex/ gender that was not even mentioned in the manuscript.

F. Suggested improvements: experiments, data for possible revision

The authors promise availability to explore the resource at http://scope.aertslab.org/#/Bravo_et_al_Liver, however, the datasets and their visualisation seem not to be yet available publically (maybe we can see them for review purposes?). At the UCSC Genome

Browser27 (https://genome.ucsc.edu/s/cbravo/Bravo_et_al_Liver) one can find the useful Chip-Seq data with transcription factor binding sites in different liver cell types, but it was not clear to me how this relates to zonation - some explanatory sentences are lacking for broader understanding.

The epigenome analysis on scATAC-seq samples is written in a less understandable manner. It is not clear in which step this data were integrated to provide finally the active regulons which should show also open chromatin structures?

Table S2. 12K MPRA metadata and measurements – there are multiple blank lines in the column AN, “activity expression”. What is a difference between “none” or a blank section? From 12000 reporter probes, how many were not found active in vivo nor in vitro in HepG2 cells? How many were “blank”?

Supplementary notes,

Chr1: 194610309-194610809 - TF labels are missing for: The enhancer accessibility only in hepatocytes, What makes the enhancer periportal and What makes the enhancer to be active.

The circadian data should be either better embedded in the liver zonation cell-and –space fate story eGRNs, or removed. I could not find how it contributed to the overall conclusions.

The authors claim that circadian rhythm was among the batch effect differences in hepatocytes based on physiological states of the mice and that (Topic 17 and Topic 75) from supplementary Figure S8 represent different phases of the circadian rhythm. How can this be justified?

It is also not clear how the publicly available scRNA-seq data on the mouse liver during different phases of the circadian rhythm from Droin et al. 2021 contributed to better understand the enhancer grammar of liver cell types and hepatocyte zonation states (Figs.S9-S10).

In light of this, the summary statement needs modification “In summary, our spatial single cell multiome atlas of the mouse liver reveals that both cell type identity and cell states, such as zonation and circadian rhythm, are congruently encoded at both the transcriptome and chromatin accessibility level.

The title, abstract and conclusions should not be generalized to both sexes if only male cells and male mice have been used in experiments. The aspect of gender-sex should be at least mentioned and if possible, elaborated.

G. References: appropriate credit to previous work?

Most referring is appropriate.

I suggest the authors to discuss the zonation and specific cell markers in light of the paper by Inverso D. et al., *Developmental Cell* 2021, <https://doi.org/10.1016/j.devcel.2021.05.001>, where they combined spatial single cell sorting with transcriptomics and quantitative proteomics/phosphoproteomics, to established the spatially resolved proteome landscape of the liver endothelium, enriching the mechanistic insight into zoned vascular signaling mechanisms. It would be interesting to learn to which extent are the zonation gene markers of C.B. Gonzalez-Blas related to gene markers of Inverso D et al. (Inverso et al., Results section: Spatial multiomics of the liver endothelium; Transcriptome zonation defines distinct L-EC signatures).

H. Clarity and context: lucidity of abstract/summary, appropriateness of abstract, introduction and conclusions

The writing and the content of the chapters are appropriate. Specific remarks were listed above.

Prof. dr. Damjana Rozman

Reviewer #4:

Remarks to the Author:

In the study, González-Blas et al performed transcriptome and genomic accessibility profiling of mouse liver tissues at single-cell level. They found zonation patterns along the portal-central axis in hepatocytes, hepatic stellate cells, and liver sinusoidal endothelial cells in terms of gene expression, region accessibility and signaling pathways. They further utilized smFISH to validate the spatial variations detected from the single-cell analysis. The authors identified two repressors, i.e., Tcf7l1 and Tbx3, as important regulators of the zonation states in hepatocytes using GRN inference and DeepLiver methods.

The zonation of liver has been well known. The authors applied new technologies to provide an insight into the TF regulatory network of the zonation in hepatocytes. Overall, 1) large datasets of single-cell multiome (snATAC+snRNA), snATAC, and snRNA of mouse liver tissue were provided in the paper. The datasets would be useful to the field. However, since those were derived from mice, the impact of the data would be limited compared with human data; 2) the two repressors of Tcf7l1 and Tbx3 identified in the manuscript have been reported in literature (Ben-Moshe et al Nature Metabolism, 2019; Brosch et al Nature Communications, 2018); 3) some of the conclusions in this paper were overstated and more evidence will be needed to support the statements.

Major:

1. In Fig S1, the number of UMIs and genes vary significantly even within each individual cell type. Was normalization (e.g., sequencing depth normalization) properly performed for the data?
2. The authors observed an obvious disproportion of active regions in the enhancer/promoter regions between mouse and human hepatocytes, and further stated that 'either that distal enhancer activation is more difficult to recapitulate in HepG2 cells, or that distal enhancers are less conserved than promoters between mouse and human'. Not enough evidence was provided to support this conclusion. Only 7,198 valid regions were captured in total, which represents a small portion of the actual regulatory regions in vivo. Thus, this conclusion is not well supported due to insufficient coverage of the regulatory regions.
3. Similar issues for the conclusion stated in lines 324-326. In addition, only one human cell line was applied. There's a possibility that the observation may be derived from unknown bias. More evidence is needed to support the conclusion in this section.
4. The authors found that Tbx3 and Tcf7l1 directly repress each other by using SCENIC+ eGRN. This is very interesting, but more supportive evidence is needed. For example, did the author check the public ChIP-seq data of TBX3 or TCF7L1 in hepatocytes. Does TBX3 directly bind to the enhancers of Tcf7l1, and vice versa?
5. In the discussion, the authors proposed a feedback loop between Tbx3 and Tcf7l1 in hepatocyte zonation. There might be several issues of this model. First, not enough evidence was provided for the direct binding of Tbx3 and Tcf7l1. Second, according to the illustration, Tcf7l2 also directly regulates Tbx3, but no data in this paper supports this conclusion. Third, according to Fig 2a, Tbx3 was highly expressed in pericentral hepatocytes. But the number of potential binding sites for Tbx3 is very small compared with that in periportal hepatocyte. More evidence is needed to support that Tbx3 directly binds to Tcf7l1 with such limited binding options.

Minor:

1. In lines 131-134, the authors described smFISH experiment in the liver but referred to Fig 1f. Similar mistake for Fig 1g in lines 140-145.
2. In lines 253-265, the authors classified hepatocyte regulons based on their zonation state and mouse status using PCA. More methodology details would be helpful for the readers to understand this

part.

3. In lines 382-384, the authors stated that 'DeepLiver predictions of the effect of enhancer mutations correlate with experimental results ($R=0.36-0.75$, Fig S17)'. But the Fig S17 showed different correlation results. Same issue in lines 385-387, there are no negative correlation values in Fig S17.

4. In lines 495-501, the authors tested selected enhancers and their Tbx3 LOF variants in HepG2 cells. It's worth to include Tcf7l1 enhancers to test the direct repression as described in lines 460-461.

Methods should be written concisely, but should contain all elements necessary to allow interpretation and replication of the results. As a guideline, Methods sections typically do not exceed 3,000 words. The Methods should be divided into subsections listing reagents and techniques. When citing previous methods, accurate references should be provided and any alterations should be noted. Information must be provided about: antibody dilutions, company names, catalogue numbers and clone numbers for monoclonal antibodies; sequences of RNAi and cDNA probes/primers or company names and catalogue numbers if reagents are commercial; cell line names, sources and information on cell line identity and authentication. Animal studies and experiments involving human subjects must be reported in detail, identifying the committees approving the protocols. For studies involving human subjects/samples, a statement must be included confirming that informed consent was obtained. Statistical analyses and information on the reproducibility of experimental results should be provided in a section titled "Statistics and Reproducibility".

All Nature Cell Biology manuscripts submitted on or after March 21 2016 must include a Data availability statement as a separate section after Methods but before references, under the heading "Data Availability". For Springer Nature policies on data availability see <http://www.nature.com/authors/policies/availability.html>; for more information on this particular policy see <http://www.nature.com/authors/policies/data/data-availability-statements-data-citations.pdf>. The Data availability statement should include:

- Accession codes for primary datasets (generated during the study under consideration and designated as "primary accessions") and secondary datasets (published datasets reanalysed during the study under consideration, designated as "referenced accessions"). For primary accessions data should be made public to coincide with publication of the manuscript. A list of data types for which submission to community-endorsed public repositories is mandated (including sequence, structure, microarray, deep sequencing data) can be found here <http://www.nature.com/authors/policies/availability.html#data>.
- Unique identifiers (accession codes, DOIs or other unique persistent identifier) and hyperlinks for datasets deposited in an approved repository, but for which data deposition is not mandated (see here

for details <http://www.nature.com/sdata/data-policies/repositories>).

- At a minimum, please include a statement confirming that all relevant data are available from the authors, and/or are included with the manuscript (e.g. as source data or supplementary information), listing which data are included (e.g. by figure panels and data types) and mentioning any restrictions on availability.
- If a dataset has a Digital Object Identifier (DOI) as its unique identifier, we strongly encourage including this in the Reference list and citing the dataset in the Methods.

We recommend that you upload the step-by-step protocols used in this manuscript to the Protocol Exchange. More details can be found at www.nature.com/protocolexchange/about.

All imaging data should be accompanied by scale bars, which should be defined in the legend. Cropped images of gels/blots are acceptable, but need to be accompanied by size markers, and to retain visible background signal within the linear range (i.e. should not be saturated). The boundaries of panels with low background have to be demarked with black lines. Splicing of panels should only be considered if unavoidable, and must be clearly marked on the figure, and noted in the legend with a statement on whether the samples were obtained and processed simultaneously. Quantitative comparisons between samples on different gels/blots are discouraged; if this is unavoidable, it should only be performed for samples derived from the same experiment with gels/blots were processed in parallel, which needs to be stated in the legend.

- For line art, graphs, charts and schematics we prefer Adobe Illustrator (.AI), Encapsulated PostScript (.EPS) or Portable Document Format (.PDF). Files should be saved or exported as such directly from the application in which they were made, to allow us to restyle them according to our journal house style.
- We accept PowerPoint (.PPT) files if they are fully editable. However, please refrain from adding PowerPoint graphical effects to objects, as this results in them outputting poor quality raster art. Text

used for PowerPoint figures should be Helvetica (preferred) or Arial.

Unprocessed scans of all key data generated through electrophoretic separation techniques need to be presented in a supplementary figure that should be labelled and numbered as the final supplementary figure, and should be mentioned in every relevant figure legend. This figure does not count towards the total number of figures and is the only figure that can be displayed over multiple pages, but should be provided as a single file, in PDF or TIFF format. Data in this figure can be displayed in a relatively informal style, but size markers and the figures panels corresponding to the presented data

must be indicated.

The total number of Supplementary Figures (not including the “unprocessed scans” Supplementary Figure) should not exceed the number of main display items (figures and/or tables (see our Guide to Authors and March 2012 editorial <http://www.nature.com/ncb/authors/submit/index.html#suppinfo>; <http://www.nature.com/ncb/journal/v14/n3/index.html#ed>). No restrictions apply to Supplementary Tables or Videos, but we advise authors to be selective in including supplemental data.

GUIDELINES FOR EXPERIMENTAL AND STATISTICAL REPORTING

REPORTING REQUIREMENTS – We are trying to improve the quality of methods and statistics reporting in our papers. To that end, we are now asking authors to complete a reporting summary that collects information on experimental design and reagents. The Reporting Summary can be found here <https://www.nature.com/documents/nr-reporting-summary.pdf> If you would like to reference the guidance text as you complete the template, please access these flattened versions at <http://www.nature.com/authors/policies/availability.html>.

We strongly recommend the presentation of source data for graphical and statistical analyses as a separate Supplementary Table, and request that source data for all independent repeats are provided when representative experiments of multiple independent repeats, or averages of two independent experiments are presented. This supplementary table should be in Excel format, with data for different figures provided as different sheets within a single Excel file. It should be labelled and numbered as one of the supplementary tables, titled “Statistics Source Data”, and mentioned in all relevant figure legends.

Author Rebuttal to Initial comments

Reviewer #1

Remarks to the Author:

In their manuscript entitled “Enhancer grammar of liver cell types and hepatocyte zonation states” Bravo González-Blas et al. uncover enhancer-driven gene regulatory networks (eGRN) that are involved in liver zonation by combining single cell multiomics, spatial omics, GRN inference, and deep learning. Specifically, the authors performed combined single cell RNA- and ATAC-seq, as well as spatial transcriptomics on the mouse liver and found that cell type identity and cell states (e.g. zonation) are encoded by the transcriptome and chromatin accessibility. Applying a tool called SCENIC+, Bravo González-Blas and colleagues identify several core general hepatocyte transcription factors, including Hnf4a, Hnf1a, Cebpa, Onecut1, Foxa1 and Nfib, and importantly, highlight Tbx3 and Tcf7l1 as key repressors of periportal and pericentral gene expression, respectively. The authors then performed a Massively Parallel Reporter Assay (MPRA) to determine enhancer sequence activity and found that around 40% of the accessible regions in hepatocytes are active. By training a hierarchical Deep Learning model, called DeepLiver, the authors provide a tool to decode and predict hepatocyte enhancer accessibility, activity, and zonation. Finally, the authors performed computational validation of the zoned transcription factors by simulated KD and OE, but also experimental validation by MPRA assays, thus, highlighting the importance of Tbx3 and Tcf7l1/2 binding sites within hepatocyte enhancers to drive zonation, while Hnf1a and Hnf4a binding was crucial for enhancer activity.

I really enjoyed reading this exciting manuscript that is of high importance, well written, has beautiful and informative illustrations and Figures, elegant bioinformatics tools, and uncovers interesting and novel aspects of mouse liver zonation regulation by eGRNs. However, some aspects of this work should be clarified/improved, as detailed below.

We thank the reviewer for the positive feedback and for critically assessing our manuscript. We have performed Tbx3 Chip-seq in the mouse liver and two additional single-cell multiome experiments in a non-starved mouse to assess the reviewer’s questions, and clarified and expanded our analyses.

Major comments:

1. Although the authors managed to create a convincing UMAP after removal of batch effects using Harmony, some conclusions are based on comparisons between cells that stem from different experimental conditions. For example, both multiome samples were collected

from starved mice. The snRNA and snATAC data are derived from mice that had continuous access to food. The authors noticed batch effects which they allocate to differences in circadian rhythm between these samples. However, it remains unclear if some of the observed differences could be rather attributed to the different experimental setups. The conclusion that certain GRNs depend on the animal's physiological state (including circadian rhythm) is therefore based on rather superficial data. The authors may consider removing this small part from their manuscript (page 6, lines 171-180, page 6, line 191). If the authors rather prefer to strengthen this idea, they should add samples from the same experimental setup (eg multiome) but derived from mice with distinct feeding procedures/from different times of the diurnal cycle (which could easily become a separate manuscript). Related to that topic: Some cell types are exclusively derived from one type of dataset (eg VECs only from Multiome vs LSECs only from snRNA/snATAC) at least according to Figure S7. The authors should comment on that and highlight this in the main text. More in general, an overall discussion about the limitations of their tools and experimental approaches should be included.

In our first version of the manuscript, we assessed the nature of the batch effects observed between unstarved and starved mice at the transcriptome and chromatin accessibility level:

1. Unsupervised enhancer clustering using topic modelling revealed two regulatory topics specifically enriched in hepatocytes from starved (topic 17) and unstarved (topic 75) mice, respectively. Two independent analyses link these topics to regulation of the circadian rhythm: 1) Gene Ontology analysis using GREAT showed enrichment for positive and negative regulation of the circadian rhythm (adjusted p-value = 10^{-19} and 10^{-7} , respectively, Fig S2) and 2) motif enrichment analysis reveals the presence of TFBS of the circadian rhythm TF Clock in topic 75, with a NES of 4.25. This suggests that regions specifically accessible in the two groups of samples are controlling the circadian rhythm genes.
2. At the transcriptome level, the circadian rhythm signatures described by Droin et al. (2021) can be used to classify the hepatocytes from the different samples. The classification of the samples agrees with the ones made based on enhancer clustering (Fig S3,S4).

Altogether, these analyses support that the batch effect is of biological nature (circadian rhythm) rather than a technical effect due to the experimental set-up. To further strengthen this hypothesis, we have performed two additional single-cell multiomics experiments on an unstarved mouse.

1. Unsupervised enhancer clustering using all data sets (with 100 topics) revealed again two regulatory topics specifically enriched in hepatocytes from starved (topic 72) and unstarved mice (topic 68), respectively (Fig R1.1). Two independent analyses link these topics to regulation of the circadian rhythm: 1) Gene Ontology analysis using GREAT showed enrichment for positive and negative regulation of the circadian rhythm (adjusted p-value = 10^{-4} and 10^{-9} , respectively) and 2) motif enrichment analysis reveals the presence of TFBS of the circadian rhythm TF Clock in topic 68, with a NES of 3.2.

Figure R1.1. Validation of circadian rhythm batch effect using additional multiome data based on enhancer clustering. *a.* Uncorrected snATAC-seq-based UMAP of 36,721 cells profiled by snATAC-seq or multiome (snRNA-seq+snATAC-seq) colored by cell type. *b.* Uncorrected snATAC-seq-based UMAP colored by sample (technique and mouse). *c.* Uncorrected snATAC-seq-based UMAP colored by topic probability using RGB encoding.

2. Circadian rhythm signatures from Droin et al. (2021) also classify cells from unstarved (ZT00) and starved mice independently of the experimental setup based on their transcriptome. In other words, the new multiome samples from an unstarved mouse are classified as ZT00, as the unstarved samples derived from independent single-cell omics experiments. The starved samples (multiomics), are classified at later ZT stages (Fig R1.2).

Figure R1.2. Validation of circadian rhythm batch effect using additional multiome data and publicly available signatures. *a.* Uncorrected snRNA-seq-based UMAP of 54,612 cells profiled by snRNA-seq or multiome (snRNA-seq+snATAC-seq) colored by cell type. *b.* Uncorrected snRNA-seq-based UMAP colored by sample (technique and mouse). *c.* Uncorrected snRNA-seq-based UMAP colored by AUC values for circadian rhythm signatures (Droin et al., 2021) using RGB encoding. *d.* Boxplot showing normalized AUC values across samples for circadian rhythm signatures. *e.* Standardized AUC values across samples for signatures on different circadian rhythm time points

We agree with the reviewer that further testing feeding procedures and sampling timepoints would be interesting; however, this point has been largely assessed by Droin et al. (2021) by scRNA-seq, from which we obtained the circadian rhythm signatures. While additional controlled time points would be needed for a stronger analysis, we believe that noting the impact of circadian rhythm and other mouse-specific batch effects in single cell experiments in the mouse liver (scRNA-seq, scATAC-seq, single-cell multiomics) is of relevance to other groups using these techniques in this system. Nevertheless, we agree with the reviewer, together with reviewer 3, that this is not the main contribution of the manuscript. **We have thus removed this section from the main text and describe it in a new Supplementary Note (Supplementary Note 1), where we further elaborate on the additional validation analyses we have performed (Supplementary Figure 5).**

With regards to the comment that some cell types are exclusively derived from one type of data set, we show in Fig S1 the actual percentages of each cell type across the samples. All cell types appear across all the experiments, with minor variations (Table R1.1). The most

notable effect is on the frozen sample, where the proportion of non-parenchymal cell types is higher compared to hepatocytes.

snRNA-seq	B_cell	BEC	cDC	Fibroblast	Hepatocytes	HSC	Kupffer	LSEC	MSC	pDC
snRNA_Fresh_Mouse-1	0.64	0.51	0.13	0.41	78.68	8.01	4.91	5.24	0.23	0.18
snRNA_Fresh_Mouse-2	0.71	1.08	0.33	1.11	67.91	12.31	6.16	7.79	0.07	0.12
snRNA_Frozen_Mouse-2	1.81	1.75	0.68	0.54	53.21	9.35	7.48	19.01	0.60	0.14
snRNA_Fresh_Mouse-3	0.69	0.21	0.26	0.18	76.76	5.28	6.29	7.74	0.45	0.16
Multiome-10x_Fresh_Mouse-4	0.10	0.57	0.20	0.11	86.64	3.43	4.21	4.16	0.06	0.08
Multiome-NST_Fresh_Mouse-5	0.12	0.20	0.16	0.14	87.78	7.38	1.28	1.68	0.08	0.24

scATAC-seq	B_cell	BEC	cDC	Fibroblast	Hepatocytes	HSC	Kupffer	LSEC	MSC	pDC
snATAC_Fresh_Mouse-6	1.11	1.27	0.54	1.35	72.00	8.32	4.56	8.86	0.05	0.08
snATAC_Fresh_Mouse-7	2.59	0.53	0.93	0.20	67.60	6.59	8.04	9.23	0.20	0.22
Multiome-10x_Fresh_Mouse-4	0.35	0.57	0.31	0.13	86.97	2.56	4.15	4.25	0.05	0.08
Multiome-NST_Fresh_Mouse-5	0.17	0.14	0.12	0.12	87.75	7.96	1.32	1.44	0.05	0.17

Table R1.1. Percentage of cell types in the different samples based on snRNA-seq or scATAC-seq annotations (top and bottom, respectively).

With regards to describing the limitations of the study, we now address this topic in the discussion.

2. Although this is a resource manuscript, *in vivo* validation showing that Tbx3 and Tcf711 act as key repressors controlling liver zonation should be performed. Besides validating this interesting finding, this would also highlight overall translational value of the resource data. The analyses performed by the authors are based on chromatin accessibility and expression, and although this can be a strong indicator, only ChIP assays allow a proper statement on target regions/binding. The authors could perform a Cut&Run approach (or any other similar preferred sequencing approach).

To further validate the role of Tbx3 as a key repressor of liver zonation, we have performed a ChIP-seq experiment *in vivo*. We predicted that Tbx3 (expressed pericentrally) binds to periportal enhancers, and this represses periportal genes in the pericentral hepatocytes. In agreement with this prediction, Tbx3 ChIP-seq signal is stronger in periportal hepatocyte regions compared to general and pericentral hepatocytes regions (Fig R2.1a). Using MACS2, we identified 23,951 peaks (q -value < 0.01), out of which 19,812 overlap with regions accessible in the mouse liver. Out of these, 4,748 overlap with (shared) hepatocyte specific peaks, and 20% of the periportal hepatocyte regions overlap with the Tbx3 ChIP-seq peaks (compared to 4% of the pericentral regions). In addition, 35% of the regions in

the Tbx3 regulon inferred by SCENIC+ overlap with these regions, and we find a strong enrichment of the Tbx3 motif in the top 1,000 Tbx3 ChIP-seq regions (NES 4.71, corresponding to 424 regions), together with other hepatocyte TFs such as Hnf4a, Cebpa, Foxa1 and Onecut1. Motif enrichment in periportal regions with Tbx3 ChIP-seq signal also reveals a stronger enrichment of Tbx3 (NES 6.32), and overall lower enrichment of other general hepatocyte TFs (Table R2.1).

		PP	Tbx3 ChIP-seq	PP \cap Tbx3 ChIP-seq
Tbx3		5.87	4.71	6.21
Hnf1a		5.51	5.73	7.50
Cebpa		5.04	9.42	4.90
Hnf4a		4.90	4.28	3.02
Onecut1		4.68	5.62	7.88
Foxa1		3.78	5.70	4.42

Table R2.1. Cistarget motif enrichment in periportal and Tbx3 ChIP-seq regions and their overlap. Values indicate cisTarget NES score in the different regions sets for the indicated motifs (top motifs found in periportal regions).

In agreement with our original observation, these regions are more accessible in periportal hepatocytes compared to pericentral (Fig R2.1b,c). Altogether, these observations support the role of Tbx3 as a repressor, as its binding in pericentral hepatocytes correlates with a decrease in accessibility. **We have added these data in Figure 2 and Extended Data 5, and added the Tbx3 ChIP-seq in the coverage plots in Figure 4.**

Figure R2.1. Assessing *Tbx3* binding on hepatocytes using in-house generated ChIP-seq data. *a.* Coverage plot showing *Tbx3* ChIP-seq coverage on pericentral and periportal hepatocyte regions. *b.* Coverage plot showing hepatocyte coverage on the top 1K *Tbx3* ChIP-seq regions. *c.* Pseudobulk accessibility profiles, ChIP-seq coverage (for *Hnf4a*, *Cebpa*, *Foxa1*, *Onecut1* and *Tbx3*), SCENIC+ region to gene links colored by correlation score and gene and *Tbx3* expression across the zoned hepatocytes classes (from PP to PC) are shown. The gene loci showed are *Cdh1* (Figure 4), *Hal* and *Tcf7l1*.

Regarding *Tcf7l1*, we performed a ChIP-seq experiment using a goat anti-Tcf3 antibody (Santa Cruz Biotechnology, sc-8635). However, the antibody was not ChIP-grade quality and the ChIP-seq was not successful (resulted in background signal). Thus, we have not included this data in the manuscript. In addition, we have not found any public *Tcf7l1* ChIP-seq data set in hepatocytes (only in nephron progenitor cells (Guo et al. 2021), and stem cells (De Jaime-Soguero et al., 2017; Sierra et al., 2018, Mukherjee et al., 2022)).

3. To validate their findings in human, the authors perform LOF of *Tbx3* in HepG2 cells using bulk MPRA. However, HepG2 cells cannot be used as an appropriate model for zonation in humans, even if they express *Tbx3*. Hepatocellular carcinoma, from which HepG2 cells are derived, generally show loss of proper liver zonation, and cells in culture in general anyway lose zoned gene expression. HepG2 cells should therefore not be used to study mechanisms related to liver zonation. These experiments and the corresponding

conclusions should be completely removed from the MS since they might be misleading in this context. The manuscript is also very exciting without this data.

In this manuscript, we have only used HepG2 to assess hepatocyte enhancer functionality, using MPRA and luciferase assays, not to explicitly validate the regulatory network of the zonation states. HepG2 expresses several hepatocyte master regulators from hepatocytes, such as Hnf4a, Cebpa, Hnf1a, Tcf7l2, Foxa1 and Onecut1, among others (Fig R3.1). HepG2 has been used to validate hepatocyte enhancers by several groups (Patwardhan et al., 2012, Inoue et al., 2017, Ersnt et al., 2016, Klein et al., 2020). In fact, Smith et al. (2013), performed MPRA experiments in the mouse liver and HepG2 using a synthetic library of enhancer formed by combinations of binding sites of 12 hepatocyte TFs (AHR/ARNT, CEBPA, FOXA1, GATA4, HNF1A, HNF4A, NR2F2, ONECUT1, PPARA, RXRA, TFAP2C and XBP1), finding a strong correlation (0.81) between the measurements in the two systems. In addition, HepG2 allows to perform more sensitive enhancer assays that are not feasible in vivo (i.e. luciferase assays). Altogether, we believe that HepG2 is a good *in vitro* model for testing hepatocyte enhancer activity, as long as the TFs controlling these enhancers are present in HepG2.

Due to a mutation in beta-catenin, WNT signaling is active in HepG2. It has also been reported in literature that HepG2 exhibits a more pericentral identity (Ardisasmita et al. 2022). HepG2 expresses Tbx3 and other pericentral markers, while periportal genes are not expressed or very lowly expressed, as observed in pericentral mouse hepatocytes (Fig R3.1).

Figure R3.1. Gene expression in HepG2 and pericentral, intermediate and periportal mouse hepatocytes. a.-d. Normalized expression levels of selected genes in HepG2 (a.), mouse pericentral hepatocytes (b.), mouse intermediate hepatocytes (c.), and mouse periportal hepatocytes (d.). Genes encoding transcription factors are highlighted in bold.

In fact, while we observe a significant correlation between the experiments performed in HepG2 and *in vivo* (0.5), it is lower than the one observed by Smith et al. (2013). The strongest difference we observe is for zoned enhancers, especially those containing Tbx3 sites (20% and 5% of the regions are active *in vivo* and in HepG2, respectively). In addition, we see that destruction of Tbx3 binding sites in hepatocytes enhancers (with binding sites for other hepatocyte TFs expressed in HepG2) can restore its activity, as measured by the luciferase assay. However, despite WNT activation, we agree with the reviewer that HepG2 does not completely recapitulate the pericentral identity. For instance, out of the 3,939 pericentrally (and pericentral-intermediate) zoned regions found in the mouse liver and conserved in the human genome, only 1,258 are accessible in HepG2 (Fig R3.2). Nevertheless, more hepatocyte-specific regions are accessible in HepG2 than in other mouse hepatocyte cell lines like Hepa1-6 and AML12, where not all TFs are expressed. We elaborate further on this topic on comment E of reviewer 3.

Figure R3.2. Coverage of general, pericentral and periportal mouse hepatocyte regions across systems. a.-c. Coverage plot on HepG2 (a.), Hepa1-6 (b.), and AML12 (c.) ATAC-seq. **d.** Coverage plots on mouse hepatocytes scATAC-seq pseudobulk, ordered by cluster from periportal to pericentral.

Altogether, taking into account its limitations as a model system, we believe that HepG2 is appropriate to study enhancers with binding sites of TFs that are expressed, such as Hnf4a, Foxa1, Cebpa, Onecut1, Hnf1a and Tbx3. In fact, throughout the manuscript, we do not use HepG2 as a zonation model, but to validate the activity of hepatocyte enhancers for which TFs are also present in HepG2.

We believe we do not overstate the results in HepG2 for the MPRA data, as experiments have been performed both in HepG2 and the mouse liver and conclusions are predominantly based on the mouse liver MPRA experiments. The luciferase activity assay, which can only be performed *in vitro*, is complementary and helps to validate Tbx3 as a repressor. For that reason, we would prefer to keep it in the paper. In conclusion, we believe that the reader can assess the validity of this particular data point, as the key message is derived from the *in vivo* data, not from HepG2. We believe that the HepG2 data supports the role of Tbx3 as a repressor, points towards the conservation of the enhancer code on a human system, and overall strengthens our manuscript. **We now address this topic in Supplementary Note 3 and Supplementary Figure 7-8.**

4. The authors state that Tbx3 and Tcf7l1 binding sites are located predominantly within hepatocyte enhancers, and in close proximity to binding sites of core transcription factors. It would be interesting to validate this predicted proximity between these repressors and the core factors, eg by a proximity ligation assay.

Indeed, we describe that Tbx3 and Tcf7l1 binding sites are located within hepatocyte enhancers, together with other binding sites of key hepatocyte transcription factors such as Hnf4a, Onecut1, Cebpa, Hnf1a and Foxa1. While we find this topic very interesting and now **address it in the discussion**, proximity ligation assays are technically challenging, and we believe that describing this mechanism is out of the scope of this paper given all the other contributions of the manuscript. In literature, TBX3 has only been tested *in vitro* with this technique (HEK293T cells) to assess its proximity to its cofactor BCL9 (Zimmerli et al., 2020). In addition, we do not predict Hnf4a, Onecut1, Cebpa, Hnf1a and Foxa1 as cofactors of Tbx3, but rather as competitors at certain target regions.

To further investigate TF co-binding *at the bulk level*, we have compared our newly generated Tbx3 ChIP-seq in the mouse liver with publicly available ChIP-seq data of Hnf4a, Onecut1, Foxa1 and Cebpa. We have found a strong overlap between the ChIP-seq data sets (Fig R4.1). For instance, from the top 1,000 Tbx3 ChIP-seq regions, 742 overlap

with the ChIP-seq peaks of Hnf4a, Foxa1, Cebpa and Onecut1. Additionally, 117 regions overlap with Hnf4a and Foxa1 ChIP-seq peaks exclusively (Fig R4.1b). As an illustration we show the periportal gene Hal locus with two enhancers bound by Tbx3. In both cases we observe ChIP-seq signal of Hnf4a, Cebpa, Foxa1 and Onecut1, with Cebpa and Onecut1 stronger on the left one and Hnf4a on the right one (Fig R4.1c). Importantly, we also find enrichment of Hnf4a, Cebpa, Foxa1 and Onecut1 in the Tbx3 ChIP-seq regions, as described in comment 2 (with NES 4.28, 9.42, 5.70 and 5.62, respectively; Table R2.1).

Figure R4.1. Comparison of Hnf4a, Cebpa, Foxa1, Onecut1, and Tbx3 ChIP-seq data.
a. Coverage plot on the top 1,000 Tbx3 ChIP-seq regions on Hnf4a, Cebpa, Foxa1, Onecut1, and Tbx3 ChIP-seq data. **b.** Overlap between Hnf4a, Foxa1, Cebpa and Onecut1 on the top 1,000 Tbx3 regions. **c.** Pseudobulk accessibility profiles, ChIP-seq coverage (for Hnf4a, Cebpa, Foxa1, Onecut1 and Tbx3), SCENIC+ region to gene links colored by correlation score and Hal and Tbx3 expression across the zoned hepatocytes classes (from PP to PC) are shown.

Note that ChIP-seq is performed at bulk level, so we cannot assess if the TFs co-bind in the same cells. While Tbx3 is expressed and bound in pericentral hepatocytes (repressing ATAC-seq signal), the other TFs generally expressed in hepatocytes may be bound to those regions only on periportal hepatocytes, rather than all hepatocytes. The ChIP-seq experiment nevertheless provides a useful addition, showing that Tbx3 and the general hepatocyte TFs can bind to the same enhancers (not necessarily at the same time point and on the same allele in the same cell; note that in this case, the proximity ligation assay would

be negative). Further elucidation of these specific biochemical interactions is outside the scope of this manuscript.

5. Last year it was published (PMID: 34129813) that hepatocytes have an open chromatin configuration for both periportal and pericentral genes, regardless in which zone they reside. The work by Bravo González-Blas et al. provides important insights into how these genes may be repressed to confer zonation but does not mention how their findings relate to such previous work. In addition, the authors did not cite any of the seminal papers by the Zaret lab, dissecting the epigenome/GRNs defining hepatic cell identity. While the authors are certainly experts in GRNs/multi-omics and cool bioinformatics, and adequately cite such work, they should be more inclusive in discussing existing liver literature.

With regards to the analysis presented by Sun et al (2021, PMID: 34129813), the authors show the ATAC-seq profile centered at the promoter of pericentral and periportal genes, respectively. Indeed, if we look at the accessibility at the *promoter* of pericentral and periportal genes in our data, we also observe less differences compared to distal regions (Fig R5.1).

Fig R5.1 Chromatin accessibility in distal zoned regions and at the TSS of zoned genes. a. Coverage plots on distal zoned pericentral and periportal regions on mouse hepatocytes scATAC-seq pseudobulk, ordered by cluster from periportal to pericentral. **b.** Coverage plot centered on the TSS of zoned genes linked to distal zoned regions on mouse hepatocytes scATAC-seq pseudobulk, ordered by cluster from periportal to pericentral.

We believe this approach is not adequate, as promoters tend to be ubiquitously accessible, and specificity is encoded by distal enhancer regions. Thus, the authors show that the open chromatin configuration at the promoter is the same regardless of zonation, but do not assess differential distal chromatin differences. We have added this analysis as Extended Data 5, referring to the work of Sun et al. (2021). We now include references to the work of the Zaret lab.

6. Spatial positioning of cells from sc/sn profiling studies is a central tool for studying zoned effects in the liver. It would be important that the authors discuss how their approach differs from what the Itzkovitz lab has developed.

In comparison to previous work from the Itzkovitz lab and others, which mostly focused in single cell transcriptomics rather than scATAC-seq, the main contribution of our manuscript is the analysis of chromatin accessibility and enhancer logic related to liver zonation. To our knowledge, this work is the first to provide a comprehensive view of the (enhancer) gene regulatory networks and the enhancer logic that underlies liver zonation, showing that specific mutations on enhancer allow to shift zonation patterns of enhancer activity. We further stress this point in the discussion.

Minor comments:

1. In general, all figures are too small. Especially, the font size makes many figures hard to read.

We have adjusted our figures to the Nature Cell Biology guidelines, making sure that there are no font sizes below 5pt.

2. Fig 1e: It is not clear to which lines exactly the highlighted genes are assigned to. Can the authors clarify this in the figure?

We have added this information to the figure legend.

3. Fig 2b-e: figure legends missing

The legends for these panels are included in the figure legend.

4. In Figure S7b it looks like mouse 1 was the CD-1 strain, however, in the methods section the authors state that this is mouse 3.

Mouse 1 was CD-1 strain. We have corrected this in the methods section.

5. In FigS9/S10 all legends/scales are missing for color coding. Also, for Figures S16a. We have added the corresponding legends.

6. Typo line 236: “assess” to “assessed”

We have fixed the typo.

7. Typo line 476: “FAC” to “FACS”

We believe that FAC-sorted is correct as FACS stands for Fluorescence-Activated Cell Sorting. FAC-sorted is used in several publications, but we have not found any using FACS-sorted.

8. Typo line 587: “;” to “.”

We have fixed the typo.

9. Figure S19 is blurry.

We have improved the readability and quality of all figures.

10. Can the authors state how they defined the terms “promoters” and “enhancers”? Regulatory elements may have both enhancer and promoter functions and distinct factors may determine these activities (see <https://doi.org/10.1038/s41576-019-0173-8>). In this regard, the authors should discuss the limitations that comes with their chosen MPRA assay.

In our MPRA library, we define promoters as chromatin accessibility regions located +/- 3,000bp from a gene TSS. In our MPRA experiments we assess the capacity of the sequences to activate a minimal promoter (i.e. enhancer activity), rather than their activity as promoters. We have clarified this in the methods section.

11. Page 6, line 182: the Halpern et al paper is not a correct reference here. This is “just” a scRNAseq resource paper without functional validation. The mechanistic role of these pathways in zonation have been published earlier elsewhere. Better use a recent review article about liver zonation that covers the original papers or cite them directly.

We have cited Halpern et al. because we have used the curated signatures provided in that manuscript as supplementary file. We have added the original references referred by Halpern et al. in Supplementary Note 1.

Reviewer #2:

Remarks to the Author:

In the manuscript entitled “Enhancer grammar of liver cell types and hepatocyte zonation states”, Bravo and colleagues generated single-cell multiomics (scRNA-seq and scATAC-seq) and spatial omics data (single molecule FISH) to reveal gene regulatory network across mouse liver cell types. They found that zonation states of the liver are regulated by transcription factors (Hnf4a, Cebpa, Hnf1a, Onecut1 and Foxa1) and repressors (Tcf7l1 and Tbx3). Furthermore,

they performed in vivo massively parallel reporter assays (MPRAs) to examine >10,000 candidate regulatory elements for their enhancer activity in mouse liver and HepG2, and identified 2,913 active enhancers. They developed a deep learning model (DeepLiver) to dissect the function of these TF binding motifs and predict their regulatory grammar. Their omics approach associating gene regulation and spatial information, as well as machine learning approach based on in vivo MPRA that allows to reveal regulatory code at base pair resolution, are impactful broadly in the gene regulatory genomics field. Their datasets, computational tools, and browsers (Scope and UCSC genome browser) are robust and useful as resources in the research community.

We thank the reviewer for these positive comments on our manuscript.

Minor comments:

1. They termed a TF with its set of predicted target enhancers and regions “eRegulon” (lines 238-239) but used “regulons” instead in the following sentences (e.g., line 240, “This analysis revealed 180 regulons”). Please check if these should be “eRegulons” or not.

Indeed, the correct word is eRegulon (*enhancer_Regulon*). **We have updated the term where relevant.**

2. MPRA reproducibility (Figure S14a, b) was not clear to me. Why 3’ mouse plasmids were not reproducible between replicates, while 5’ mouse plasmids reproducible? Please add some explanation about how the MPRA data are reproducible in the text.

The 5’ experiments were both performed in vivo in mice, while for the 3’ experiments we are comparing experiments performed in mice versus those performed in HepG2. Within the replicates for each experiment (5’ mouse experiment 1, 5’ mouse experiment 2, 3’ mouse and 3’ HepG2), we observe high reproducibility (0.82-1, Extended Data 6a). When comparing the 3’ experiments in mouse and HepG2 we observe a correlation of 0.48 (Extended Data 6b). The correlation between the two 5’ experiment in mice is 0.69. We believe that the differences observed in the 3’ library is largely due to biological differences between HepG2 and the mouse liver. For instance, Tbx3 regions show less activity in HepG2 (20% and 5% active in vivo and HepG2, respectively). Tbx3 is expressed in HepG2, while in the mouse liver is only expressed by pericentral hepatocytes.

These regions are repressed in HepG2 and pericentral hepatocytes by Tbx3, but not in periportal hepatocytes. Thus, these regions are more active in vivo compared to HepG2. We have modified the legend of this figure to stress that both 5' experiments are done in vivo, while one of the 3' experiments is done in vivo and the other in HepG2, respectively.

3. I am curious whether DeepLiver is useful for de novo functional motif discovery. For example, in figure 4c, around the 310bp region in the Cdh1 enhancer seems to associate negatively with the accessibility but not highlighted here. Does this sequence overlap with any known TF motifs? Any other regions that are potentially interesting as functional motifs found in this analysis?

Indeed, that sequence of nucleotides overlaps with the PU.1 motif. The predictions of the first model of DeepLiver are shown on top of Fig 4c, where the sequence is observed. This model learns to classify enhancers based on their accessibility across cell types in the mouse liver. On top, we show the relevance of the nucleotides to make the region specifically accessible in hepatocytes (belonging to topic 43 in our enhancer topic model). As PU.1 is a common pattern in enhancers accessible in immune cells (e.g. Kupffer cells) the presence of the motif does not contribute to make the region specifically accessible in hepatocytes. In the figure we highlight the patterns that are relevant to make the regions specifically accessible in hepatocytes, zoned, and active. We have clarified the figure to stress this point, by naming the track as hepatocyte-specific instead of topic 43, and clarifying the legend of the figure.

Reviewer #3:

Remarks to the Author:

A. Summary of the key results

The manuscript of C.B. Gonzalez-Blas represents a complex systems biology paper representing the spatial multiomics atlas of the mouse liver sinusoid, with characteristic zonation from periportal to pericentral regions, at the single cell level. The major added value is the prediction and validation of cell-type specific gene regulatory networks (eGRNs) through analysis of active enhancers, and also the cell state specific eGRNs that depend also on the cell location. The massive parallel reporter assay (MPRA) in vitro and in vivo aided in defining

active regulons that are characteristic for each cell type or for groups of cells. A DeepLiver deep learning model was applied to validate experimental data, especially to predict enhancer accessibility and activity, as well as zonation state of a cell. An interesting observation is that in the mouse more distal enhancers are active compared to promoters, while in HepG2 immortal cells situation was the opposite. The authors propose that either distal enhancer activation is more difficult in HepG2 cells, or that distal enhancers are less conserved than promoters between mouse and human.

While it is not completely novel that the cell state changes in transcription and chromatin accessibility in hepatocytes, liver sinusoidal endothelial cells and hepatic stellate cells depend on zonation, the novelty lies in determining transcription repressors (by eGRN mapping) that define the periportal zonation (Tbx3), and pericentral zonation (Tcf711). Tbx3 is a transcriptional repressor essential during early embryonic development, in the formation other organ systems, and in tumorigenesis while Tcf711 predominantly acts as a repressor of Wnt target gene expression, (while together with Lef1 can act as transcriptional activator). The five transcription factors that were determined to control the core hepatocyte gene regulatory networks (Hnf4a, Cebpa, Hnf1a, Onecut1 and Foxa1) were confirmed also in validation experiments where data from MPRA were applied to train the DeepLiver model. The above transcription factors were identified as drivers of enhancer specificity. It is interesting that Tbx3 and Tcf711 expression profiles are anti-correlated with the accessibility of their potential target regions, i. e. Tbx3 is expressed only in pericentral hepatocytes, while its candidate target regions are only accessible periportally. Novel is also the finding that Tbx3 and Tcf711 repress each other and in this manner control the zonation of downstream gene expression.

B. Originality and significance

Previous published single-cell and spatial transcriptomics studies have shown that not only hepatocyte function, but also the transcriptome, varies along the periportal-pericentral liver lobule axis, described also as zonation. The novelty of this paper is to elucidate how zonation interacts with the gene regulatory networks (GRNs) of hepatic cells and to apply single cell data to predict whether a regulatory region is active. This is a significant work and might represent a good data resource for the liver scientists interested in zonation of the liver metabolism, where not only the cell type but also location of the cells defines its metabolic state, described herein by multiomics single cell data. The challenge of how to infer GRNs from single cell data to predict whether a regulatory region is active was also solved in this

work by combination of experimental and deep learning modelling, that allowed also mutagenesis in silico.

A drawback is to investigate the only the male gender – HepG2 cells are of male origin (HepG2), and only male mice were used. The data and conclusions should thus not be generalized for both sexes. Liver is, after the gonads, the most sexually dimorphic organ (Lefebvre P, Steals B, Nature Endocrinol, 2021, <https://www.nature.com/articles/s41574-021-00538-6>, Cvitanovic et al., Hepatology 2017 <https://pubmed.ncbi.nlm.nih.gov/28520105/> and other relevant references). Vandel J et al., Hepatology 2021 (<https://aasldpubs.onlinelibrary.wiley.com/doi/10.1002/hep.31312>) recently described how the large-scale analysis of transcriptomic profiles from human livers emphasized the sexually dimorphic nature of NASH (nonalcoholic steatohepatitis) as a liver disease state and its link with fibrosis. They call for the integration of sex as a major determinant of liver responses to liver disease progression and the responses to drugs.

We agree with the reviewer that the impact of sexual dimorphism on the mechanism we describe is a key point. To address this, we have analyzed additional publicly available snRNA-seq data from wild-type male and female livers (Goldfarb et al., 2022). After quality control (see *Methods*), this data set contains 4,860 liver cells from 3 different male mice and 5,342 cells from 3 different female mice (Fig RB1a-b). Importantly, neither the expression of the core hepatocyte TFs we describe in this manuscript nor their targets (measured as the AUC enrichment of the SCENIC+ eRegulons) are affected by gender (Fig RB1c-d). In other words, the expression of Tbx3, Tcf7l1, Hnf4a, Hnf1a, Cebpa, Foxa1, Onecut1 and their targets is comparable between male and female mice.

We performed differential gene expression based on gender for all the cell types in the liver, and identified 56 and 65 genes upregulated in male and female hepatocytes, respectively (LogFC > 0.75, adjusted p-value < 0.05, Fig RB1e). Only 1 TF was upregulated in male hepatocytes, Bcl6, while 4 TFs were upregulated in female hepatocytes, namely Rfx4, Cux2, Esr1 and Esrrg. Except for Rfx4, these TFs have been previously reported to regulate sex differences in the mouse liver (Meyer et al., 2009, Conforto et al., 2012, O'Brien et al., 2021). Gene Ontology analysis of these differentially expressed genes points to metabolic differences between male and female hepatocytes, with genes related to lipid metabolism upregulated in females (adjusted p-value: 10^{-7} , Fig RB1f-g). Importantly, none of these genes have been found as Tcf7l1 or Tbx3 targets.

Altogether, this independent analysis suggests that the core mechanism we describe in our manuscript is not affected by sexual dimorphism. Nevertheless, additional single-cell

multiomics data in female livers and/or disease models may be needed in the future to study sex-biased gene regulation in wild-type mice and upon disease at the enhancer-GRNs resolution. We have added a new Supplementary Note (Supplementary Note 1, Supplementary Figure 6) with these analyses, refer to these results in the text, and include this aspect as a limitation of the study in the discussion.

Figure B1. Impact of sex dimorphism in gene expression in the mouse liver. a.-b. UMAP of 4,860 liver cells from 3 different male mice and 5,342 cells from 3 different female mice colored by cell type (a.) and by gender (b., male: blue, female: pink). **c.** UMAP from male and female livers colored by gene expression or eRegulon enrichment. **d.** Normalised gene expression distribution across female and male hepatocytes for selected genes. **e.** Barplot reporting the

number of differentially expressed genes between male and female livers across cell types f.-g. STRING network based on the genes upregulated in male (f.) and female (g.) hepatocytes.

C. Data & methodology: validity of approach, quality of data, quality of presentation

The paper is relatively easy to read despite comprehensive methodology. Experimental and computational approaches are appropriate for such complex questions. The methodology is up-to date. The number of replicates is stated. I focus on main experimental approaches and their presentation. In brief:

- From snRNAseq (10x) and multi-ome gene expression 14 hepatic cell types were identified.
- The epigenome analysis on scATAC-seq samples was performed.
- E-regulons represent a crucial part of the manuscript. Gene-based and region-based regulons were scored based on other datasets that led to 180 high quality regulons.
- smFISH image analysis resulted in identification of 19 clusters with 11 cell types that were annotated based on marker gene expression with a panel of 100 selected genes across cell types and cell states in the liver that represents a crucial reagent.
- Hi-C and Chip-Seq publically available data were used to validate the regulons.
- MPRA was applied to measure the regulatory function of DNA sequences.
- Downstream analyses included the pseudotime order which represents the distance along the portal-central axis and identifies numbers of genes and regions in hepatic stellate cells (HSC) Liver sinusoidal endothelial cells (LSEC) and in hepatocytes (that hold about 10 times more genes and 20 – 40 times more regions compared to the other two cell types). Regulons are then stratified by zonation and the sample by PCA.

It is not clear why the authors included the circadian rhythm signatures from Droin et al. (2021) - this part is not well elaborated and I could not find the link to the rest of analysis, especially to zonation.

We believe that noting the impact of circadian rhythm and other batch effects in single cell experiments in the mouse liver (scRNA-seq, scATAC-seq, single-cell multiomics) is of relevance to other groups using these techniques in this system. However, we agree with this reviewer, together with reviewer 1, that is not the main contribution of the manuscript. We have thus removed this section from the main text and describe it in a new Supplementary Note (Supplementary Note 1), where we further elaborate on the additional validation analyses we have performed (Supplementary Figure 5).

D. Appropriate use of statistics and treatment of uncertainties

I am not a computation specialist. From the approaches described, the computation is comprehensive, multi-level and statistically sound (adjusted p values) and up to date data integration techniques have been applied.

We thank the reviewer for assessing the statistics used in our manuscript.

E. Conclusions: robustness, validity, reliability

Conclusions based on experimental data and computational predictions are largely concordant. Data were proven from different angles. DeepLiver proved to be a good prediction tool. A bit confusing is the use of human HepG2 cells and not a mouse hepatic cell line, such as Hepa1. A mouse cell line would make it easier to answer some questions, also if the distal enhancer activation is more difficult in immortalized hepatic cell lines compared to the cells in the liver – a very important question to be addressed. Translation to the human remains difficult for the topic of zonation. Data from 2D cell cultures are useful but with limiting biological relevance that we have to acknowledge. On top of that is also the importance of the liver sex/gender that was not even mentioned in the manuscript.

In this manuscript, we have only used HepG2 to assess enhancer functionality, using MPRA and luciferase assays. HepG2 has been used to validate hepatocyte enhancers by several groups (Patwardhan et al., 2012, Inoue et al., 2017, Ersnt et al., 2016, Klein et al., 2020). In fact, Smith et al. (2013), performed MPRA experiments in the mouse liver and HepG2 using a synthetic library of enhancer formed by combinations of binding sites of 12 hepatocyte TFs (AHR/ARNT, CEBPA, FOXA1, GATA4, HNF1A, HNF4A, NR2F2, ONECUT1, PPARA, RXRA, TFAP2C and XBPI), finding a strong correlation (0.81) between the measurements in the two systems. In addition, HepG2 allows to perform more sensitive enhancer assays that are not feasible in vivo (i.e. luciferase assays). Altogether, HepG2 is a good *in vitro* model for testing hepatocyte enhancer activity, as long as the TFs controlling these enhancers are present in HepG2. We refer to comment 3 from reviewer 1 for further analyses on HepG2 as a model to test core pericentral hepatocytes.

The suggestion by the reviewer to use mouse hepatocyte cell lines is interesting. To assess their relevance as model system (and if relevant, to use them for enhancer-luciferase assays), we re-used RNA-seq and generated new ATAC-seq data on two mouse hepatocyte cell lines, namely AML12 and Hepa1-6, besides HepG2. At the transcriptome level, HepG2 expresses several hepatocyte master regulators from hepatocytes, such as Hnf4a, Cebpa, Hnf1a, Tcf7l2, Foxa1 and Onecut1, among others. On the other hand, AML12 shows reduced expression of Cebpa, Foxa1, Hnf1a, and Hnf4a compared to HepG2; and Hepa1-6, of Cebpa and Foxa1 (Fig RE1).

Figure E1. Normalized gene expression of selected genes across systems, namely AML12, Hepa1-6, HepG2 and snRNA-seq pseudobulks of pericentral, intermediate and periportal hepatocytes. Transcription factors are highlighted in bold.

Since HepG2 was used to test enhancer activity, we compared the accessibility of the 12,000 enhancers library on the different cell lines. In this library we included shared regions (accessible in hepatocytes and other cell types, of which 56% are promoters) and hepatocyte-specific regions (generally accessible and zoned). We observed that shared regions were largely accessible across the 3 cell lines, while HepG2 showed more accessibility in hepatocyte specific regions compared to the other cell lines (Fig E2).

Figure E2. Coverage of CHEQ-seq library regions across systems. a.-d. ATAC-seq coverage plot on HepG2 (a.), Hepa1-6 (b.), AML12 (c.) and mouse hepatocytes pseudobulk, ordered by cluster from periportal to pericentral (d.).

To further assess differences at the enhancer activity level, we performed an MPRA experiment in AML12. As expected from the chromatin accessibility profiles, only positive controls and shared regions were significantly active compared to the negative control (Fig E3).

Figure E3. Enhancer activity LogFC across systems for the 12,000 enhancers library. a.-c. MPRA Log₂ Fold-Change boxplots per enhancer class (top) and eRegulon (bottom) in HepG2 (a.), in vivo (b.), and in AML12 (c.). The asterisks indicate the significance compared to shuffle (****: p-value <= 0.0001, ***: p-value <= 0.001, **: p-value <= 0.01, *: p-value <= 0.05, ns: not significant). G: General, PP: Periportal, I: Intermediate, PC: Pericentral.

Altogether, we believe that HepG2 is a more suitable model system to study enhancers following the enhancer code we are depicting, regulated by Hnf4a, Foxa1, Cebp, Onecut1, Hnf1a and Tbx3. These TFs are expressed in HepG2, but not all of them are expressed in Hepal-6 and AML12 at similar levels. We now address this topic in a new Supplementary Note (Supplementary Note 3, Supplementary Figure 7-8).

Regarding the impact of sex dimorphism, we have extensively assessed this topic in comment C.

F. Suggested improvements: experiments, data for possible revision

The authors promise availability to explore the resource at http://scope.aertslab.org/#/Bravo_et_al_Liver, however, the datasets and their visualisation seem not to be yet available publically (maybe we can see them for review purposes?). At the UCSC Genome Browser²⁷ (https://genome.ucsc.edu/s/cbravo/Bravo_et_al_Liver) one can

find the useful Chip-Seq data with transcription factor binding sites in different liver cell types, but it was not clear to me how this relates to zonation - some explanatory sentences are lacking for broader understanding.

The resource at http://scope.aertslab.org/#/Bravo_et_al_Liver is available, and **we now include a full description of the files there in Supplementary Note 5.** The UCSC resource is also available, but during the revision period there was a migration of the data which may have affected the exploration of the reviewer. **We include a detailed description of the files in this session in Supplementary Note 6.**

The epigenome analysis on scATAC-seq samples is written in a less understandable manner. It is not clear in which step this data were integrated to provide finally the active regulons which should show also open chromatin structures?

Regulatory topics and Differentially Accessible Regions (DARs) derived from the scATAC-seq data are used to derive TF cistromes, that is, sets of regions in which the TF motif is present. These TF-region relationships are integrated with region-gene and TF-gene links derived by SCENIC+ using a leading-edge approach to form the final *enhancer-Regulons*. **We have clarified this in the Methods section, and also refer to the SCENIC+ manuscript for further details (Bravo & De Winter, 2023).**

Table S2. 12K MPRA metadata and measurements – there are multiple blank lines in the column AN, “activity expression”. What is a difference between “none” or a blank section? From 12000 reporter probes, how many were not found active in vivo nor in vitro in HepG2 cells? How many were “blank”?

MPRA data was analyzed with DESEQ2, using the plasmid fractions as controls and the cDNA fractions as treatment. In the case of very low number of reads in the control or cDNA fractions for an enhancer, DESEQ2 set its adjusted p-value to NA. In other words, a blank section means that the enhancer was not detected/lowly detected (in the library after cloning and/or in the experiment) and we cannot assess its activity with confidence, while ‘None’ indicates that the enhancer was well covered in the plasmid and cDNA fractions but was not found differentially expressed in the cDNA fraction. In total, we could assess the activity of 7,198 enhancers with high-confidence, meaning that 4,802 were blank. Out of these 7,198 enhancers measured with high confidence, 4,285 were found not active in the mouse liver nor HepG2.

Supplementary notes

Chr1: 194610309-194610809 - TF labels are missing for: The enhancer accessibility only in hepatocytes, What makes the enhancer periportal and What makes the enhancer to be active. We have added the corresponding labels.

The circadian data should be either better embedded in the liver zonation cell-and –space fate story eGRNs, or removed. I could not find how it contributed to the overall conclusions. The authors claim that circadian rhythm was among the batch effect differences in hepatocytes based on physiological states of the mice and that (Topic 17 and Topic 75) from supplementary Figure S8 represent different phases of the circadian rhythm. How can this be justified? It is also not clear how the publicly available scRNA-seq data on the mouse liver during different phases of the circadian rhythm from Droin et al. 2021 contributed to better understand the enhancer grammar of liver cell types and hepatocyte zonation states (Figs.S9-S10). In light of this, the summary statement needs modification “In summary, our spatial single cell multiome atlas of the mouse liver reveals that both cell type identity and cell states, such as zonation and circadian rhythm, are congruently encoded at both the transcriptome and chromatin accessibility level.

We believe that noting the impact of circadian rhythm and other batch effects in single cell experiments in the mouse liver (scRNA-seq, scATAC-seq, single-cell multiomics) is of relevance to other groups using these techniques in this system. However, we agree with the reviewer, together with reviewer 1, that is not the main contribution of the manuscript. **We have thus removed this section from the main text and describe it in a new Supplementary Note (Supplementary Note 1), where we further elaborate in the additional validation analyses we have performed.**

We refer to reviewer 1 question 1 for more details on the additional analyses and explanations regarding this topic.

The title, abstract and conclusions should not be generalized to both sexes if only male cells and male mice have been used in experiments. The aspect of gender-sex should be at least mentioned and if possible, elaborated.

We have added a supplementary note and a supplementary figure regarding sex dimorphism (see comment B), refer to these results in the text, and include this aspect as a limitation of the study in the discussion.

G. References: appropriate credit to previous work?

Most referring is appropriate. I suggest the authors to discuss the zonation and specific cell markers in light of the paper by Inverso D. et al., *Developmental Cell* 2021, <https://doi.org/10.1016/j.devcel.2021.05.001>, where they combined spatial single cell sorting with transcriptomics and quantitative proteomics/ phosphoproteomics, to established the spatially resolved proteome landscape of the liver endothelium, enriching the mechanistic insight into zoned vascular signaling mechanisms. It would be interesting to learn to which extent are the zonation gene markers of C.B. Gonzalez-Blas related to gene markers of Inverso D et al. (Inverso et al., Results section: Spatial multiomics of the liver endothelium; Transcriptome zonation defines distinct L-EC signatures).

We now refer to the work from Inverso et al., in the text. Out of the 220 genes we identify as zoned along the LSEC porto-central axis, 151 overlap with genes identified by Inverso et al. Note that Inverso et al. markers are less stringent and they only compare endothelial cells (LSEC and VECs), finding 1,563 PC, 2,435 PP, 274 sinusoid and 616 vessel genes. We performed differential gene expression between pericentral and periportal LSEC, finding 612 and 566 marker genes, out of which 332 and 312 overlap with Inverso et al. markers. For VECs, we identified 138 and 166 pericentral and periportal marker genes, respectively; out of which 69 and 102 overlap with Inverso et al., markers.

H. Clarity and context: lucidity of abstract/summary, appropriateness of abstract, introduction and conclusions

The writing and the content of the chapters are appropriate. Specific remarks were listed above. We thank the reviewer for her assessment.

Prof. dr. Damjana Rozman

Reviewer #4:

Remarks to the Author:

In the study, González-Blas et al performed transcriptome and genomic accessibility profiling of mouse liver tissues at single-cell level. They found zonation patterns along the portal-central axis in hepatocytes, hepatic stellate cells, and liver sinusoidal endothelial cells in terms of gene expression, region accessibility and signaling pathways. They further utilized smFISH to validate the spatial variations detected from the single-cell analysis. The authors identified two

repressors, i.e., Tcf7l1 and Tbx3, as important regulators of the zonation states in hepatocytes using GRN inference and DeepLiver methods.

The zonation of liver has been well known. The authors applied new technologies to provide an insight into the TF regulatory network of the zonation in hepatocytes. Overall, 1) large datasets of single-cell multiome (snATAC+snRNA), snATAC, and snRNA of mouse liver tissue were provided in the paper. The datasets would be useful to the field. However, since those were derived from mice, the impact of the data would be limited compared with human data; 2) the two repressors of Tcf7l1 and Tbx3 identified in the manuscript have been reported in literature (Ben-Moshe et al Nature Metabolism, 2019; Brosch et al Nature Communications, 2018); 3) some of the conclusions in this paper were overstated and more evidence will be needed to support the statements.

We thank the reviewer for assessing our manuscript. We provide additional evidence to support our conclusions, including new sc-multiome data, ChIP-seq data against Tbx3. We also improved the discussion to better describe the novelty of our work compared to previous publications.

Major:

1. In Fig S1, the number of UMIs and genes vary significantly even within each individual cell type. Was normalization (e.g., sequencing depth normalization) properly performed for the data?

In Fig S1, we show the number of UMIs and genes **before** sequencing depth normalization. **We have added this to the figure legend.** Note that the number of expressed genes will not change despite sequencing depth normalization. Biologically, hepatocytes exhibit higher levels of gene expression compared to non-parenchymal cell types. The variability observed within a cell type can be due to biological heterogeneity (e.g. zonation in hepatocytes, HSC, and LSEC, cell cycle in a subset of Kupffer cells, activation levels) or technical effects (e.g. drop-outs).

With regards to the difference between the samples, we do not observe a relationship between lower coverage and sequencing saturation (Table R4.1). We believe that the differences between the samples are technical due to the protocol used (snRNA-seq or multiome). Overall, we observe that in the multiome samples there is a lower number of genes and UMIs recovered, which we find relevant to report. To further validate this point, we have downsampled the experiments to the same sequencing depth, observing that the variation between cell types and experimental runs is preserved (Fig R4.1). For the analysis, we have performed LogCPM.

normalization using the Scanpy workflow, a commonly used scRNA-seq analysis pipeline in the field.

Sample	Sequencing saturation
snRNA_Fresh_Mouse-1	43%
snRNA_Fresh_Mouse-2	30%
snRNA_Frozen_Mouse-2	31%
snRNA_Fresh_Mouse-3	70%
Multiome-10x_Fresh_Mouse-4	55%
Multiome-NST_Fresh_Mouse-5	48%

Table R4.1. Sequencing saturation per sample.

Figure R4.1. Distribution of number of UMIs, expressed genes and mitochondrial reads across downsampled experiments.

2. The authors observed an obvious disproportion of active regions in the enhancer/promoter regions between mouse and human hepatocytes, and further stated that ‘either that distal enhancer activation is more difficult to recapitulate in HepG2 cells, or that distal enhancers are less conserved than promoters between mouse and human’. Not enough evidence was provided

to support this conclusion. Only 7,198 valid regions were captured in total, which represents a small portion of the actual regulatory regions in vivo. Thus, this conclusion is not well supported due to insufficient coverage of the regulatory regions.

Given our data set, we observe less distal enhancers exclusively active in the HepG2 compared to the mouse liver (46 versus 64%), while more promoters are exclusively active in HepG2 compared to the mouse liver (54% versus 27%). This suggests that distal enhancer activation is more difficult to recapitulate in HepG2 cells. From the regions in the library, using liftover with a minimum of 50% of the bases conserved, out of the 2,561 promoters, 83% are conserved between human and mouse, while 68% of the 3,711 enhancers are conserved. This suggests that distal enhancers are less conserved. Nevertheless, since the statement is based in a subset of regulatory regions and not the full repertoire, **we have removed the conclusion from the text and just report the statistics.**

3. Similar issues for the conclusion stated in lines 324-326. In addition, only one human cell line was applied. There's a possibility that the observation may be derived from unknown bias. More evidence is needed to support the conclusion in this section.

The statement refers to the activity in the mouse liver, not HepG2. The library was tested in 7 different mice. We have replaced the sentence by: 'In summary, around 40% of **the tested** accessible regions in hepatocytes are active, and **the sequence of these enhancers** is not only predictive of enhancer accessibility, but also of enhancer activity.'

4. The authors found that Tbx3 and Tcf7l1 directly repress each other by using SCENIC+ eGRN. This is very interesting, but more supportive evidence is needed. For example, did the author check the public ChIP-seq data of TBX3 or TCF7L1 in hepatocytes. Does TBX3 directly bind to the enhancers of Tcf7l1, and vice versa?

We have generated Tbx3 ChIP-seq data in the mouse liver, and identified 9 regions linked to Tcf7l1 bound by Tbx3 (Fig R4.2). These regions are: chr6:72565432-72565932, chr6:72570561-7257106, chr6:72583406-72583906, chr6:72584081-72584581, chr6:72598767-72599267, chr6:72620191-72620691, chr6:72630335-72630835, chr6:72650361-72650861 and chr6:72653567-7265406.

Figure 4.2. *Tbx3* binding on the *Tcf7l1* locus. Pseudobulk accessibility profiles, ChIP-seq coverage (for *Hnf4a*, *Cebpa*, *Foxa1*, *Onecut1* and *Tbx3*), SCENIC+ region to gene links colored by correlation score and *Tcf7l1* and *Tbx3* expression across the zoned hepatocytes classes (from PP to PC) are shown.

Regarding *Tcf7l1*, we performed a ChIP-seq experiment using a goat anti-Tcf3 antibody (Santa Cruz Biotechnology, sc-8635). However, this antibody is not ChIP-grade and the experiment was not successful (resulted in background signal). Thus, we have not included this data in the manuscript. In addition, we have not found any public *Tcf7l1* ChIP-seq data set in hepatocytes (only in nephron progenitor cells (Guo et al. 2021), and stem cells (De Jaime-Soguero et al., 2017; Sierra et al., 2018, Mukherjee et al., 2022)). Nevertheless, SCENIC+ predicts 19 regions bound by *Tcf7l1* (based on the presence of *Tcf7l1* motifs) whose accessibility correlates with *Tbx3* expression and anti-correlates with *Tcf7l1* expression (Fig R4.3).

Figure 4.3. *Tcf7l1* binding on the *Tbx3* locus. Pseudobulk accessibility profiles, SCENIC+ predicted TFBSs (i.e. region-based eRegulon for *Hnf4a*, *Cebpa*, *Foxa1*, *Onecut1* and *Tcf7l1*), SCENIC+ region to gene links colored by correlation score and *Tcf7l1* and *Tbx3* expression across the zoned hepatocytes classes (from PP to PC) are shown.

We now include these plots in Supplementary Figure 10.

5. In the discussion, the authors proposed a feedback loop between *Tbx3* and *Tcf7l1* in hepatocyte zonation. There might be several issues of this model. First, not enough evidence was provided for the direct binding of *Tbx3* and *Tcf7l1*. Second, according to the illustration, *Tcf7l2* also directly regulates *Tbx3*, but no data in this paper supports this conclusion. Third, according to Fig 2a, *Tbx3* was highly expressed in pericentral hepatocytes. But the number of potential binding sites for *Tbx3* is very small compared with that in periportal hepatocyte. More evidence is needed to support that *Tbx3* directly binds to *Tcf7l1* with such limited binding options.

We believe our data supports the feedback loop between *Tcf7l1* and *Tbx3*:

- SCENIC+ identifies *Tbx3* as a target of *Tcf7l1* and vice versa. This means that there are periportally accessible regions with the *Tbx3* binding site within ± 150 kb of *Tcf7l1*, and that there are pericentrally accessible regions with the *Tcf7l1/2* site within ± 150 kb of *Tbx3*. In addition, *Tcf7l1* and *Tbx3* gene expression is anticorrelated.
- For *Tbx3*, we have generated a new ChIP-seq data set. We identified 9 regions linked to *Tcf7l1* bound by *Tbx3* (see comment 4).

- Tcf711 has been previously shown to repress Tbx3 (Athanasouli et al, 2023) and SCENIC+ predicts 19 Tcf711 binding sites in the Tbx3 locus that correlate with Tbx3 expression (and anti-correlates with Tcf711 expression).
- While Tbx3 is not found as a target of Tcf712 by SCENIC+ because gene expression does not correlate, activation of Tbx3 by WNT, depending on Tcf712, has been previously shown (Zimmerli et al, 2020). In addition, pericentrally accessible sites with Tcf711/2 motifs are found in the Tbx3 locus to which Tcf712 may bind. **We have made the arrow from Tcf712 to Tbx3 dashed in the figure.**

We agree with the reviewer that Tbx3 binding is not occurring in all periportal chancers, but rather a subset (i.e. Tbx3 is not the sole TF responsible for pericentral-periportal specific chromatin accessibility). We find that this set of regions is reproducible across mice, while other periportal regions may depend on other variable TFs, such as Egr1, Sox9 and Foxq1, as shown in Fig 2c. In addition, single cell data is very sparse, which reduces the sensitivity to detect negative correlations, and thus, may lead to false negatives. Note that in the Tbx3 ChIP-seq data, we only detected the Tbx3 motif enriched in the top 1,000 regions, where we identified 424 regions with the Tbx3 motif. **We have stressed this point in the discussion.** Altogether, we agree that further validation of these interactions could be performed, yet this is not the only nor main contribution of the manuscript. **We now include the figure and the potential model as Supplementary Figure 10.**

Minor:

1. In lines 131-134, the authors described smFISH experiment in the liver but referred to Fig 1f. Similar mistake for Fig 1g in lines 140-145.

We have fixed the errors and now refer to the correct panels (g and e-f, respectively).

2. In lines 253-265, the authors classified hepatocyte regulons based on their zonation state and mouse status using PCA. More methodology details would be helpful for the readers to understand this part.

We have elaborated in the methodology of this analysis in the methods section.

3. In lines 382-384, the authors stated that ‘DeepLiver predictions of the effect of enhancer mutations correlate with experimental results ($R=0.36-0.75$, Fig S17)’. But the Fig S17 showed

different correlation results. Same issue in lines 385-387, there are no negative correlation values in Fig S17.

The correlation values with DeepLiver range from 0.36 to 0.75, as shown in Fig S17e. We have replaced ‘R=0.36-0.75’ to ‘with a correlation ranging between 0.36 and 0.75’. With regards to lines 385-387, the correlation values from DeepLiver are 0.64 and 0.36 as shown in Fig S17e. The negative correlation values are for other methods (as shown in Kircher et al., 2019).

4. In lines 495-501, the authors tested selected enhancers and their Tbx3 LOF variants in HepG2 cells. It's worth to include Tcf711 enhancers to test the direct repression as described in lines 460-461.

Due to a mutation in beta-catenin, WNT signaling is active in HepG2. HepG2 expresses Tbx3 and exhibits a more pericentral identity (Ardisasmita et al. 2022). In fact, there is no expression on Tcf711 in HepG2. Hence, we can only assess Tbx3 LOF in HepG2 cells, but not Tcf711 LOF. In addition, assessing the effects of Tcf711 LOF is more difficult, as Tcf712 recognizes and binds to the same DNA sequence.

Decision Letter, first revision:

10th August 2023

Dear Stein,

Thank you for submitting your revised manuscript "Enhancer grammar of liver cell types and hepatocyte zonation states" (NCB-A50026A). It has now been seen by the original referees and their comments are below. The reviewers find that the paper has improved in revision, and therefore we'll be happy in principle to publish it in Nature Cell Biology, pending minor revisions to comply with our editorial and formatting guidelines.

If the current version of your manuscript is in a PDF format, please email us a copy of the file in an editable format (Microsoft Word or LaTeX)-- we cannot proceed with PDFs at this stage.

Thank you again for your interest in Nature Cell Biology. Please do not hesitate to contact me if you have any questions.

Best regards,
Stelios

Stylianos Lefkopoulos, PhD
He/him/his
Associate Editor
Nature Cell Biology
Springer Nature
Heidelberger Platz 3, 14197 Berlin, Germany

E-mail: stylianos.lefkopoulos@springernature.com
Twitter: @s_lefkopoulos

Reviewer #1 (Remarks to the Author):

The authors addressed all my concerns and should be congratulated to their elegant work!

Reviewer #2 (Remarks to the Author):

The authors fully addressed my comments. I would recommend this manuscript be published in

Nature Cell Biology.

Reviewer #3 (Remarks to the Author):

The authors have significantly improved the manuscript and included a wealth of novel data. They have addressed my major criticisms and now discussed the sex (gender) aspect of their liver study. They have analyzed the publically available snRNAseq data from female mouse livers. After data analysis they reported that neither the expression of the core hepatocyte TFs important for the MS, nor their targets are affected by gender and included novel panel in Figure and in Supplemental data. For the time being I think this is OK. However, experiments done in parallel, with same methods on both sexes, would be most relevant. The authors acknowledge that additional single-cell multi-omics data in female livers and/or disease models may be needed in the future to study sex-biased gene regulation in wild-type mice and upon disease at the enhancer-GRNs resolution. I certainly agree with this statement.

I also agree with deleting the circadian part of the study. The data are in the database (and in Supplementary info) and researchers interested in the clock part can find interesting genes. However, with only one time point we can monitor only changing of the amplitude that is affected by several factors and is usually not used as a measure of the circadian disruption. For the future, a circadian experiment would certainly be very important.

Concerning the criticism that only HepG2 cell line was used as a model to study enhancer functionality, the authors provided data about testing other cell models, also from the mouse, and concluded that HepG2 showed more accessibility in hepatocyte specific regions compared to the other cell lines, for the TFs of interest. This part is now well elaborated and documented and also included in the Supplementary data.

Authors, upon request, also compared their data with a similar recent study, which increased the impact and also underlines the novelty.

Reviewer #4 (Remarks to the Author):

Most of the concerns have been addressed. I don't have any further comments.

Decision Letter, final checks:

Our ref: NCB-A50026A

1st September 2023

Dear Dr. Aerts,

Thank you for your patience as we've prepared the guidelines for final submission of your Nature Cell

Biology manuscript, "Enhancer grammar of liver cell types and hepatocyte zonation states" (NCB-A50026A). Please carefully follow the step-by-step instructions provided in the attached file, and add a response in each row of the table to indicate the changes that you have made. Please also check and comment on any additional marked-up edits we have proposed within the text. Ensuring that each point is addressed will help to ensure that your revised manuscript can be swiftly handed over to our production team.

In recognition of the time and expertise our reviewers provide to Nature Cell Biology's editorial process, we would like to formally acknowledge their contribution to the external peer review of your manuscript entitled "Enhancer grammar of liver cell types and hepatocyte zonation states". For those reviewers who give their assent, we will be publishing their names alongside the published article.

Nature Cell Biology offers a Transparent Peer Review option for new original research manuscripts submitted after December 1st, 2019. As part of this initiative, we encourage our authors to support increased transparency into the peer review process by agreeing to have the reviewer comments, author rebuttal letters, and editorial decision letters published as a Supplementary item. When you submit your final files please clearly state in your cover letter whether or not you would like to participate in this initiative. Please note that failure to state your preference will result in delays in accepting your manuscript for publication.

Cover suggestions

COVER ARTWORK: We welcome submissions of artwork for consideration for our cover. For more information, please see our guide for cover artwork.

Nature Cell Biology has now transitioned to a unified Rights Collection system which will allow our Author Services team to quickly and easily collect the rights and permissions required to publish your work. Approximately 10 days after your paper is formally accepted, you will receive an email in providing you with a link to complete the grant of rights. If your paper is eligible for Open Access, our Author Services team will also be in touch regarding any additional information that may be required to arrange payment for your article.

Please note that *Nature Cell Biology* is a Transformative Journal (TJ). Authors may publish their research with us through the traditional subscription access route or make their paper immediately open access through payment of an article-processing charge (APC). Authors will not be required to make a final decision about access to their article until it has been accepted. Find out more about

Transformative Journals

Please use the following link for uploading these materials:
[Redacted]

Best regards,

Kendra Donahue
Staff
Nature Cell Biology

On behalf of

Stylios Lefkopoulos, PhD
He/him/his
Associate Editor
Nature Cell Biology
Springer Nature
Heidelberger Platz 3, 14197 Berlin, Germany

E-mail: stylios.lefkopoulos@springernature.com
Twitter: @s_lefkopoulos

Reviewer #1:

Remarks to the Author:

The authors addressed all my concerns and should be congratulated to their elegant work!

Reviewer #2:

Remarks to the Author:

The authors fully addressed my comments. I would recommend this manuscript be published in Nature Cell Biology.

Reviewer #3:

Remarks to the Author:

The authors have significantly improved the manuscript and included a wealth of novel data. They have addressed my major criticisms and now discussed the sex (gender) aspect of their liver study. They have analyzed the publically available snRNAseq data from female mouse livers. After data analysis they reported that neither the expression of the core hepatocyte TFs important for the MS, nor their targets are affected by gender and included novel panel in Figure and in Supplemental data. For the time being I think this is OK. However, experiments done in parallel, with same methods on both sexes, would be most relevant. The authors acknowledge that additional single-cell multi-omics data in female livers and/or disease models may be needed in the future to study sex-biased gene regulation in wild-type mice and upon disease at the enhancer-GRNs resolution. I certainly agree with this statement.

I also agree with deleting the circadian part of the study. The data are in the database (and in Supplementary info) and researchers interested in the clock part can find interesting genes. However, with only one time point we can monitor only changing of the amplitude that is affected by several factors and is usually not used as a measure of the circadian disruption. For the future, a circadian experiment would certainly be very important.

Concerning the criticism that only HepG2 cell line was used as a model to study enhancer functionality, the authors provided data about testing other cell models, also from the mouse, and concluded that HepG2 showed more accessibility in hepatocyte specific regions compared to the other cell lines, for the TFs of interest. This part is now well elaborated and documented and also included in the Supplementary data.

Authors, upon request, also compared their data with a similar recent study, which increased the impact and also underlines the novelty.

Reviewer #4:

Remarks to the Author:

Most of the concerns have been addressed. I don't have any further comments.

Author Rebuttal, first revision:

Reviewer #1:

Remarks to the Author:

The authors addressed all my concerns and should be congratulated to their elegant work!

We thank the reviewer for the positive assessment of our manuscript.

Reviewer #2:

Remarks to the Author:

The authors fully addressed my comments. I would recommend this manuscript be published in Nature Cell Biology.

We thank the reviewer for the positive assessment of our manuscript.

Reviewer #3:

Remarks to the Author:

The authors have significantly improved the manuscript and included a wealth of novel data. They have addressed my major criticisms and now discussed the sex (gender) aspect of their liver study. They have analyzed the publically available snRNAseq data form female mouse livers. After data analysis they reported that neither the expression of the core hepatocyte TFs important for the MS, nor their targets are affected by gender and included novel panel in Figure and in Supplementary data. For the time being I think this is OK. However, experiments done in parallel, with same methods on both sexes, would be most relevant. The authors acknowledge that additional single-cell multi-omics data in female livers and/or disease models may be needed in the future to study sex-biased gene regulation in wild-type mice and upon disease at the enhancer-GRNs resolution. I certainly agree with this statement.

Our analysis of publicly available snRNA-seq, projecting the inferred eGRNs in both male and female liver data, showed no differences for the core hepatocyte eGRNs. While we agree that performing additional experiments and analyses single-cell data on female and/or diseased livers could be relevant in the future, we believe that is out of the scope of this study, and we have addressed these limitations in the discussion. We are happy that the reviewer agrees with these changes.

I also agree with deleting the circadian part of the study. The data are in the database (and in Supplementary info) and researchers interested in the clock part can find interesting genes.

However, with only one time point we can monitor only changing of the amplitude that is

affected by several factors and is usually not used as a measure of the circadian disruption. For the future, a circadian experiment would certainly be very important.

We agree with the reviewer that additional experiments and analyses on the circadian rhythm could be relevant in the future. Nevertheless, we believe the starvation data is relevant to assess batch effects and identify the ‘core’ eGRN in this study, and could be expanded further in new studies. As the reviewer points out, we agree to keep this data in Supplementary Information for the interested audience.

Concerning the criticism that only HepG2 cell line was used as a model to study enhancer functionality, the authors provided data about testing other cell models, also from the mouse, and concluded that HepG2 showed more accessibility in hepatocyte specific regions compared to the other cell lines, for the TFs of interest. This part is now well elaborated and documented and also included in the Supplementary data.

We are happy that the reviewer is satisfied with the analysis and the detailed Supplementary Note.

Authors, upon request, also compared their data with a similar recent study, which increased the impact and also underlines the novelty.

We thank the reviewer for the assessment of our manuscript.

Reviewer #4:

Remarks to the Author:

Most of the concerns have been addressed. I don't have any further comments.

We thank the reviewer for the assessment of our manuscript.

Final Decision Letter:

Dear Stein,

I am pleased to inform you that your manuscript, "Single-cell spatial multi-omics and deep learning dissect enhancer-driven gene regulatory networks in liver zonation", has now been accepted for publication in Nature Cell Biology. Congratulations to you and the whole team!

Please note that *Nature Cell Biology* is a Transformative Journal (TJ). Authors may publish their research with us through the traditional subscription access route or make their paper immediately open access through payment of an article-processing charge (APC). Authors will not be required to make a final decision about access to their article until it has been accepted. Find out more about Transformative Journals

If you have not already done so, we strongly recommend that you upload the step-by-step protocols used in this manuscript to the Protocol Exchange (www.nature.com/protocolexchange), an open online resource established by Nature Protocols that allows researchers to share their detailed experimental know-how. All uploaded protocols are made freely available, assigned DOIs for ease of citation and are fully searchable through nature.com. Protocols and Nature Portfolio journal papers in which they are used can be linked to one another, and this link is clearly and prominently visible in the online versions of both papers. Authors who performed the specific experiments can act as primary authors for the Protocol as they will be best placed to share the methodology details, but the Corresponding Author of the present research paper should be included as one of the authors. By uploading your Protocols to Protocol Exchange, you are enabling researchers to more readily reproduce or adapt the methodology you use, as well as increasing the visibility of your protocols and papers. You can also establish a dedicated page to collect your lab Protocols. Further information can be found at www.nature.com/protocolexchange/about

With kind regards,
Stelios

Stylianos Lefkopoulos, PhD
He/him/his
Senior Editor, Nature Cell Biology
Springer Nature
Heidelberger Platz 3, 14197 Berlin, Germany

E-mail: stylianos.lefkopoulos@springernature.com
Twitter: @s_lefkopoulos
LinkedIn: [linkedin.com/in/stylianos-lefkopoulos-81b007a0](https://www.linkedin.com/in/stylianos-lefkopoulos-81b007a0)
